# Reconstructing voice identity from noninvasive auditory cortex recordings

**Charly Lamothe[1,2]\*, Etienne Thoret[1,2,3,4], Régis Trapeau[1], Bruno L Giordano[1], Julien Sein[1,5], Sylvain Takerkart[1], Stephane Ayache[2], Thierry Artieres[2,6]\*, Pascal Belin[1]\***

[1]La Timone Neuroscience Institute UMR 7289, CNRS, Aix-Marseille University, Marseille, France; [2]Laboratoire d'Informatique et Systèmes UMR 7020, CNRS, Aix-Marseille University, Marseille, France; [3]Perception, Representation, Image, Sound, Music UMR 7061, CNRS, Marseille, France; [4]Institute of Language Communication & the Brain, Marseille, France; [5]Centre IRM-INT@CERIMED, Marseille, France; [6]École Centrale de Marseille, Marseille, France

## eLife Assessment

This study used deep neural networks (DNN) to reconstruct voice information (viz., speaker identity), from fMRI responses in the auditory cortex and temporal voice areas, and assessed the representational content in these areas with decoding. A DNN-derived feature space approximated the neural representation of speaker identity-related information. The findings are **valuable** and the approach **solid**, yielding insight into how a specific model architecture can be used to relate the latent spaces of neural data and auditory stimuli to each other.

**\*For correspondence:**
charlylmth@gmail.com (CL);
thierry.artieres@univ-amu.fr (TA);
pascal.belin@univ-amu.fr (PB)

**Competing interest:** The authors declare that no competing interests exist.

**Abstract** The cerebral processing of voice information is known to engage, in human as well as non-human primates, 'temporal voice areas' (TVAs) that respond preferentially to conspecific vocalizations. However, how voice information is represented by neuronal populations in these areas, particularly speaker identity information, remains poorly understood. Here, we used a deep neural network (DNN) to generate a high-level, small-dimension representational space for voice identity—the 'voice latent space' (VLS)—and examined its linear relation with cerebral activity via encoding, representational similarity, and decoding analyses. We find that the VLS maps onto fMRI measures of cerebral activity in response to tens of thousands of voice stimuli from hundreds of different speaker identities and better accounts for the representational geometry for speaker identity in the TVAs than in A1. Moreover, the VLS allowed TVA-based reconstructions of voice stimuli that preserved essential aspects of speaker identity as assessed by both machine classifiers and human listeners. These results indicate that the DNN-derived VLS provides high-level representations of voice identity information in the TVAs.

## Introduction

The human voice carries speech, but is also an 'auditory face'uent that carries much valuable information on the stable physical characteristics of the speaker (hereafter, 'identity-related'; *Belin et al., 2004*; *Belin et al., 2011*). The ability of listeners to extract identity-related information from a voice such as gender, age, or unique identity even in brief stimuli plays a crucial role in our social interactions. Yet, its neural bases remain poorly understood compared to those of speech processing. Studies over the past two decades have clearly established via complementary neuroimaging techniques that the cerebral processing of voice information involves a set of temporal voice areas (TVAs)

in secondary auditory cortical regions of the human (fMRI: *Belin et al., 2000*; *Kriegstein and Giraud, 2004*; *Pernet et al., 2015*; EEG, MEG: *Charest et al., 2009*, *Capilla et al., 2013*, *Barbero et al., 2021*; Electrophysiology: *Rupp et al., 2022*, *Zhang et al., 2021*) as well as macaque (*Petkov et al., 2008*; *Bodin et al., 2021*) brain. The TVAs respond more strongly to sounds of voice – with or without speech (*Pernet et al., 2015*; *Rupp et al., 2022*; *Trapeau et al., 2022*)—and categorize voice apart from other sounds (*Bodin et al., 2021*) but the nature of the information encoded at these stages of cortical processing, especially with respect to speaker identity-related information, remains largely unknown (*Blank et al., 2014*; *Belin et al., 2018*).

In recent years, deep neural networks (DNNs) have emerged as a powerful tool for representing complex visual data, such as images (*LeCun et al., 2015*), videos (*Liu et al., 2020*), or audio (*Chorowski et al., 2019*). In the auditory domain, DNNs have been shown to provide valuable representations—so-called feature or latent spaces—for modeling the cerebral processing of sound (brain encoding; speech: *Kell et al., 2018*; *Millet et al., 2022*; *Tuckute et al., 2023*; semantic content: *Caucheteux et al., 2022*; *Caucheteux and King, 2022*; *Caucheteux et al., 2023*; *Giordano et al., 2023*; music: *Güçlü et al., 2016*), or reconstructing the stimuli heard by a participant (brain decoding) (*Akbari et al., 2019*). They have not yet been used to explain cerebral representations of identity-related information due in part to the focus on speech information (*von Kriegstein et al., 2003*).

Here, we addressed this challenge by training a 'Variational autoencoder' (VAE; *Kingma and Welling, 2014*) DNN to reconstruct voice spectrograms from 182,000 250 ms voice samples from 405 different speaker identities in 8 different languages from the CommonVoice database (*Ardila et al., 2020*). Brief (250ms) samples were used to emphasize speaker identity-related information in voice, already available after a few hundred milliseconds (*Schweinberger et al., 1997*; *Lavan, 2023*), over linguistic information unfolding over longer periods (word, >350ms; *Mcallister et al., 1994*). While 250ms is admittedly short compared to standards of, for example computational speaker identification that typically uses 2–3 s samples, this short duration is sufficient to allow near-perfect gender classification and performance levels well above chance for speaker discrimination. This brief duration allowed the presentation of many more stimuli to our participants in the scanner while preserving acceptable behavioral and classifier performance levels.

State-of-the-art studies have primarily relied on task-optimized neural networks (i.e. DNN trained using supervised learning to classify a category from the input) to study sensory cortex processes (*Yamins and DiCarlo, 2016*; *Schrimpf et al., 2018*). They can reach high accuracies in brain encoding (*Khaligh-Razavi and Kriegeskorte, 2014*; *Schrimpf et al., 2018*; *Han et al., 2019*). However, there is increasing evidence that unsupervised learning, such as that used for the VAE, also offers potentially valuable computational models for investigating brain processing (*Higgins et al., 2021*; *Zhuang et al., 2021*; *Millet et al., 2022*; *Millet et al., 2022*). Thus, latent spaces derived by a VAE, exploited within encoding, representational similarity, and decoding frameworks, offer a potentially promising tool for investigating the representations of voice stimuli in the secondary auditory cortex (*Naselaris et al., 2011*). Autoencoders learn to compress stimuli with high dimensionality into a lower-dimensional space that nonetheless allows reconstruction of the original stimuli via an inverse transformation learned by the second part of the network called the decoder.

We trained such a model to learn to compress spectrotemporal representations of voice samples into a *voice latent space* (VLS). In order to test whether VLS accounts well for cerebral activity in response to voice stimuli, we scanned three healthy volunteers using fMRI to measure an indirect index of their cerebral activity across 10+hours of scanning each in response to ~12,000 of the voice samples, denoted *BrainVoice* in the following (different from the ones used to train the DNN). The small number of participants does not allow for generalization at the general population level as in standard fMRI studies. However, it allows testing for replicability as in comparable studies involving 10+hours of scanning per participant (*VanRullen and Reddy, 2019*). Different stimulus sets were used across participants to provide a stringent test of replicability based on subject-level analyses. Stimuli consisted of randomly spliced 250 ms excerpts of speech samples from the CommonVoice database (*Ardila et al., 2020*) by 119 speakers in 8 languages.

We first asked how the VLS could account for the brain responses to speaker identities (encoding) measured in A1 and the TVAs in comparison with a linear autoencoder's latent space (LIN). This approach was chosen to compare a representation learned linearly under similar conditions (same input data, learning algorithm, reconstruction objective, and latent space size) with the VLS, which has non-linear transformations

and a regularized latent space. For this, we used a general linear model (GLM) of fMRI responses to the speaker identities, resulting in one voxel activity map per speaker (*Figure 2—figure supplement 1*). Then, we computed the average VLS coordinates of the fMRI voice stimuli for each speaker identity, which may be seen as a speaker representation in the VLS (see *Identity-based and stimulus-based representations* section). Next, we trained a linear voxel-based encoding model to predict the speaker voxel activity maps from the speaker's VLS coordinates. As VAE achieves compression through a series of nonlinear transformations (*Wetzel, 2017*), we choose to contrast its results with a linear autoencoder's latent space. This method has previously been applied to fMRI-based image reconstructions (*Cowen et al., 2014*; *VanRullen and Reddy, 2019*; *Mozafari et al., 2020*).

The extent to which the VLS allows linearly predicting the fMRI recordings does not provide insight into the representational geometries, i.e., the differences between the patterns of cerebral activity for speaker identity. We addressed this question by using representational similarity analysis (RSA; *Kriegeskorte et al., 2008*) to test which model better accounts for the representational geometry for voice identities in the auditory cortex. Using RSA as a model comparison framework is relevant to examining the brain-model relationship from complementary angles (*Diedrichsen and Kriegeskorte, 2017*; *Giordano et al., 2023*; *Tuckute et al., 2023*). We built speaker x speaker representational dissimilarity matrices (RDMs) capturing pairwise differences in cerebral activity or model predictions between all pairs of speakers; then, we examined how well the LIN and VLS-derived RDMs correlated with the cerebral RDMs from A1 and the TVAs.

A robust test of the validity of models of brain activity, and a long-standing goal in computational neurosciences, is the reconstruction of a stimulus presented to a participant from the evoked brain responses. While reconstruction of visual stimuli (images, videos) from cerebral activity has been performed by a number of groups (*VanRullen and Reddy, 2019*; *Mozafari et al., 2020*; *Le et al., 2022*; *Gaziv et al., 2022*; *Dado et al., 2022*; *Chen et al., 2023*), validating the DNN-derived representational spaces, comparable work in the auditory domain is scarce, almost exclusively concentrated on linguistic information (*Santoro et al., 2017*). *Akbari et al., 2019* used a DNN to reconstruct speech stimuli based on ECoG recording of auditory cortex activity, an invasive method compared to techniques like fMRI. They obtained a good phonetic recognition rate but chance-level gender categorization performance from reconstructed spectrograms and no evaluation of speaker identity discrimination.

Here, we built on the linear relationship uncovered in our encoding analysis between the VLS and the fMRI recordings to invert it and try to predict VLS coordinates from the recorded fMRI data; then, using the decoder, we reconstructed the spectrograms of stimuli presented to the participants (*Wu et al., 2006*; *Naselaris et al., 2011*). The voice identity information available in the reconstructed stimuli was finally assessed by human listeners using both machine learning classifiers and behavioral tasks.

## Results

### Voice information in the Voice Latent Space (VLS)

*Figure 1a* shows the architecture of the VAE, with its encoder that reduces an input spectrogram to a highly compressed, 128-dimension *voice latent space* (VLS) representation and its decoder that reconstructs the spectrogram from this VLS representation. We selected this latent space size as it was the first value that produced satisfactory reconstructions. Points in the VLS correspond to voice samples with different identities and phonetic content. A line segment in the VLS contains points corresponding to perceptual interpolations between its two extremities (*Figure 1b*; *Audio file 1*). VLS coordinates of samples presented to the participants averaged by speaker identity suggest that a major organizational dimension of the latent space is voice gender (*Figure 1b*; colored by age or language in *Figure 1—figure supplement 1*).

In order to probe the informational content of the VLS, linear classifiers were trained to categorize the voice stimuli from 405 speakers by gender (two classes), age (two classes: before/above 30 years old, the median age in our sample), or identity (119 classes, cf Methods) based on VLS coordinates, or their LIN features *Figure 1c, d and e*; we aggregated the stimuli from the three participants; for each model, we computed the latent space of each stimulus and averaged the latent spaces by speaker identity, leading to 405 128-dimensional vectors. We then trained linear classifiers using a fivefold cross-validation scheme (see *Characterization of the autoencoder latent space*). The decoding

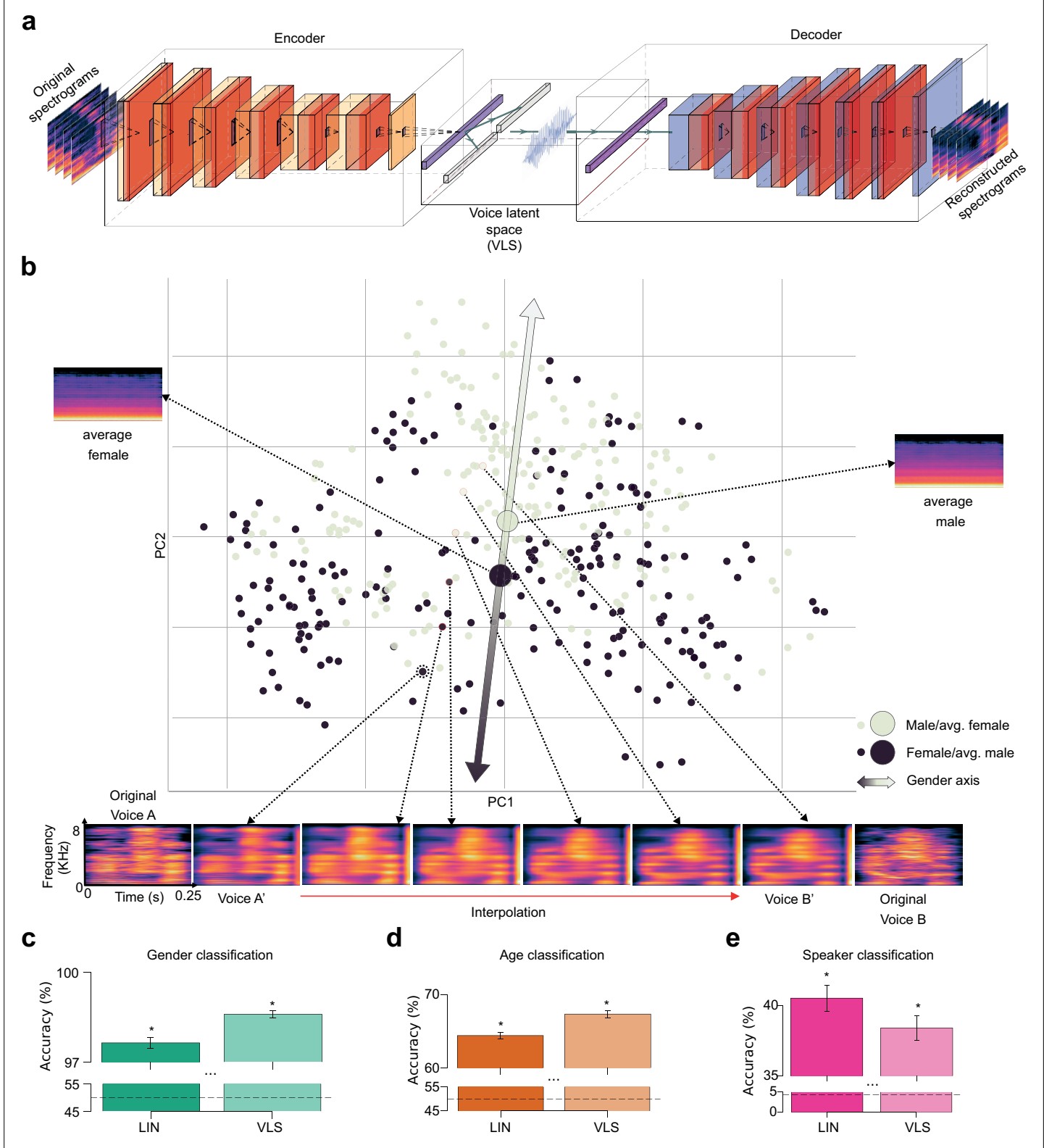

**Figure 1.** DNN-derived Voice Latent Space (VLS). (**a**) Variational autoencoder (VAE) Architecture. Two networks learned complementary tasks. An encoder was trained using 182 K voice samples to compress their spectrogram into a 128-dimension representation, the voice latent space (VLS), while a decoder learned the reverse mapping. The network was trained end-to-end by minimizing the difference between the original and reconstructed spectrograms. (**b**) Distribution of the 405 speaker identities along the first 2 principal components of the VLS coordinates from all sounds, averaged

*Figure 1 continued on next page*

*Figure 1 continued*

by speaker identity. Each disk represents a speaker's identity colored by gender. PC2 largely maps onto voice gender (ANOVAs on the first two components: PC1: F(1, 405)=0.10, p=0.74; PC2: F(1, 405)=11.00, p<0.001). Large disks represent the average of all male (black) or female (gray) speaker coordinates, with their associated reconstructed spectrograms (note the flat fundamental frequency ($f_0$) and formant frequency contours caused by averaging). The bottom of the spectrograms illustrates an interpolation between stimuli of two different speaker identities: spectrograms at the extremes correspond to two original stimuli (**A, B**) and their VLS-reconstructed spectrograms (**A', B'**). Intermediary spectrograms were reconstructed from linearly interpolated coordinates between those two points in the VLS (red line) (*Audio file 1*). (**c, d, e**) Performance of linear classifiers at categorizing speaker gender (chance level: 50%), age (young/adult, chance level: 50%), or identity (119 identities, chance level: 0.84%) based on VLS or Linear model (LIN) coordinates. Error bars indicate the standard error of the mean (s.e.m.) across fivefolds. All ps <0.05. The horizontal black dashed lines indicate chance levels. *: p<0.05.

The online version of this article includes the following figure supplement(s) for figure 1:

**Figure supplement 1.** Projections of the DNN-derived Voice Latent Space (VLS).

accuracy was significantly above chance level (Wilcoxon signed-rank test, all W=15, p=0.03125) for all classifications (LIN: gender (mean accuracy ±s.d.)=97.64 ± 1.77%; age: 64.39 ± 4.54%; identity: 40.52 ± 9.14%; VLS: gender: 98.59 ± 1.19%; age: 67.31 ± 4.86%; identity: 38.40 ± 8.75%).

Thus, despite its low number of dimensions (each input spectrogram has 401x21 = 8421 parameters and is summarized in the VLS by a mere 128 dimensions), the VLS appears to meaningfully represent the different sources of voice information perceptually available in the vocal stimuli. This representational space, therefore, constitutes a relevant candidate for linearly modeling voice stimulus representations by the brain.

## Brain encoding

For assessing generalization performances of decoding models and brain-based reconstruction, six test stimuli were repeated more often (60 times) for each participant to provide robust estimates of their induced cerebral activity (see Methods). We first modeled these responses to voice using a general linear model (GLM; *Friston et al., 1994*) with several nuisance regressors as an initial denoising step (*Figure 2—figure supplement 2*), then used a second GLM modeling cerebral responses to the different speaker identities (*Figure 2—figure supplement 1*), resulting in one voxel activity map per speaker (*Figure 2—figure supplement 1*). We independently localized in each participant several regions of interest (ROIs) on which subsequent analyses were focused: the anterior, middle, and posterior TVAs in each hemisphere (individually localized via an independent 'voice localizer scan' and MNI coordinates provided in *Pernet et al., 2015*; *Figure 2—figure supplement 1*) as well as primary auditory cortex (A1) using a probabilistic map in MNI space (*Penhune et al., 1996*; *Figure 2—figure supplement 1d*).

We used a linear voxel-based encoding model to test whether VLS linearly maps onto cerebral responses to speaker identities measured with fMRI in the different ROIs. A regularized linear regression model (Methods) was trained on a subset of the data (fivefold cross-validation scheme) to predict the voxel maps for each speaker identity. For each fold, the trained model was tested on the held-out speaker identities (*Figure 2a*). The model's performance was assessed for each ROI using the Pearson correlation score between each voxel's actual and predicted responses (*Schrimpf et al., 2021*). Similar predictions were tested with features derived from LIN, as well as with more recent unsupervised/self-supervised speech models that have been shown to align to auditory responses to speech in the temporal cortex: Wav2Vec (*Millet et al., 2022*) and HuBERT (*Li et al., 2023*; *Figure 2—figure supplement 3*). *Figure 2b* shows the distribution of correlation coefficients obtained for each of the ROIs for the 2 sets of features across voxels, hemispheres, and participants.

One-sample t-tests showed that the means of Fisher z-transformed coefficients for both LIN features and VLS were significantly higher than zero (LIN: A1 t(197)=7.25, p<0.0001, pTVA t(175)=4.49, p<0.0001, mTVA t(164)=9.12, p<0.0001 and aTVA t(147)=6.81, p<0.0001; VLS: A1 t(197)=4.76, p<0.0001, mTVA t(164)=10.12, p<0.0001 and aTVA t(147)=5.52, p<0.0001 but not pTVA t(175)=-1.60; *Appendix 1—table 2* and *Appendix 1—table 3*).

A mixed ANOVA performed on the Fisher z-transformed coefficients with Feature (VLS, LIN) and ROI (A1, pTVA, mTVA, aTVA) as factors showed a significant effect of Feature (F(3, 683)=56.65, p<0.0001), a significant effect of ROI (F(3, 683)=18.50, p<0.0001), and a moderate interaction Feature x ROI (F(3, 683)=5.25, p<0.01). Post-hoc comparisons revealed that the mean of correlation coefficients

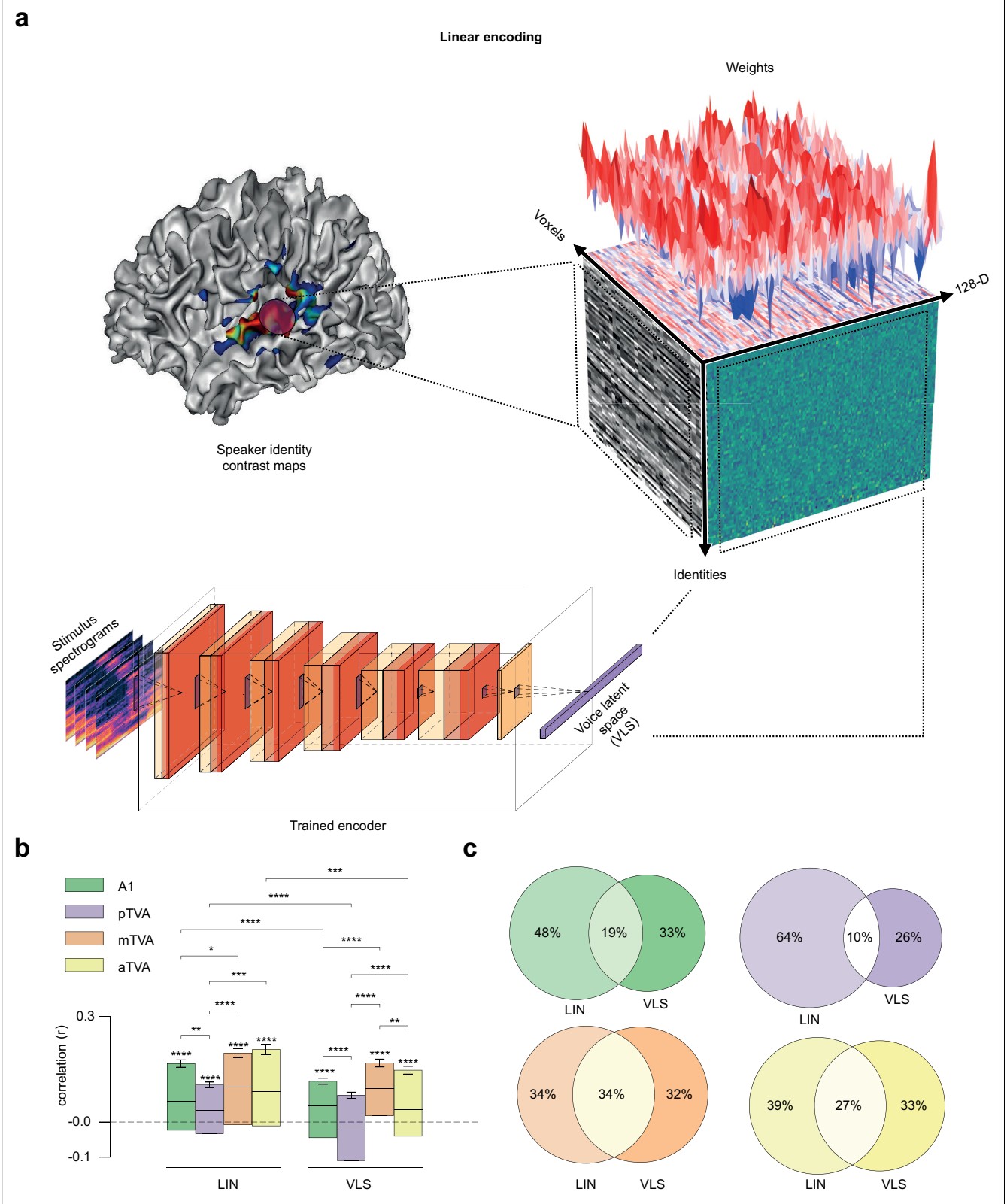

**Figure 2.** Predicting brain activity from the VLS. (**a**) Linear brain activity prediction from VLS for ~135 speaker identities in the different ROIs. We first fit a GLM to predict the Blood Oxygenation Level-Dependent (BOLD) responses to each voice speaker identity. Then, using the trained encoder, we computed the average VLS coordinates of the voice stimuli presented to the participants based on speaker identity. Finally, we trained a linear voxel-based encoding model to predict the speaker voxel activity maps from the speaker VLS coordinates. The cube illustrates the linear relationship between

*Figure 2 continued on next page*

*Figure 2 continued*

the fMRI responses to speaker identity and the VLS coordinates. The left face of the cube represents the activity of the voxels for each speaker's identity, with each line corresponding to one speaker. The right face displays the VLS coordinates for each speaker's identity. The cube's top face shows the encoding model's weight vectors. (**b**) Encoding results. For each region of interest, the model's performance was assessed using the Pearson correlation score between the true and the predicted responses of each voxel on the held-out speaker identities. Pearson's correlation coefficients were computed for each voxel on the speakers' axis and then averaged across hemispheres and participants. Similar predictions were tested with the LIN features. Error bars indicate the standard error of the mean (s.e.m) across voxels. *p<0.05; **p<0.01; **p<0.001; ****p<0.0001. (**c**) Venn diagrams of the number of voxels in each ROI with the LIN, the VLS, or both models. For each ROI and each voxel, we checked whether the test correlation was higher than the median of all participant correlations (intersection circle), and if not, which model (LIN or VLS) yielded the highest correlation (left or right circles).

The online version of this article includes the following figure supplement(s) for figure 2:

**Figure supplement 1.** Brain activity in response to voice measured by fMRI.

**Figure supplement 2.** Denoising of the fMRI BOLD responses.

**Figure supplement 3.** Extended predicted brain encoding results with state-of-the-art models.

was higher for LIN than for VLS in A1 (t(197)=4.02, p<0.0001), pTVA (t(175)=6.64, p<0.0001), aTVA (t(147)=3.78, p<0.001) but not in mTVA (t(164)=0.58) (*Appendix 1—table 4*); and that the voxel patterns are better predicted in mTVA than in A1 for both models (LIN: t(361)=2.36, p<0.05; VLS: t(361)=4.91, p<0.0001; *Appendix 1—table 5*). However, inspecting the distribution of model-voxel correlations, we found that both models account for different parts of the voice identity responses and differ across ROIs (*Figure 2c*).

## Representational similarity analysis (RSA)

For RSA, we built speaker x speaker representational dissimilarity matrices (RDMs), capturing for each ROI the dissimilarity in voxel space between each pair of speaker voxel maps ('brain RDMs'; Methods) using Pearson's correlation (*Walther et al., 2016*). We compared these four bilateral brain RDMs (A1, aTVA, mTVA, pTVA) to two 'model RDMs' capturing speaker pairwise feature differences predicted by LIN and the VLS (*Figure 3a*) built using cosine distance (*Xing et al., 2015*; *Bhattacharya et al., 2017*; *Wang et al., 2018*). *Figure 3b* shows for each ROI the Spearman correlation coefficients between the brain RDMs and the two model RDMs for each participant and hemisphere (*Kriegeskorte et al., 2008*; *Figure 3c* for an example of brain-model correlation).

To correct for multiple comparisons across models, these brain-model correlation coefficients were compared to zero using a 'maximum statistics' approach based on random permutations of the model RDMs' rows and columns (*Maris and Oostenveld, 2007*; Methods; *Figure 3b*). For the LIN model, only one brain-model RDM correlation was significantly different from zero (one-tailed test, corrected for multiple comparisons across models): in mTVA, right hemisphere in S3 (p=0.0500). For the VLS model, in contrast, five significant brain-model RDM correlations were observed in all four ROIs: in A1, right hemisphere in S3 (p=0.0142); pTVA: right hemisphere in S3 (p=0.0160); mTVA: left hemisphere in S3 (p=0.007); aTVA: left hemispheres in S1 (p=0.0417) and S3 (p=0.0001; *Appendix 1—table 6*).

A two-way repeated-measures ANOVA with Feature (VLS, LIN) and ROI (A1, pTVA, mTVA, aTVA) as factors performed on the Fisher z-transformed correlation coefficients showed a tendency towards a significant effect of Feature (F(1, 2)=22.53, p=0.04), and no ROI (F(3, 6)=1.79, p=0.30) or interaction effects (F(3, 6)=1.94, p=0.22). We compared the correlation coefficients between the VLS and LIN models within participants and hemispheres using one-tailed tests, based on the a priori hypothesis that the VLS models would exhibit greater brain-model correlations than the LIN models (Methods). The results revealed two significant differences in one of the three participants, both favoring the VLS model (S3: right pTVA, p=0.0366; left aTVA, p=0.00175; *Appendix 1—table 7*).

## Decoding and reconstruction

We finally inverted the brain-VLS relationship to predict linearly VLS coordinates based on fMRI measurements (*Figure 4a*; see 'Brain decoding' in Methods) and reconstructed via the trained decoder the spectrograms of 18 Test Stimuli (3 participants x 6 stimuli per participant; see *Figure 4b*, and Supplementary Audio 2; audio estimated from spectrogram through phase reconstruction).

We first assessed the nature of the reconstructed stimuli by using a DNN trained to categorize natural audio events (*Howard et al., 2017*): all reconstructed versions of the 18 Test Stimuli were categorized as 'speech' (1 class out of 521 no 'voice' classes). To evaluate the preservation of voice identity

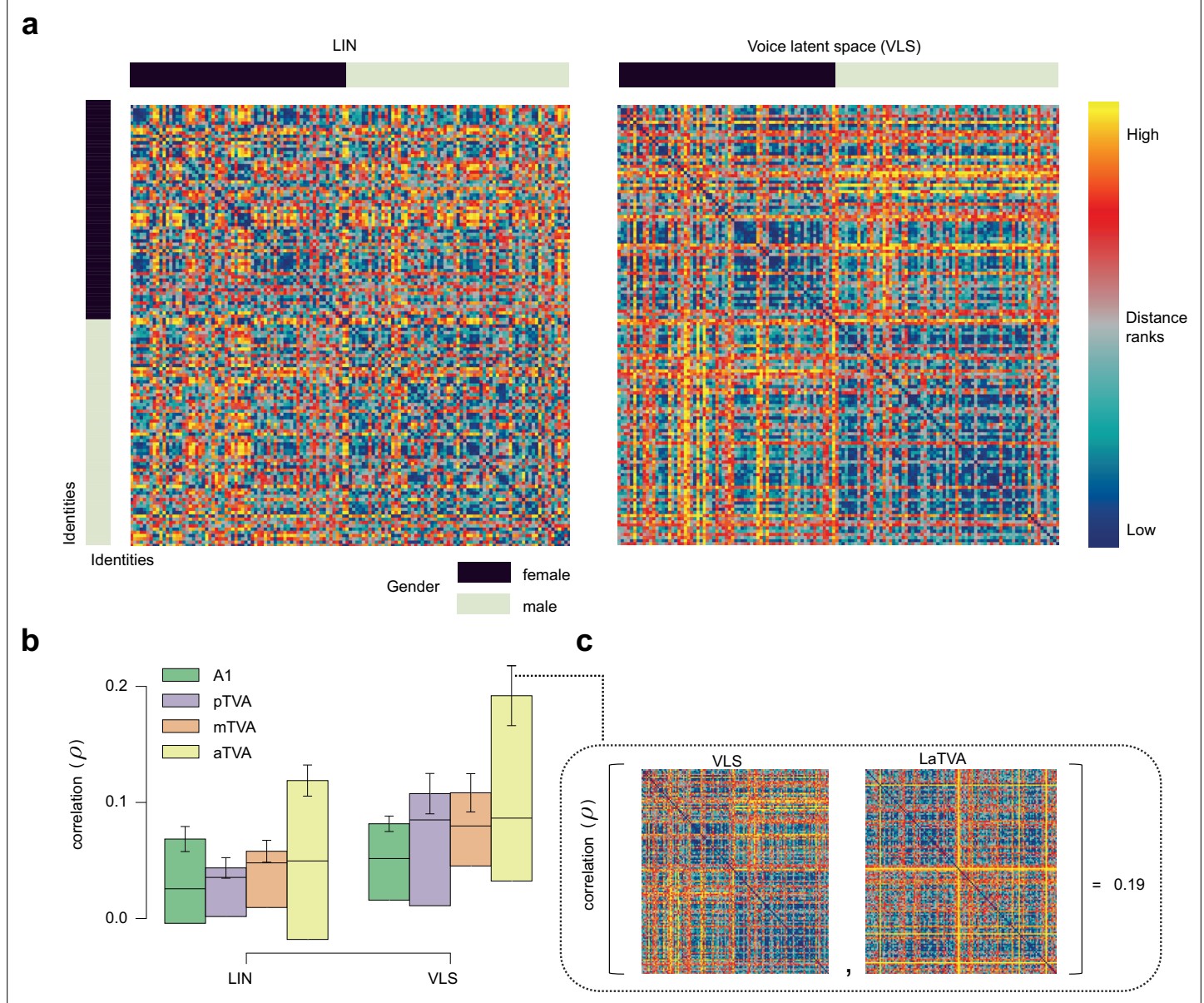

**Figure 3.** The VLS better explains representational geometry for voice identities in the TVAs than the linear model. (**a**) Representational dissimilarity matrices (RDMs) of pairwise speaker dissimilarities for ~135 identities (arranged by gender, sidebars), according to LIN and VLS. (**b**) Spearman correlation coefficients between the brain RDMs for A1, the 3 TVAs, and the 2 model RDMs. Error bars indicate the standard error of the mean (s.e.m) across brain-model correlations. (**c**) Example of brain-model RDM correlation in the TVAs. The VLS RDM and the brain RDM yielding one of the highest correlations (LaTVA) are shown in the insert.

information in the reconstructed voices, pre-trained linear classifiers were used to classify the speaker gender (2 classes), age (2 classes), and identity (17 classes; one identity was shared across participants) of the 18 reconstructed Test Stimuli. The mean of the accuracy distribution obtained across random classifier initializations (20 per ROI) used on the stimuli reconstructed from the induced brain activity was significantly above chance level for gender (LIN: pTVA (mean accuracy ±s.d.): 72.08±5.48, t(39)=25.15; VLS: A1: 61.11±2.15, t(39)=32.25; pTVA: 63.89±2.78, t(39)=31.22), age (LIN: pTVA: 54.58±4.14, t(39)=6.90; aTVA: 63.96±12.55, t(39)=6.94; VLS: pTVA: 65.00±7.26, t(39)=12.89; aTVA: 60.42±5.19, t(39)=12.54), and identity (LIN: A1: 9.20±9.23, t(39)=2.24; pTVA: 9.48±4.90, t(39)=4.59; aTVA: 9.41±6.28, t(39)=3.51; VLS: pTVA: 16.18±7.05, t(39)=9.11; aTVA: 8.23±4.70, t(39)=3.12) (*Figure 5a-c*; *Appendix 1—table 8*; *Appendix 1—table 9*;*Appendix 1—table 10*).

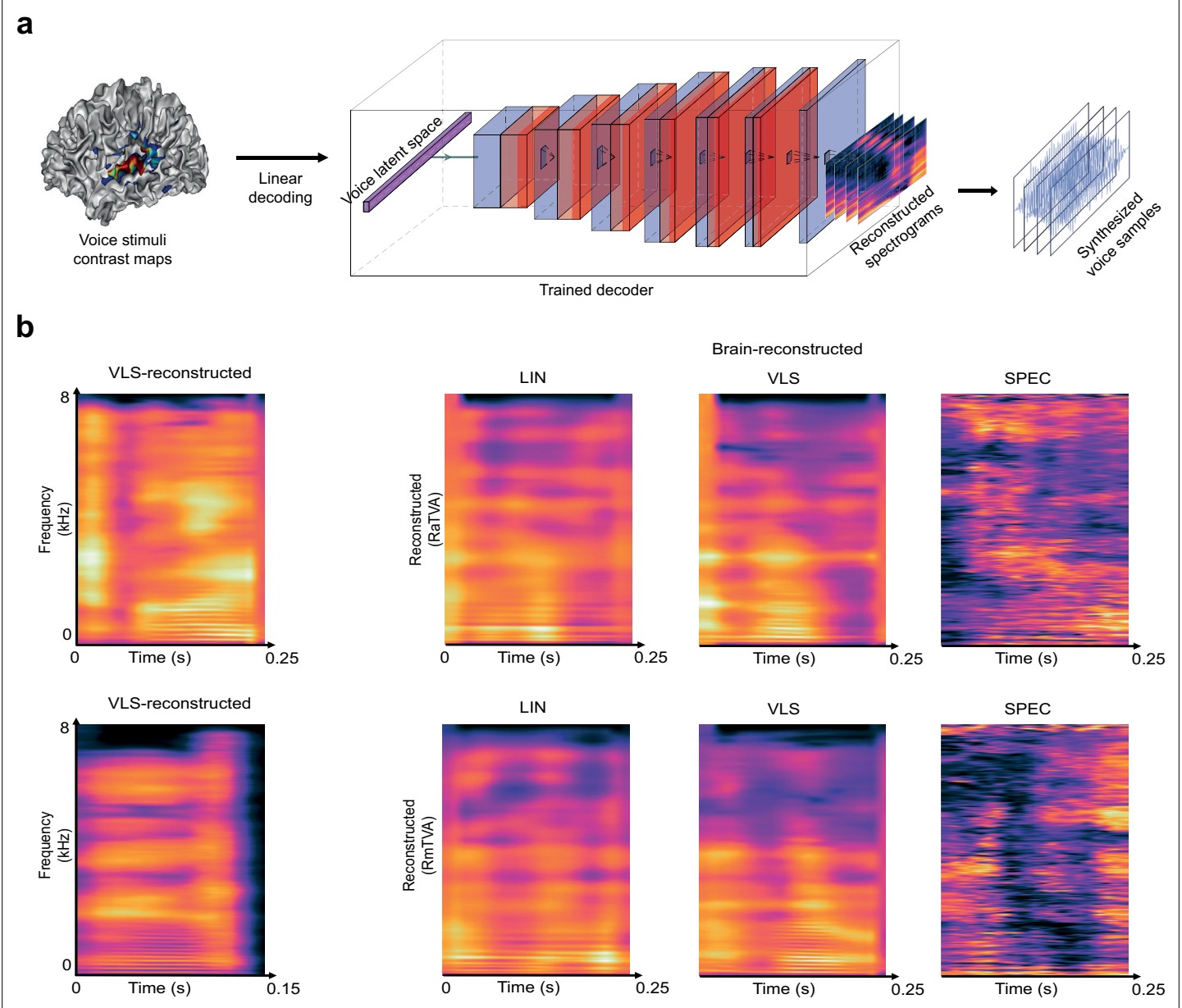

**Figure 4.** Reconstructing voice identity from brain recordings. (**a**) A linear voxel-based decoding model was used to predict the VLS coordinates of 18 Test Stimuli based on fMRI responses to ~12,000 Train stimuli in the different ROIs. To reconstruct the audio stimuli from the brain recordings, the predicted VLS coordinates were then fed to the trained decoder to yield reconstructed spectrograms, synthesized into sound waveforms using the Griffin-Lim phase reconstruction algorithm (*Griffin and Jae, 1983*). (**b**) Reconstructed spectrograms of the stimuli presented to the participants. The left panels show the spectrogram of example original stimuli reconstructed from the VLS, and the right panels show brain-reconstructed spectrograms via LIN or VLS autoencoder-based representations, and SPEC, direct regression from the audio spectrograms (*Audio file 2*).

Two-way ANOVAs with Feature (VLS, LIN) and ROI (A1, pTVA, mTVA, aTVA) as factors performed on classification accuracy scores (gender, age, identity) revealed for gender classifications significant effects of Feature $F_{(1, 312)}=12.82$, $p<0.0005$ and ROI (gender: $F_{(3, 312)}=245.06$, $p<0.0001$; age: $F_{(3, 312)}=64.49$, $p<0.0001$; identity: $F_{(3, 312)}=14.49$, $p<0.0001$), as well as Feature x ROI interactions (gender: $F_{(3, 312)}=56.74$, $p<0.0001$; age: $F_{(3, 312)}=4.31$, $p<0.001$; identity: $F_{(3, 312)}=8.82$, $p<0.0001$). Post-hoc paired t-tests indicated that the VLS was better than LIN in preserving gender, age, and identity information in at least one TVA compared with A1 (gender: aTVA: $t_{(39)}=5.13$, $p<0.0001$; age: pTVA: $t_{(39)}=9.78$, $p<0.0001$; identity: pTVA: $t_{(39)}=4.01$, $p<0.0005$; all tests in *Appendix 1—table 11*). Post-hoc two-sample t-tests comparing ROIs revealed significant differences in all classifications, in

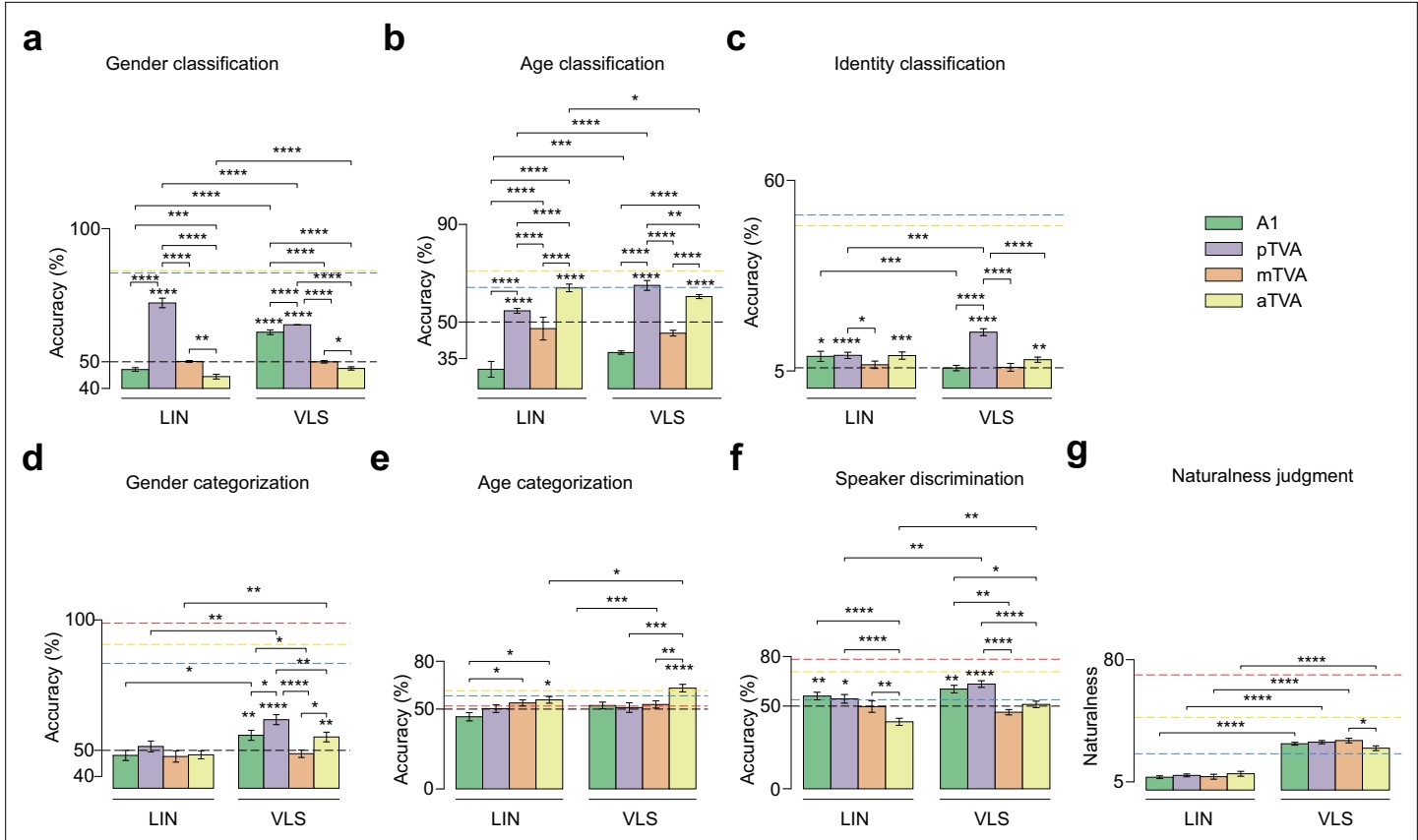

**Figure 5.** Behavioral and machine classification of the reconstructed stimuli. (**a, b, c**). Decoding voice identity information in brain-reconstructed spectrograms. Performance of linear classifiers at categorizing speaker gender (chance level: 50%), age (chance level: 50%), and identity (17 identities, chance level: 5.88%). Error bars indicate s.e.m. across 40 random classifier initializations per ROI (instance of classifiers; 2 hemispheres x 20 seeds). The horizontal black dashed line indicates the chance level. The blue and yellow dashed lines indicate the LIN and VLS ceiling levels, respectively. *p<0.05; **p<0.001, ***p<0.001; ****p<0.0001. (**d, e, f**) Listener performance at categorizing speaker gender (chance level: 50%) and age (chance level: 50%), and at identity discrimination (two forced choice task, chance level: 50%) in the brain-reconstructed stimuli. Error bars indicate s.e.m. across participant scores. The horizontal black dashed line indicates the chance level, while the red, blue, and yellow dashed lines indicate the ceiling levels for the original stimuli, the LIN-reconstructed, and the VLS-reconstructed, respectively. *p<0.05; **p<0.01; ***p<0.001, ***p<0.0001. (**g**) Perceptual ratings of voice naturalness in the brain-reconstructed stimuli' as assessed by human listeners, between 0 and 100 (zoomed between 5 and 80). *p<0.05, ****p<0.0001.

particular with pTVA outperforming other ROIs in gender (LIN: pTVA vs A1: t(78)=22.40, p<0.0001; pTVA vs mTVA: t(78)=10.92, p<0.0001; pTVA vs aTVA: t(78)=31.47, p<0.0001; VLS: pTVA vs A1: t(78)=4.94, p<0.0001; pTVA vs mTVA: t(78)=13.96, p<0.0001; pTVA vs aTVA: t(78)=22.06, p<0.0001), age (LIN: pTVA vs A1: t(78)=7.26, p<0.0001; pTVA vs mTVA: t(78)=10.11, p<0.0001; VLS: pTVA vs A1: t(78)=5.71, p<0.0001; pTVA vs mTVA: t(78)=10.11, p<0.0001; pTVA vs aTVA: t(78)=3.21, p<0.005) and identity (LIN: pTVA vs mTVA: t(78)=2.27, p<0.05; VLS: pTVA vs A1: t(78)=6.45, p<0.0001; pTVA vs mTVA: t(78)=6.62, p<0.0001; pTVA vs aTVA: t(78)=5.85, p<0.0001; *Appendix 1—table 12*).

We further evaluated voice identity information in the reconstructed stimuli by testing human participants (n=13) in a series of 4 online experiments assessing the reconstructed stimuli on (i) naturalness judgment, (ii) gender categorization, (iii) age categorization, and (iv) speaker categorization (Methods). The naturalness rating task showed that the VLS-reconstructed stimuli sounded more natural compared to LIN-reconstructed ones, as revealed by a two-way repeated-measures ANOVA (factors: Feature and ROI) with a strong effect of Feature (F(1, 12)=53.72, p<0.0001) and a small ROI x Feature interaction (F(3, 36)=5.36, p<0.005). Post-hoc paired t-tests confirmed the greater naturalness of VLS-reconstructed stimuli in both A1 and the TVAs (all ps <0.0001; *Figure 5g*). For the gender task, one-sample t-tests showed that categorization of the reconstructed stimuli was only significantly above chance level for the VLS (A1: (mean accuracy ±s.d.) 55.77±10.84, t(25)=2.66, p<0.01; pTVA: 61.75±7.11, t(25)=8.26, p<0.0001; aTVA: 55.13±9.23, t(25)=2.78, p<0.01). Regarding the age

and speaker categorizations, results also indicated that both the LIN- and VLS-reconstructed stimuli yielded above-chance performance in the TVAs (age: LIN: aTVA, 55.77±14.95, t(25)=1.93, p<0.05; VLS: aTVA, 63.14±11.82, t(25)=5.56, p<0.0001; identity: LIN: pTVA: 54.38±9.34, t(17)=1.93, p<0.05; VLS: pTVA: 63.33±6.75, t(17)=8.14, p<0.0001; *Appendix 1—Tables 13–15*). Two-way repeated-measures ANOVAs revealed a significant effect of ROI for all categories (gender: F(3, 27)=5.90, p<0.05; age: F(3, 36)=14.25, p<0.0001; identity: F(3, 24)=38.85, p<0.0001), and a Feature effect for gender (F(1, 9)=43.61, p<0.0001) and identity (F(1, 8)=14.07, p<0.001), but not for age (F(1, 12)=4.01, p=0.07), as well as a ROI x Feature interaction for identity discrimination (F(3, 24)=3.52, p<0.05; *Appendix 1— table 16* and *Appendix 1—table 17* for the model and ROI comparisons).

## Discussion

In this study, we examined to what extent the cerebral activity elicited by brief voice stimuli can be explained by machine-learned representational spaces, specifically focusing on identity-related information. We trained a linear model and a DNN model to reconstruct 100,000 s of short voice samples from 100+speakers, providing low-dimensional spaces (LIN and VLS), which we related to fMRI measures of cerebral response to thousands of experimental stimuli. We find: (i) that 128 dimensions are sufficient to explain a portion of the brain activity elicited by the voice samples and yield brain-based voice reconstructions that preserve identity-related information; (ii) that the DNN-derived VLS shows improved performance relative to the LIN space, particularly in yielding more brain-like representational spaces and more naturalistic voice reconstructions; (iii) that different ROIs have different degrees of brain-model relationship, with marked differences between A1 and the a, m, and pTVAs.

Low-dimensional spaces generated by machine learning have been used to approximate cerebral face representations and reconstruct recognizable faces based on fMRI (*VanRullen and Reddy, 2019*; *Dado et al., 2022*). In the auditory domain, however, they have mainly been used with a focus on linguistic (speech) information, ignoring identity-related information (but see *Akbari et al., 2019*). Here, we applied them to brief voice stimuli with minimal linguistic content but already rich identity-related information and found that as little as 128 dimensions account reasonably well for the complexity of cerebral responses to thousands of these voice samples as measured by fMRI (*Figure 2*). LIN and VLS both showed brain-like representational geometries, particularly the VLS in the aTVAs (*Figure 3*). They made possible what is, to our knowledge, the first fMRI-based voice reconstructions to preserve voice-related identity information such as gender, age, or even individual identity, as indicated by above-chance categorization or discrimination performance by both machine classifiers (*Figure 5a–c*) and human listeners (*Figure 5d–f*).

Estimation of fMRI responses (encoding) by LIN yielded correlations largely comparable to those by VLS (*Figure 2b*), although many voxels were only explained by one or the other space (*Figure 2c*). However, in the RSA, VLS yielded higher overall correlations with brain RDMs (*Figure 3*), suggesting a representational geometry closer to that instantiated in the brain than LIN. Further, VLS-reconstructed stimuli sounded more natural than the LIN-reconstructed ones (*Figure 5g*) and yielded both the best speaker discrimination by listeners (*Figure 5f*) and speaker classification by machine classifiers (*Figure 5c*). Unlike LIN, which was generated via linear transforms, VLS was obtained through a series of nonlinear transformations (*Wetzel, 2017*). The fact that the VLS outperforms LIN in decoding performance indicates that nonlinear transformation is required to better account for the brain representation of voices (*Naselaris et al., 2011*; *Cowen et al., 2014*; *Han et al., 2019*).

Comparisons between ROIs revealed important differences between A1 and the a, m, and pTVAs. For both LIN and VLS, fMRI signal (encoding) predictions were more accurate for the mTVAs than for A1, and for A1 than for the pTVAs (*Figure 2b*). The aTVAs yielded the highest correlations with the models in the RSA (*Figure 3*). Stimulus reconstructions (*Figure 4*) based on the TVAs also yielded better gender, age, and identity classification than those based on A1, with gender and identity best preserved in the pTVA- and, to a lesser extent, in the aTVA-based reconstructions (*Figure 5*). These results show that the a and pTVAs not only respond more strongly to vocal sounds than A1, but they also represent identity-related information in voice better than mTVA, which was previously anticipated in some neuroimaging studies (Gender: *Charest et al., 2013*; Identity: *Belin and Zatorre, 2003*; *Maguinness et al., 2018*; *Roswandowitz et al., 2018*; *Aglieri et al., 2021*). Moreover, several recent studies, using intracranial recordings, either through ECoG electrode grids (*Zhang et al., 2021*) or sEEG recordings (*Rupp et al., 2022*), found evidence that supports the idea of a hierarchical

organization of voice patches in the temporal lobe, where the information flow starts from the mTVA patches and moves in two directions: one from mTVA to the anterior TVA (aTVA) and the other one from mTVA to posterior TVA (pTVA).

Overall, we show that a DNN-derived representational space provides an interesting approximation of the cerebral representations of brief voice stimuli that can preserve identity-related information. We find it remarkable that such results could be obtained to explain sound representations despite the poor temporal resolution of fMRI. Future work combining more complex architectures to time-resolved measures of cerebral activity, such as magneto-encephalography (*Défossez et al., 2023*) or ECoG (*Pasley et al., 2012*), will likely yield better models of the cerebral representations of voice information.

## Methods

### Experimental procedure overview

Three participants attended 13 MRI sessions each. The first session was dedicated to acquiring high-resolution structural data, as well as to identifying the voice-selective areas of each participant using a 'voice localizer' based on different stimuli than those in the same experiment (*Pernet et al., 2015*; see below).

The next 12 sessions began with the acquisition of two fast structural scans for inter-session realignment purposes, followed by six functional runs, during which the main stimulus set of the experiment was presented. Each functional run lasted approximately 12 min, during which 240 experimental stimuli were presented in a rapid event-related design with a jittered inter-stimulus interval of 2.8–3.2 s.

Participants 1 and 2 attended all scanning sessions (72 functional runs in total); due to unforeseen personal health reasons, Participant 3's participation was limited to 24 runs.

Participants were instructed to stay still in the scanner while listening to the stimuli. To maintain participants' awareness during functional scanning, they were asked to press an MRI-compatible button each time they heard the same stimulus two times in a row, a rare event occurring 3% of the time (correct button hits [median accuracy ±s.d.]: S1=96.67 ± 7.10, S2=100.00 ± 0.89, S3=95.00 ± 3.68).

Scanning sessions were spaced apart by at least 2 days to avoid possible auditory fatigue due to the exposure to scanner noise. To ensure that participants' hearing abilities did not vary across scanning sessions, hearing thresholds were measured before each session using a standard audiometric procedure (*Martin and Champlin, 2000*; ISO 2004) and compared with the thresholds obtained prior to the first session.

### Participants

This study was part of the project 'Réseaux du Langage' and was promoted by the National Center for Scientific Research (CNRS). It has been given favorable approval by the local ethics committee (Comité de Protection des Personnes Sud-Méditerranée) on the date of 13th February 2019. The National Agency for Medicines (ANSM) has been informed of this study, which is registered under the number 2017-A03614-49. Three native French human speakers were scanned (all females, 26–33 years old). Participants gave written informed consent and received a compensation of 40€ per hour for their participation. All were right-handed, and no one had a hearing disorder or neurological disease. All participants had normal hearing thresholds below 15 dB HL for octave frequencies between 0.125 and 8 kHz.

### Stimuli

The auditory stimuli were divided into two sequences. One 'voice localizer' sequence to identify the voice-selective areas of each participant (*Pernet et al., 2015*) and 'main voice stimuli'.

#### Voice localizer stimuli

The voice localizer stimuli consisted of 96 complex sounds of 500ms grouped in four categories of human voice, macaque vocalizations, marmoset vocalizations, and complex non-vocal sounds (more details in *Bodin et al., 2021*).

## Main voice stimuli

The main stimulus set consisted of brief human voice sounds sampled from the Common Voice dataset (*Ardila et al., 2020*). Stimuli were organized into four main category levels: language (English, French, Spanish, Deutch, Polish, Portuguese, Russian, Chinese), gender (female/male), age (young/adult; young: teenagers and twenties; adult: thirties to sixties included), and identity (S1: 135 identities; S2: 142 identities; S3: 128 identities; ~44 samples per identity). Throughout the manuscript, the term 'gender' rather than 'sex' was utilized in reference to the demographic information obtained from the participants of the Common Voice dataset (*Ardila et al., 2020*), as it was the terminology employed in the survey ('male/female/other'). Stimulus sets were different for each participant and the number of stimuli per set also varied slightly (number of unique stimuli: Participant 1, N=6150; Participant 2, N=6148; Participant 3, N=5123). For each participant, six stimuli were selected randomly among the sounds having a higher energy (as measured by the amplitude envelope reaching an arbitrary threshold, likely corresponding to vowels) and were repeated extensively (60 times), to improve the performance of the brain decoding (*VanRullen and Reddy, 2019*; *Horikawa and Kamitani, 2017*; *Chang et al., 2019*); these will be called the 'repeated' stimuli hereafter, the remaining stimuli were presented twice. The third participant attended 5 BrainVoice sessions instead of 12, one BrainVoice session corresponding to 1030 stimuli (1024 unique stimuli and 6 'test' stimuli). Specifically, 5270 stimuli were presented to the third participant instead of ~12,000 for the two others. Among these 5270 stimuli, 5120 unique stimuli were presented once, as for the two other participants, 6 'test' stimuli were presented 25 times (150 trials). The dataset was fully balanced, with an equal number of samples for each combination of language, gender, age, and identity. Furthermore, to minimize potential adaptation effects, the stimuli were also balanced within each run according to these categories, and identity was balanced across sessions.

All stimuli of the main set were resampled at 24,414 Hz and adjusted in duration (250ms). For each stimulus, a fade-in and a fade-out were applied with a 15ms cosine ramp to their onset and offset and were normalized by dividing the root mean square amplitude. During fMRI sessions, stimulus presentations were controlled using custom Matlab scripts (Mathworks, Natick, MA, USA) interfaced with an RM1 Mobile Processor (Tucker-David Technologies, Alachua, USA). The auditory stimuli were delivered pseudo-randomly through MRI-compatible earphones (S14, SensiMetrics, USA) with no filtering and at a comfortable sound pressure level of around 85 dB SPL that allowed for clear and intelligible listening.

## Computational models

We used two computational models to learn representational space for voice signals: Linear Autoencoder (LIN) and Deep Variational Autoencoder (VAE; *Kingma and Welling, 2014*). Both are encoder-decoder models that are learned to reproduce at their output their input while going through a low-dimensional representation space usually called latent space (that we will call *voice latent space* since they are learned on voice data). The autoencoders were trained on a dataset of 182 K sounds from the Common Voice dataset (*Ardila et al., 2020*), balanced in gender, language, and identity to reduce the bias in the synthesis (*Gutierrez, 2021*). Both models operate on sounds which were represented as spectrograms that we describe below. These representations were tested in all the encoding/decoding and RSA analyses.

## Spectrograms

We used amplitude spectrograms as input for the models that we describe below. Short-term Fourier transforms of the waveform were computed using a sliding window of length 50ms with a hop size of 12.5ms (hence an overlap of 37.5ms) and applying a Hamming window of size 800 samples before computing the Fourier transform of each slice. Only the magnitude of the spectrogram was kept, and the phase of the complex representation was removed. At the end, a 250ms sound is represented by a 21×401 matrix with 21 time steps and 401 frequency bins.

We used a custom code based on *numpy.fft* package (*Harris et al., 2020*). The size and the overlap between the sliding windows of the spectrogram were chosen to conform with the uncertainty principle between time and frequency resolution. The main constraint was to find a trade-off between accurate phase reconstruction with the *Griffin and Jae, 1983* and a reasonable size of the spectrogram.

## Deep neural network

We designed a deep variational autoencoder (VAE; *Kingma and Welling, 2014*) of 15 layers with an intermediate hidden representation of 128 neurons that we refer to as the *voice latent space* (VLS). In an autoencoder model, the two sub-network components, the *Encoder* and the *Decoder*, are jointly learned on complementary tasks (*Figure 1a*). The Encoder network (noted $Enc$ hereafter; 7 layers) learns to map an input, $s$ (a spectrogram of a sound), onto a (128-dimensional) *voice latent space* representation ($z$; in blue in the middle of *Figure 1a*), while the Decoder (noted $Dec$ hereafter; 7 layers) aims at reconstructing the spectrogram $s$ from $z$. The learning objective of the full model is to make the output spectrogram $Dec\left(Enc\left(s\right)\right)$ as close as possible to the original one $s$. This reconstruction objective is defined as the L2 loss, $\| Dec(Enc(s)) - s \|^2$. The parameters of the Encoder and of the Decoder are jointly learned using gradient descent to optimize the average L2 loss computed on the training set $\sum_{s \in Training\ Set} \| Dec(Enc(s)) - s \|^2$. We trained this DNN on the Common Voice dataset (*Ardila et al., 2020*) according to VAE learning procedure (as explained in *Kingma and Welling, 2019*) until convergence (network architecture and particularities of the training procedure are provided in *Appendix 1—table 1*), using the PyTorch python package (*Paszke et al., 2019*). Before feeding the spectrograms to the autoencoder, we standardized each of the 401 frequency bands separately by centering all the data corresponding to each frequency band at every time step in all spectrograms, which involved removing their mean and dividing by their standard deviation. This separate standardization of frequency bands resulted in a smaller reconstruction error compared to standardizing across all the bands.

## Linear autoencoder

We trained a linear autoencoder on the same dataset (described above) to serve as a linear baseline. Both the *Encoder* and the *Decoder* networks consisted of a single fully-connected layer, without any activation functions. Similar to the VAE, the latent space obtained from the *Encoder* was a 128-dimensional vector. The parameters of both the *Encoder* and the *Decoder* were jointly learned using gradient descent to optimize the average L2 loss computed on the training set.

## Neuroimaging data acquisition

Participants were scanned using a 3 Tesla Prisma scanner (Siemens Healthcare, Erlangen, Germany) equipped with a 64-channel receiver head-coil. Their movements were monitored during the acquisition using the software FIRMM (*Dosenbach et al., 2017*). The whole-head high-resolution structural scan acquired during the first session was a T1-weighted multi-echo MPRAGE (MEMPRAGE) (TR = 2.5 s, TE = 2.53, 4.28, 6.07, 7.86ms, TI = 1000ms flip angle: 8°, matrix size = 208 × 300 × 320; resolution 0.8 × 0.8 × 0.8 mm³, acquisition time: 8min22s). Lower resolution scans acquired during all other sessions were T1-weighted MPRAGE scans (TR = 2.3 s, TE = 2.88ms, TI = 900ms, flip angle: 9°, matrix size = 192 × 240×256; resolution 1×1 × 1 mm³, sparse sampling with 2.8 times undersampling and compressed sensing reconstruction, acquisition time: 2 min 37). Functional imaging was performed using an EPI sequence (multiband factor = 5, TR = 462ms, TE = 31.2ms, flip angle: 45°, matrix size = 84 × 84×35, resolution 2.5×2.5 × 2.5 mm³). Functional slices were oriented parallel to the lateral sulci with a z-axis coverage of 87.5 mm, allowing them to fully cover both the TVAs (*Pernet et al., 2015*) and the FVAs (*Belin et al., 2018*). The physiological signals (heart rate and respiration) were measured with Siemens' external sensors.

## Pre-processing of neuroimaging data and general linear modeling

Tissue segmentation and brain extraction were performed on the structural scans using the default segmentation procedure of SPM 12 (*Ashburner, 2012*). The preprocessing of the BOLD responses involved correcting motion, registering inter-runs, detrending, and smoothing the data. Each functional volume was realigned to a reference volume taken from a steady period in the session that was spatially the closest to the average of all sessions. Transformation matrices between anatomical and functional data were computed using boundary-based registration (FSL; *Smith et al., 2004*). The data were respectively detrended and smoothed using the *nilearn* functions *clean_img* and *smooth_img* (kernel size of 3 mm; *Abraham et al., 2014*), resulting in the matrix $Y \in R^{S \times V}$, with $S$ the number of scans and $V$ the number of voxels.

A first general linear model (GLM) was fit to regress out the noise by predicting $Y$ from a 'denoised' design matrix composed of $R = 38$ regressors of nuisance (*Figure 2—figure supplement 2*). These regressors of nuisance, also called covariates of no interest, included: 6 head motion parameters (three variables for the translations, three variables for the rotations); 18 'RETROICOR' regressors (*Glover et al., 2000*) using the *TAPAS PhysIO* package (*Kasper et al., 2017*; with the hyperparameters set as specified in Snoek et al.) were computed from the physiological signals; 13 regressors modeling slow artifactual trends (sines and cosines, cut frequency of the high-pass filter = 0.01 Hz); and a confound-mean predictor. The design matrix was convolved with a hemodynamic response function (HRF) with a peak at 6 s and an undershoot at 16 s (*Glover, 1999*). We note the convolved design matrix as $X_d \in R^{S \times R}$. The 'denoised' GLM's parameters $\beta_d \in R^{R \times V}$ were optimized to minimize the amplitude of the residual $\beta_d = argmin_{\beta \in R^{R \times V}} \| Y - X_d\beta \|^2$. We used a lag-1 autoregressive model (ar(1)) to model the temporal structure of the noise (*Friston et al., 2002*). The *denoised* BOLD signal $Y_d$ was then obtained from the original one according to $Y_d = Y - (X_d\beta_d) \in R^{S \times V}$.

A second 'stimulus' GLM model was used to predict the denoised responses for each stimulus based on the denoised betas from the first GLM using a design matrix $X_s \in R^{S \times (N_s+1)}$ (which was convolved with a hemodynamic response function, HRF as above) and a parameters matrix $\beta_s \in R^{(N_s+1) \times V}$ where $N_S$ stands for the number of stimuli. The last row (resp. column) of $\beta_s$ (resp. $X_s$) stands for a silence condition. Again, $\beta_s$ was learned to minimize the residual $\beta_s = argmin_{\beta \in R^{(N_s+1) \times V}} \| Y_d - X_s\beta \|^2$. Once learned, each of the first $N_s$ line of $\beta_s$ was corrected by subtracting the $(N_s+1)^{th}$ line, yielding the contrast maps for stimuli $\widetilde{\beta}_s \in R^{N_s \times V}$. We note hereafter $\widetilde{\beta}_s [i, :] \in R^V$ the contrast map for a given stimulus; it is the $i^{th}$ line of $\widetilde{\beta}_s$.

A third 'identity' GLM was fit to predict the BOLD responses of each voice speaker identity, using a design matrix $\beta_i \in R^{(N_i+1) \times V}$ and a design matrix $X_i \in R^{S \times (N_i+1)}$ (which was again convolved with a hemodynamic response function, HRF) where $N_s$ stands for the number of unique speakers. Again, the last row/column in $\beta_i$ and $X_i$ stands for the silent condition. $\beta_i$ is learned to minimize the residual $\beta_i = argmin_{\beta \in R^{(N_i+1) \times V}} \| Y_d - X_i\beta \|^2$ (*Figure 2—figure supplement 1*). Again, the final speaker contrast maps were obtained by contrasting (i.e., subtracting) the regression coefficients in a row of $\beta_i$ with the silence condition (last row; *Figure 2—figure supplement 1*), yielding $\widetilde{\beta}_i \in R^{N_s \times V}$. Here, the $j^{th}$ row of $\widetilde{\beta}_i$, $\widetilde{\beta}_i [j, :] \in R^V$, represents the amplitude of the BOLD response of the contrast map for speaker $j$ (i.e. to all the stimuli from this speaker).

A fourth 'localizer' GLM model was used to predict the denoised BOLD responses of each sound category from the *Voice localizer stimuli* presented above. The procedure was similar to that described for the two previous GLM models. Once the GLM was learned, we contrasted the human voice category with the other sound categories in order to localize for each participant the posterior Temporal Voice Area (pTVA), medial Temporal Voice Area (mTVA), and anterior Temporal Voice Area (aTVA) in each hemisphere. The center of each TVA corresponded to the local maximum of the voice >non-voice t-map whose coordinates were the closest to the TVAs reported in *Pernet et al., 2015*. The analyses were carried on for each ROI of each hemisphere.

Additionally, we defined for each participant the primary auditory cortex (A1) as the maximum value of the probabilistic map (non-linearly registered to each participant functional space) of Heschl's gyri provided with the MNI152 template (*Penhune et al., 1996*), intersected with the sound vs silence contrast map.

## Identity-based and stimulus-based representations

We performed analyses either at the stimulus level, for example predicting the neural activity of a participant listening to a given *stimulus* ($\widetilde{\beta}_s$'s lines) from the *voice latent space* representation of this stimuli, or at the speaker identity level, for example predicting the average neural activity in response to stimuli of a given speaker *identity* ($\widetilde{\beta}_i$'s lines) from this speaker's *voice latent space* representation. The identity-based analyses were used for the characterization of the *voice latent space* (*Figure 1*), the brain encoding (*Figure 2*), and the representational similarity analysis (*Figure 3*), while the stimulus-based analyses were used for the brain decoding analyses (*Figures 4 and 5*).

We conducted stimulus-based analyses to examine the relationship between stimulus contrast maps in neural activity ($\widetilde{\beta}_s$) and the encodings of individual stimulus spectrograms computed by the encoder of an autoencoder model (either linear or deep variational autoencoder) on the computational side.

We will note $z_s^{lin} \in R^{N_s \times 128}$ the encodings of stimuli by the LIN model and $z_s^{vae} \in R^{N_s \times 128}$ the encodings of stimuli computed by the VAE model. The encoding of the k[th] stimuli by one of these models is the k[th] row of the corresponding matrix and it is noted as $z_s^{model}[k, :]$.

For identity-based analyses, we studied relationships between identity contrast maps in $\widetilde{\beta}_i$ on the neural activity side, and an encoding of speaker identity in the VLS implemented by an autoencoder model (LIN or VAE) on the computational side, for example we note $z_i^{vae}[j]$ the representation of speaker $j$ as computed by the *vae* model. We chose to define a speaker identity-based representation as the average of a set of sample-based representations for stimuli from this speaker, for example $z_i^{model}[j] = 1/\mid S_j \mid \sum_{k \in S_j} z_s^{model}[k, :]$ where $S_j$ stands for the set of stimuli by speaker $j$ and *model* stands for *vae* or *lin*. Averaging in the *voice latent space* is expected to be much more powerful and relevant than averaging in the input space spectrograms (*VanRullen and Reddy, 2019*).

## Characterization of the autoencoder latent space

We characterized the organization of the *voice latent space* (VLS) and of the features computed by the linear autoencoder (LIN) by measuring through classification experiments the presence of information about speaker's gender, age, and identity in the representations learned by these models.

We first computed the speaker's identity *voice latent space* representations for each of the 405 speakers in the main voice dataset (135+142 + 128 see *Stimuli* section) as explained above.

Next, we used these speakers' *voice latent space* representation to investigate if the gender, age, and identity were encoded in the VLS. To do so, we divided the data into separate train and test sets and learned classifiers to predict gender, age, or identity from the train set. The balanced (to avoid the small effects associated with unbalanced folds) accuracy of the classifiers was then evaluated on the test set. The higher the performance on the test set, the more confident we are that the information is encoded in the VLS. More specifically, for each task (gender, age, identity), we trained a Logistic Regression classifier (linear regularized logistic regression; L2 penalty, tol = 0.0001, fit_intercept = True, intercept_scaling = 1, max_iter = 100) using the scikit-learn python package (*Pedregosa et al., 2018*).

We performed five train-test splits with 80% of the data in the training and 20% in the test set. For each split, we used fivefold cross-validation on the training set to select the optimal value for the regularization hyperparameter C (searching between 10 values logarithmically spaced on the interval [−3,+3]). We then computed the generalization performance on the test set of the model trained on the full training set with the best hyperparameter value. Reported results were then averaged over fivefolds. Note that data were systematically normalized with a scaler fitted on the training set. We used a robust scaling strategy for these experiments (removing the median, then scaling to the quantile range; 25th quantile and 75th quantile), which occurs to be more relevant with a small training set.

To investigate how speaker identity information is encoded in the latent space representations of speakers' voices, we computed speaker identity *voice latent space* representations by averaging 20 stimulus-based representations, in order to obtain a limited amount of data per identity that could be distributed across training and test datasets.

We tested whether the mean of the distribution of accuracy scores obtained for fivefolds was significantly above chance level using the Wilcoxon signed-rank test.

## Brain encoding

We performed encoding experiments on identity-based representations for each of the three participants (*Figure 2*). For each participant, we learn a regularized linear regression that predicts a speaker-based neural activity, for example the $j^{th}$ speaker's contrast map $\widetilde{\beta}_i[j] \in R^V$, from this speaker's voice latent space representation, that we note $z_i^{model}[j] \in R^{128}$ (*Figure 2a*), where $i$ is the voxel index. We carried out these regression analyses for each ROI (A1, pTVA, mTVA, aTVA) in each hemisphere and participant, independently.

The regression model parameters $\widehat{W}_{encod} \in R^{128 \times V}$ were learned according to:

$$\widehat{W}_{encod} = argmin_{W_{encod} \in R^{128 \times V}} \sum_{j=1 \dots N_i} \left( z_i^{model}[j] \times W_{encod} - \tilde{\beta}_i[j] \right)^2 + \lambda \|W_{encod}\|^2$$

where $\lambda$ is a hyperparameter tuning the optimal tradeoff between the data fit and the penalization terms above. We used the ridge regression with built-in cross-validation as implemented as *RidgeCV* in the scikit-learn library (*Pedregosa et al., 2018*).

The statistical significance of each result was assessed using the following procedure: We repeated the following experiment 20 times with different random seeds. Each time, we performed five train-test splits with 80% of the data in the training and 20% in the test set. For each split, we used RidgeCV (relying on leave-one-out) on the training set to select the optimal value for the hyperparameter $\lambda$ (searching between 10 values logarithmically spaced on the interval $[10^{-1}; 10^8]$). Following standard practice in machine learning, we then computed the generalization performance on the test set of the model trained on the full training set with the best hyperparameter value. Reported results are then averaged over 20 experiments. Note that here again, with small training sets, data were systematically normalized in each experiment using robust scaling.

The evaluation relied on the 'brain score'-inspired procedure (*Schrimpf et al., 2018*) which evaluates the performance of the ridge regression with a Pearson's correlation score. Correlations between measured neural activities $\widetilde{\beta_i}$ and predicted ones $\widehat{z_i^{model}} * W_{encod}$ were computed for each voxel and averaged over repeated experiments (folds and seeds), yielding one correlation value for every voxel and for every setting. The significance of the results was assessed with one-sample t-tests for the Fisher z-transformed correlation scores (3 x participants x 2 hemispheres x V voxels). For each region of interest, the scores are reported across participants and hemispheres (*Figure 2b*). The exact same procedure was followed for the LIN modeling, and for the Wav2Vec and HuBERT models (*Figure 2—figure supplement 3*).

In order to determine which of the two feature spaces (VLS, LIN) and which of the two ROIs (A1, TVAs) yielded the best prediction of neural activity, we compared the means of distributions of correlations coefficients using a mixed ANOVA performed on the Fisher z-transformed coefficients (dependent variable: correlation; between factor: ROI; repeated measurements: Feature; between-participant identifier: voxel).

For each ROI, we then used t-tests to perform post-hoc contrasts for the VLS-LIN difference in brain encoding performance (comparison tests in *Figure 2b*; *Appendix 1—table 4*). We finally conducted two-sample t-tests between the brain encoding model's scores trained to predict A1 and those trained to predict TVAs to test the significance of the A1-TVAs difference (*Appendix 1—table 5*).

The statistical tests were all performed using the *pingouin* python package (*Vallat, 2018*).

## Representational similarity analysis

The RSA analyses were carried out using the package *rsatoolbox* (*Schütt et al., 2021*; https://github.com/rsagroup/rsatoolbox; *Schütt et al., 2025*). For each participant, region of interest and hemisphere, we computed the cerebral Representational Dissimilarity Matrix (RDM) using Pearson's correlation between the speaker identity-specific response patterns of the GLM estimates $\widetilde{\beta_i}$ (*Walther et al., 2016*; *Figure 3a*). The model RDMs were built using cosine distance (*Xing et al., 2015*; *Bhattacharya et al., 2017*; *Wang et al., 2018*), capturing speaker pairwise feature differences predicted by the computational models LIN and the VLS (*Figure 3a*). For greater comparability with the rest of the analyses described here, the GLM estimates and the computational models' features were first normalized using robust scaling. We computed the Spearman correlation coefficients between the brain RDMs for each ROI and the two model RDMs (*Figure 3b*). We assessed the significance of these brain-model correlation coefficients within a permutation-based 'maximum statistics' framework for multiple comparison correction (one-tailed inference; N permutations = 10,000 for each test; permutation of rows and columns of distance matrices, see *Giordano et al., 2023* and *Maris and Oostenveld, 2007*; see *Figure 3b*). We evaluated the VLS-LIN difference using a two-way repeated-measures ANOVA on the Fisher z-transformed Spearman correlation coefficients (dependent variable: correlation; within factors: ROI and Feature; participant identifier: participant hemisphere pair). The same permutation framework was also used to assess the significance of the difference between the RSA correlation for the VLS and LIN models.

## Brain decoding

Brain decoding was investigated at the stimulus level. The stimuli's voice latent space representations $z_s^{model} \in R^{N \times 128}$ and voice samples' contrast maps $\widetilde{\beta_s} \in R^{N \times V}$ were divided into train and test splits,

normalized across voice samples using robust scaling, and then fit into the training set. For every participant and each ROI, we trained a $L_2$-regularized linear model $W \in R^{V \times 128}$ to predict the voice samples' latent vectors from the voice samples' contrast maps (*Figure 4a*). The hyperparameter selection and optimization were made similarly as in the Brain encoding scheme. Training was performed on non-repeated stimuli (see Stimuli section). We then used the trained models to predict for each participant the six repeated stimuli that were the most presented. Waveforms were estimated starting from the reconstructed spectrograms using the Griffin-Lim phase reconstruction algorithm (*Griffin and Jae, 1983*).

We then used classifier analyses to assess the presence of voice information (gender, age, speaker identity) in the reconstructed latent representations (i.e. the latent representation predicted from the brain activity of a participant listening to a specific stimulus) (*Figure 5a, b and c*). To this purpose, we first trained linear classifiers to categorize the training voice stimuli (participant 1, N=6144; participant 2, N=6142; participant 3, N=5117; total, N=17403) by gender (2 classes), age (2 classes) or identity (17 classes) based on VLS coordinates. Secondly, we used the previously trained classifiers to predict the identity information based on the VLS derived from the brain responses of the 18 Test voice stimuli (3 participants x 6 stimuli). We first tested using one-sample t-tests that the mean of the distribution of accuracy scores obtained across random classifier initializations of classifiers (2 hemispheres x 20 seeds = 40) was significantly above chance level, for each category, ROI, and model. We then evaluated the difference in performance at preserving identity-related information depending on the model or ROI via two-way ANOVAs (dependent variable: accuracy; between factors: Feature and ROI). We performed post-hoc planned paired t-tests between each model pair to test the significance of the VLS-LIN difference. Two-sample t-tests were finally used to test the significance of the A1-TVAs difference.

## Listening tests

We recruited 13 participants through the online platform Prolific (https://www.prolific.com/) for a series of online behavioral experiments. All participants reported having normal hearing. The purpose of these experiments was to evaluate how well voice identity information and naturalness are preserved in fMRI-based reconstructed voice excerpts. In the main session, participants carried out 4 tasks, in the following order: 'speaker discrimination' (~120 min), 'perceived naturalness' (~30 min), 'gender categorization' (~30 min), 'age categorization' (~30 min). The experiment lasted 3 hr and 35 min, and each participant was paid £48. 12 participants performed the speaker discrimination task, and all participants performed the other tasks.

Prior to the main experiment session, participants carried out a short loudness-change detection task to ensure that they wore headphones and that they were attentive and properly set up for the main experiment (*Woods et al., 2017*). On each of the 12 trials, participants heard 3 tones and were asked to identify which tone was the least loud by clicking on one of 3 response buttons: 'First', 'Second', or 'Third'. Participants were admitted to the main experiment only if they achieved perfect performance in this task. We additionally refined the participant pool by excluding those who performed badly on the original stimuli, by retaining only the subjects whose performance was above the 25th percentile of accuracy (gender and age categorizations: as all participants performed well (*Figure 5d and e*, red dotted lines); speaker discrimination: 9/12 participants performed above the threshold of 64%).

The next three tasks were each carried out on the same set of 342 experimental stimuli, each presented on a different trial: 18 original stimuli, 36 stimuli reconstructed directly from the LIN and the VLS models, and 18 stimuli x 2 models x 4 regions of interest x 2 hemispheres = 288 brain-reconstructed stimuli.

In the 'perceived naturalness' task, participants were asked to rate how natural the voice sounded on a scale ranging from 'Not at all natural' to 'Highly natural' (i.e. similar to a real recording), and were instructed to use the full range of the scale.

During the 'gender categorization' task, participants categorized the gender by clicking on a 'Female' or 'Male' button.

Finally, in the 'age categorization' task, participants categorized the age of the speaker by clicking on a 'Younger' or 'Older' button.

In the 'speaker discrimination' task, participants carried out 684 trials (342 experimental stimuli x 2) with short breaks in between. On each trial, they were presented with 2 short sound stimuli, one after the other, and participants had to indicate whether they were from the same speaker or not. The speech material was selected randomly and was different between two stimuli.

To evaluate the performance of the participants, we first conducted one-sample t-tests to examine whether the mean accuracy score calculated from their responses was significantly higher than the chance level for each model and ROI. Next, we used two-way repeated-measures ANOVAs to assess the variation in participants' performances in identifying identity-related information (dependent variable: accuracy; between-participant factors: Feature and ROI). To determine the statistical significance of the VLS-LIN difference, we carried out post-hoc planned paired t-tests between each model pair. Finally, we employed two-sample t-tests to evaluate the statistical significance of the A1-TVAs difference.

## Acknowledgements

We thank Bruno Nazarian for the design of an MRI-compatible button. We thank Jean-Luc Anton and Kepkee Loh for useful discussions. This work was funded by the European Research Council (ERC) under the European Union's Horizon 2020 research and innovation program (grant agreement no. 788240). This work was performed in the Center IRM-INT@CERIMED (UMR 7289, AMU-CNRS), platform member of France Life Imaging network (grant ANR-11-INBS-0 0 06). This work, carried out within the Institute of Convergence ILCB (ANR-16-CONV-0002), has benefited from support from the French government (*France 2030*), managed by the French National Agency for Research (ANR) and the Excellence Initiative of Aix-Marseille University (A*MIDEX).

## Additional information

### Funding

| Funder | Grant reference number | Author |
| --- | --- | --- |
| European Research Council | https://cordis.europa.eu/project/id/788240 | Charly Lamothe<br>Régis Trapeau<br>Pascal Belin |
| France Life Imaging network | ANR-11-INBS-0 0 06 | Pascal Belin |
| Institute of Convergence ILCB | ANR-16-CONV-0002 | Pascal Belin |

The funders had no role in study design, data collection and interpretation, or the decision to submit the work for publication.

### Author contributions

Charly Lamothe, Conceptualization, Data curation, Formal analysis, Investigation, Visualization, Methodology, Writing – original draft, Writing – review and editing; Etienne Thoret, Conceptualization, Software, Formal analysis, Investigation, Methodology, Writing – review and editing; Régis Trapeau, Data curation, Software, Validation, Methodology, Writing – review and editing; Bruno L Giordano, Conceptualization, Software, Methodology, Writing – review and editing; Julien Sein, Resources, Data curation, Methodology, Writing – review and editing; Sylvain Takerkart, Stephane Ayache, Methodology, Writing – review and editing; Thierry Artieres, Conceptualization, Supervision, Validation, Methodology, Writing – original draft, Writing – review and editing; Pascal Belin, Conceptualization, Supervision, Validation, Methodology, Writing – original draft, Project administration, Writing – review and editing

### Author ORCIDs

Charly Lamothe https://orcid.org/0000-0001-9918-8258
Etienne Thoret https://orcid.org/0000-0002-8214-6278
Régis Trapeau https://orcid.org/0000-0003-1137-8669

Bruno L Giordano ⓘ https://orcid.org/0000-0001-7002-0486
Julien Sein ⓘ https://orcid.org/0000-0003-1767-5330
Sylvain Takerkart ⓘ https://orcid.org/0000-0001-8410-0962
Stephane Ayache ⓘ https://orcid.org/0000-0003-2982-7127
Thierry Artieres ⓘ https://orcid.org/0000-0003-3696-0321
Pascal Belin ⓘ https://orcid.org/0000-0002-7578-6365

## Ethics

This study was part of the project 'Réseaux du Langage' and was promoted by the National Center for Scientific Research (CNRS). It has been given favorable approval by the local ethics committee (Comité de Protection des Personnes Sud-Méditerranée) on the date of 13th February 2019. The National Agency for Medicines (ANSM) has been informed of this study, which is registered under the number 2017-A03614-49. Three native French human speakers were scanned (all females, 26-33 years old). Participants gave written informed consent and received a compensation of 40€ per hour for their participation. All were right-handed and no one had a hearing disorder or neurological disease. All participants had normal hearing thresholds below 15 dB HL, for octave frequencies between 0.125 and 8 kHz.

Reviewer #1 (Public review): https://doi.org/10.7554/eLife.98047.3.sa1
Reviewer #2 (Public review): https://doi.org/10.7554/eLife.98047.3.sa2
Reviewer #3 (Public review): https://doi.org/10.7554/eLife.98047.3.sa3
Author response https://doi.org/10.7554/eLife.98047.3.sa4

# Additional files

## Supplementary files

MDAR checklist

Audio file 1. Voice latent space interpolation. The audio files are two original voice samples (A, B); the synthesized voice samples from the spectrograms of the autoencoder reconstructions of the original two voice samples (A', B'); the synthesized voice samples from the spectrograms of the linearly interpolated *voice latent space* (VLS; A_to_B; *Figure 1c*).

> A.wav: Original voice sample of a female Chinese speaker
> A'.wav: Voice sample A.wav reconstructed by the autoencoder
> B.wav: Original voice sample of a male French speaker
> B'.wav: Voice sample B.wav reconstructed by the autoencoder
> A_to_B_lx.wav: Reconstructed voice samples from the linear interpolation between A and B VLS, where x is the interpolation step (0.2, 0.4, 0.6, 0.8).

Audio file 2. Brain-based voice reconstructions. The audio files are reconstructed voice samples from the fMRI responses in the speakers' temporal voice areas (TVAs). These sounds were used in the quantitative and subjective voice identity tests (*Figure 4*). The samples below are from a German and a Spanish speaker. The sounds are reconstructed for each speaker using 2 models: LIN and VLS.

> example1_orig.wav: Original voice sample of a male German speaker
> example1_VLS_RaTVA.wav: Reconstructed voice sample from fMRI activity in the right anterior temporal voice area (RaTVA) using the VLS model
> example1_LIN_RaTVA.wav: Reconstructed voice sample from fMRI activity in the right anterior temporal voice area (RaTVA) using the LIN model
> example1_SPEC_RaTVA.wav: Reconstructed voice sample from fMRI activity in the right anterior temporal voice area (RaTVA) using the audio spectrogram
> example2_orig.wav: Original voice sample of a male Spanish speaker
> example2_VLS_LmTVA.wav: Reconstructed voice sample from fMRI activity in the left middle voice area (LmTVA) using the VLS model
> example2_LIN_LmTVA.wav: Reconstructed voice sample from fMRI activity in the left middle voice area (LmTVA) using the LIN model

example2_SPEC_LmTVA.wav: Reconstructed voice sample from fMRI activity in the left middle voice area (LmTVA) using the audio spectrogram

## Data availability

The preprocessed data and codes are publicly available on Zenodo: https://doi.org/10.5281/zenodo. 15797933.

The following dataset was generated:

| Author(s) | Year | Dataset title | Dataset URL | Database and Identifier |
|---|---|---|---|---|
| Lamothe C, Thoret E, Trapeau R, Giordano BL, Sein J, Takerkart S, Ayache S, Artières T, Belin P | 2025 | Data and code for publication "Reconstructing Voice Identity from Noninvasive Auditory Cortex Recordings" | https://doi.org/ 10.5281/zenodo. 15797933 | Zenodo, 10.5281/ zenodo.15797933 |

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

# Appendix 1

**Appendix 1—table 1.** Architecture of the VAE network.

The architecture of the VAE consists of 15 layers with an intermediate hidden representation of 128 neurons that will stand for the VLS. The Encoder network (*Enc*; 7 layers) learns to map an input, *s* (a spectrogram of a sound), onto the (128-dimensional) VLS, while the Decoder (*Dec*; 7 layers) aims at reconstructing the spectrogram *s* from *z*. The learning objective of the full model is to make the output spectrogram $Dec(Enc(s))$ as close as possible to the original one *s*. The model was trained until convergence (approximately 1000 epochs). Hyperparameter search was conducted to determine the suitable learning rate. BN: batch normalization; FC: fully connected; ReLU: Rectified Linear Unit

| Name | Layer | # Filters | Filter size | Stride | Activation |
|---|---|---|---|---|---|
| Encoder | Conv2D+BN2 D | 64 | 6x3 | 2x2 | ReLU |
| | Conv2D+BN2 D | 128 | 6x2 | 2x2 | ReLU |
| | Conv2D+BN2 D | 256 | 6x2 | 2x1 | ReLU |
| | Conv2D+BN2 D | 512 | 6x2 | 2x1 | ReLU |
| | Conv2D | 7 | 6x2 | 1x1 | - |
| Bottleneck | FC | 256 | - | - | - |
| Decoder | ConvTrans2D+BN2 D | 512 | 27x3 | 1x1 | ReLU |
| | ConvTrans2D+BN2 D | 256 | 4x2 | 2x1 | ReLU |
| | ConvTrans2D+BN2 D | 128 | 4x2 | 2x1 | ReLU |
| | ConvTrans2D+BN2 D | 64 | 4x2 | 2x2 | ReLU |
| | ConvTrans2D | 1 | 4x2 | 2x2 | - |
| Batch size | 64 | | | | |
| Loss function | MSE +KLdivergence | | | | |
| Optimizer | Adam, learningrate =0.00005 | | | | |
| | betas = (0.5, 0.999) | | | | |

**Appendix 1—table 2.** Comparing the performance of the human listeners at discriminating speaker identity-related information by ROI.

This table reports the significance of the A1-TVAs difference in the speaker identity categorization and discrimination performance. Two-sample t-tests were conducted between the scores of human listeners at discriminating the speaker gender (2 classes), age (2 classes), and identity (17 classes) of the 18 Test Stimuli that were reconstructed from the VLS features with those from LIN features. s.e.m.=standard error of the mean. Here are reported the results of the statistical tests, t-value, degree of freedom (dof), p-value, degree of significance (unc. sig.), 95% confidence interval (CI95%), effect size (Cohen-d), Bayes Factor (BF10), and statistical power (power) for each speaker identity information and ROI.

| Subject | ROI | Correlation | s.e.m. | T | dof | p-val | unc. sig. | CI95% | cohen-d | BF10 | power |
|---|---|---|---|---|---|---|---|---|---|---|---|
| s1 | LA1 | 0.13 ± 0.15 | 0.03 | 4.78E+00 | 32 | 1.91E-05 | **** | [0.08, inf] | 8.30E-01 | 1.22E+03 | 1.00 |
| | RA1 | 0.21 ± 0.14 | 0.03 | 8.08E+00 | 32 | 1.57E-09 | **** | [0.16, inf] | 1.41E+00 | 7.74E+06 | 1.00 |
| | LmTVA | 0.32 ± 0.13 | 0.02 | 1.34E+01 | 32 | 5.25E-15 | **** | [0.28, inf] | 2.34E+00 | 1.27E+12 | 1.00 |
| | RmTVA | 0.16 ± 0.07 | 0.01 | 1.11E+01 | 26 | 1.21E-11 | **** | [0.13, inf] | 2.13E+00 | 7.53E+08 | 1.00 |
| | LpTVA | 0.07 ± 0.13 | 0.02 | 3.15E+00 | 32 | 1.76E-03 | ** | [0.03, inf] | 5.50E-01 | 2.14E+01 | 0.92 |
| | RpTVA | 0.04 ± 0.08 | 0.02 | 2.56E+00 | 31 | 7.82E-03 | ** | [0.01, inf] | 4.50E-01 | 6.05E+00 | 0.80 |
| | LaTVA | 0.27 ± 0.15 | 0.03 | 1.00E+01 | 30 | 2.30E-11 | **** | [0.23, inf] | 1.80E+00 | 4.20E+08 | 1.00 |
| | RaTVA | 0.11 ± 0.10 | 0.02 | 5.26E+00 | 25 | 9.42E-06 | **** | [0.07, inf] | 1.03E+00 | 2.42E+03 | 1.00 |

*Appendix 1—table 2 Continued*

| Subject | ROI | Correlation | s.e.m. | T | dof | p-val | unc. sig. | CI95% | cohen-d | BF10 | power |
|---------|-----|-------------|--------|---|-----|-------|-----------|-------|---------|------|-------|
| | A1 | 0.17 ± 0.15 | 0.02 | 8.80E+00 | 65 | 5.58E-13 | **** | [0.14, inf] | 1.08E+00 | 1.48E+10 | 1.00 |
| | mTVA | 0.25 ± 0.14 | 0.02 | 1.38E+01 | 59 | 1.71E-20 | **** | [0.22, inf] | 1.79E+00 | 2.85E+17 | 1.00 |
| | pTVA | 0.06 ± 0.11 | 0.01 | 4.02E+00 | 64 | 7.84E-05 | **** | [0.03, inf] | 5.00E-01 | 2.81E+02 | 0.99 |
| | aTVA | 0.20 ± 0.15 | 0.02 | 9.63E+00 | 56 | 8.92E-14 | **** | [0.16, inf] | 1.28E+00 | 8.76E+10 | 1.00 |
| | TVAs | 0.16 ± 0.16 | 0.01 | 1.39E+01 | 181 | 8.43E-31 | **** | [0.14, inf] | 1.03E+00 | 3.76E+27 | 1.00 |
| s2 | LA1 | 0.04 ± 0.11 | 0.02 | 2.16E+00 | 32 | 1.94E-02 | * | [0.01, inf] | 3.80E-01 | 2.83E+00 | 0.68 |
| | RA1 | -0.01 ± 0.11 | 0.02 | n/a | n/a | n/a | n/a | n/a | n/a | n/a | n/a |
| | LmTVA | -0.02 ± 0.09 | 0.02 | n/a | n/a | n/a | n/a | n/a | n/a | n/a | n/a |
| | RmTVA | 0.03 ± 0.11 | 0.02 | 1.17E+00 | 21 | 1.27E-01 | ns | [-0.01, inf] | 2.50E-01 | 8.20E-01 | 0.31 |
| | LpTVA | -0.01 ± 0.10 | 0.02 | n/a | n/a | n/a | n/a | n/a | n/a | n/a | n/a |
| | RpTVA | 0.04 ± 0.10 | 0.03 | 1.38E+00 | 16 | 9.37E-02 | ns | [-0.01, inf] | 3.30E-01 | 1.11E+00 | 0.37 |
| | LaTVA | -0.05 ± 0.12 | 0.02 | n/a | n/a | n/a | n/a | n/a | n/a | n/a | n/a |
| | RaTVA | 0.03 ± 0.12 | 0.03 | 1.18E+00 | 19 | 1.26E-01 | ns | [-0.01, inf] | 2.60E-01 | 8.56E-01 | 0.31 |
| | A1 | 0.02 ± 0.11 | 0.01 | 1.19E+00 | 65 | 1.19E-01 | ns | [-0.01, inf] | 1.50E-01 | 5.31E-01 | 0.32 |
| | mTVA | 0.00 ± 0.10 | 0.02 | 5.00E-02 | 46 | 4.81E-01 | ns | [-0.02, inf] | 1.00E-02 | 3.17E-01 | 0.06 |
| | pTVA | 0.01 ± 0.10 | 0.02 | 5.10E-01 | 45 | 3.07E-01 | ns | [-0.02, inf] | 7.00E-02 | 3.61E-01 | 0.13 |
| | aTVA | -0.02 ± 0.12 | 0.02 | n/a | n/a | n/a | n/a | n/a | n/a | n/a | n/a |
| | TVAs | -0.00 ± 0.11 | 0.01 | n/a | n/a | n/a | n/a | n/a | n/a | n/a | n/a |
| s3 | LA1 | 0.04 ± 0.08 | 0.01 | 2.89E+00 | 32 | 3.39E-03 | ** | [0.02, inf] | 5.00E-01 | 1.21E+01 | 0.88 |
| | RA1 | 0.03 ± 0.13 | 0.02 | 1.48E+00 | 32 | 7.39E-02 | ns | [-0.00, inf] | 2.60E-01 | 1.01E+00 | 0.42 |
| | LmTVA | 0.04 ± 0.09 | 0.02 | 2.43E+00 | 28 | 1.10E-02 | * | [0.01, inf] | 4.50E-01 | 4.72E+00 | 0.76 |
| | RmTVA | 0.07 ± 0.09 | 0.02 | 4.38E+00 | 28 | 7.48E-05 | **** | [0.04, inf] | 8.10E-01 | 3.64E+02 | 1.00 |
| | LpTVA | 0.03 ± 0.12 | 0.02 | 1.48E+00 | 28 | 7.45E-02 | ns | [-0.00, inf] | 2.80E-01 | 1.06E+00 | 0.42 |
| | RpTVA | 0.04 ± 0.08 | 0.01 | 2.83E+00 | 35 | 3.87E-03 | ** | [0.02, inf] | 4.70E-01 | 1.05E+01 | 0.87 |
| | LaTVA | 0.09 ± 0.13 | 0.03 | 3.15E+00 | 23 | 2.24E-03 | ** | [0.04, inf] | 6.40E-01 | 1.91E+01 | 0.92 |
| | RaTVA | 0.07 ± 0.12 | 0.03 | 2.41E+00 | 17 | 1.38E-02 | * | [0.02, inf] | 5.70E-01 | 4.61E+00 | 0.75 |
| | A1 | 0.04 ± 0.11 | 0.01 | 2.80E+00 | 65 | 3.38E-03 | ** | [0.01, inf] | 3.40E-01 | 9.50E+00 | 0.87 |
| | mTVA | 0.06 ± 0.09 | 0.01 | 4.76E+00 | 57 | 6.83E-06 | **** | [0.04, inf] | 6.30E-01 | 2.76E+03 | 1.00 |
| | pTVA | 0.04 ± 0.10 | 0.01 | 2.92E+00 | 64 | 2.40E-03 | ** | [0.02, inf] | 3.60E-01 | 1.29E+01 | 0.89 |
| | aTVA | 0.08 ± 0.13 | 0.02 | 4.00E+00 | 41 | 1.28E-04 | *** | [0.05, inf] | 6.20E-01 | 2.05E+02 | 0.99 |
| | TVAs | 0.06 ± 0.11 | 0.01 | 6.62E+00 | 164 | 2.46E-10 | **** | [0.04, inf] | 5.20E-01 | 3.49E+07 | 1.00 |
| all | LA1 | 0.07 ± 0.12 | 0.01 | 5.58E+00 | 98 | 1.05E-07 | **** | [0.05, inf] | 5.60E-01 | 1.21E+05 | 1.00 |
| | RA1 | 0.08 ± 0.16 | 0.02 | 4.82E+00 | 98 | 2.60E-06 | **** | [0.05, inf] | 4.80E-01 | 5.85E+03 | 1.00 |
| | LmTVA | 0.13 ± 0.19 | 0.02 | 6.37E+00 | 86 | 4.45E-09 | **** | [0.10, inf] | 6.80E-01 | 2.55E+06 | 1.00 |
| | RmTVA | 0.09 ± 0.10 | 0.01 | 7.55E+00 | 77 | 3.72E-11 | **** | [0.07, inf] | 8.50E-01 | 2.55E+08 | 1.00 |
| | LpTVA | 0.03 ± 0.12 | 0.01 | 2.66E+00 | 90 | 4.59E-03 | ** | [0.01, inf] | 2.80E-01 | 6.39E+00 | 0.84 |
| | RpTVA | 0.04 ± 0.09 | 0.01 | 4.01E+00 | 84 | 6.63E-05 | **** | [0.02, inf] | 4.30E-01 | 3.00E+02 | 0.99 |
| | LaTVA | 0.11 ± 0.19 | 0.02 | 5.07E+00 | 83 | 1.20E-06 | **** | [0.07, inf] | 5.50E-01 | 1.27E+04 | 1.00 |
| | RaTVA | 0.07 ± 0.12 | 0.01 | 5.01E+00 | 63 | 2.34E-06 | **** | [0.05, inf] | 6.30E-01 | 7.30E+03 | 1.00 |
| | A1 | 0.07 ± 0.14 | 0.01 | 7.25E+00 | 197 | 4.67E-12 | **** | [0.06, inf] | 5.20E-01 | 1.54E+09 | 1.00 |

*Appendix 1—table 2 Continued on next page*

*Appendix 1—table 2 Continued*

| Subject | ROI | Correlation | s.e.m. | T | dof | p-val | unc. sig. | CI95% | cohen-d | BF10 | power |
|---|---|---|---|---|---|---|---|---|---|---|---|
| | mTVA | 0.11 ± 0.16 | 0.01 | 9.12E+00 | 164 | 1.30E-16 | **** | [0.09, inf] | 7.10E-01 | 4.40E+13 | 1.00 |
| | pTVA | 0.04 ± 0.11 | 0.01 | 4.49E+00 | 175 | 6.53E-06 | **** | [0.02, inf] | 3.40E-01 | 2.00E+03 | 1.00 |
| | aTVA | 0.09 ± 0.17 | 0.01 | 6.81E+00 | 147 | 1.14E-10 | **** | [0.07, inf] | 5.60E-01 | 7.58E+07 | 1.00 |
| | TVAs | 0.08 ± 0.15 | 0.01 | 1.18E+01 | 488 | 9.58E-29 | **** | [0.07, inf] | 5.30E-01 | 2.93E+25 | 1.00 |

**Appendix 1—table 3.** Assessing the significance of brain encoding performance with LIN features.
This table reports the significance of the brain encoding performance with LIN features. We compared the distribution of Pearson's correlation coefficients to the chance level of 0.0 by conducting one-sample t-tests. Using a linear model, we calculated the correlation between the voxels in the speaker activity maps and the predicted voxels from the LIN features. s.e.m. = standard error of the mean. all = we combined the scores of all participants before computing the test. Here are reported the results of the statistical tests, t-value, degree of freedom (dof), p-value, degree of significance (unc. sig.), 95% confidence interval (CI95%), effect size (Cohen-d), Bayes Factor (BF10), and statistical power (power) for each participant and ROI.

| Subject | ROI | Correlation | s.e.m. | T | dof | p-val | unc. sig. | CI95% | cohen-d | BF10 | power |
|---|---|---|---|---|---|---|---|---|---|---|---|
| s1 | LA1 | 0.03 ± 0.11 | 0.02 | 1.46E+00 | 32 | 7.71E-02 | ns | [-0.00, inf] | 2.50E-01 | 9.75E-01 | 0.41 |
| | RA1 | 0.13 ± 0.09 | 0.02 | 8.06E+00 | 32 | 1.67E-09 | **** | [0.10, inf] | 1.40E+00 | 7.28E+06 | 1.00 |
| | LmTVA | 0.25 ± 0.16 | 0.03 | 8.95E+00 | 32 | 1.58E-10 | **** | [0.20, inf] | 1.56E+00 | 6.77E+07 | 1.00 |
| | RmTVA | 0.08 ± 0.09 | 0.02 | 4.89E+00 | 26 | 2.24E-05 | **** | [0.05, inf] | 9.40E-01 | 1.09E+03 | 1.00 |
| | LpTVA | -0.03 ± 0.12 | 0.02 | n/a | n/a | n/a | n/a | n/a | n/a | n/a | n/a |
| | RpTVA | -0.06 ± 0.11 | 0.02 | n/a | n/a | n/a | n/a | n/a | n/a | n/a | n/a |
| | LaTVA | 0.15 ± 0.16 | 0.03 | 5.34E+00 | 30 | 4.43E-06 | **** | [0.10, inf] | 9.60E-01 | 4.70E+03 | 1.00 |
| | RaTVA | 0.03 ± 0.11 | 0.02 | 1.55E+00 | 25 | 6.70E-02 | ns | [-0.00, inf] | 3.00E-01 | 1.19E+00 | 0.44 |
| | A1 | 0.08 ± 0.11 | 0.01 | 5.65E+00 | 65 | 1.93E-07 | **** | [0.06, inf] | 7.00E-01 | 7.57E+04 | 1.00 |
| | mTVA | 0.17 ± 0.15 | 0.02 | 8.69E+00 | 59 | 1.91E-12 | **** | [0.14, inf] | 1.12E+00 | 4.57E+09 | 1.00 |
| | pTVA | -0.04 ± 0.12 | 0.02 | n/a | n/a | n/a | n/a | n/a | n/a | n/a | n/a |
| | aTVA | 0.10 ± 0.15 | 0.02 | 4.94E+00 | 56 | 3.68E-06 | **** | [0.07, inf] | 6.50E-01 | 4.93E+03 | 1.00 |
| | TVAs | 0.07 ± 0.17 | 0.01 | 5.79E+00 | 181 | 1.52E-08 | **** | [0.05, inf] | 4.30E-01 | 6.37E+05 | 1.00 |
| s2 | LA1 | 0.04 ± 0.14 | 0.02 | 1.51E+00 | 32 | 7.01E-02 | ns | [-0.00, inf] | 2.60E-01 | 1.05E+00 | 0.43 |
| | RA1 | 0.01 ± 0.12 | 0.02 | 3.60E-01 | 32 | 3.59E-01 | ns | [-0.03, inf] | 6.00E-02 | 3.96E-01 | 0.10 |
| | LmTVA | 0.04 ± 0.07 | 0.01 | 3.07E+00 | 24 | 2.61E-03 | ** | [0.02, inf] | 6.10E-01 | 1.66E+01 | 0.91 |
| | RmTVA | 0.08 ± 0.10 | 0.02 | 3.95E+00 | 21 | 3.64E-04 | *** | [0.05, inf] | 8.40E-01 | 9.46E+01 | 0.99 |
| | LpTVA | -0.01 ± 0.10 | 0.02 | n/a | n/a | n/a | n/a | n/a | n/a | n/a | n/a |
| | RpTVA | 0.02 ± 0.13 | 0.03 | 7.30E-01 | 16 | 2.39E-01 | ns | [-0.03, inf] | 1.80E-01 | 6.29E-01 | 0.17 |
| | LaTVA | -0.01 ± 0.08 | 0.02 | n/a | n/a | n/a | n/a | n/a | n/a | n/a | n/a |
| | RaTVA | 0.02 ± 0.08 | 0.02 | 1.00E+00 | 19 | 1.64E-01 | ns | [-0.01, inf] | 2.20E-01 | 7.26E-01 | 0.25 |
| | A1 | 0.02 ± 0.13 | 0.02 | 1.38E+00 | 65 | 8.61E-02 | ns | [-0.00, inf] | 1.70E-01 | 6.66E-01 | 0.39 |
| | mTVA | 0.06 ± 0.09 | 0.01 | 4.92E+00 | 46 | 5.72E-06 | **** | [0.04, inf] | 7.20E-01 | 3.43E+03 | 1.00 |
| | pTVA | -0.00 ± 0.11 | 0.02 | n/a | n/a | n/a | n/a | n/a | n/a | n/a | n/a |
| | aTVA | 0.00 ± 0.08 | 0.01 | 4.10E-01 | 48 | 3.43E-01 | ns | [-0.02, inf] | 6.00E-02 | 3.36E-01 | 0.11 |
| | TVAs | 0.02 ± 0.10 | 0.01 | 2.65E+00 | 141 | 4.46E-03 | ** | [0.01, inf] | 2.20E-01 | 5.41E+00 | 0.84 |
| s3 | LA1 | 0.01 ± 0.09 | 0.02 | 3.50E-01 | 32 | 3.66E-01 | ns | [-0.02, inf] | 6.00E-02 | 3.94E-01 | 0.10 |

*Appendix 1—table 3 Continued on next page*

*Appendix 1—table 3 Continued*

| Subject | ROI | Correlation | s.e.m. | T | dof | p-val | unc. sig. | CI95% | cohen-d | BF10 | power |
|---------|-----|-------------|--------|---|-----|-------|-----------|-------|---------|------|-------|
| | RA1 | 0.03 ± 0.11 | 0.02 | 1.62E+00 | 32 | 5.78E-02 | ns | [-0.00, inf] | 2.80E-01 | 1.21E+00 | 0.48 |
| | LmTVA | 0.05 ± 0.14 | 0.03 | 2.03E+00 | 28 | 2.61E-02 | * | [0.01, inf] | 3.80E-01 | 2.34E+00 | 0.63 |
| | RmTVA | 0.09 ± 0.08 | 0.02 | 5.64E+00 | 28 | 2.41E-06 | **** | [0.06, inf] | 1.05E+00 | 8.29E+03 | 1.00 |
| | LpTVA | 0.00 ± 0.10 | 0.02 | 2.20E-01 | 28 | 4.12E-01 | ns | [-0.03, inf] | 4.00E-02 | 4.04E-01 | 0.08 |
| | RpTVA | 0.01 ± 0.11 | 0.02 | 4.50E-01 | 35 | 3.30E-01 | ns | [-0.02, inf] | 7.00E-02 | 3.93E-01 | 0.11 |
| | LaTVA | 0.04 ± 0.12 | 0.03 | 1.60E+00 | 23 | 6.16E-02 | ns | [-0.00, inf] | 3.30E-01 | 1.31E+00 | 0.46 |
| | RaTVA | 0.11 ± 0.12 | 0.03 | 3.65E+00 | 17 | 9.96E-04 | *** | [0.06, inf] | 8.60E-01 | 4.13E+01 | 0.97 |
| | A1 | 0.02 ± 0.10 | 0.01 | 1.49E+00 | 65 | 7.09E-02 | ns | [-0.00, inf] | 1.80E-01 | 7.69E-01 | 0.43 |
| | mTVA | 0.07 ± 0.11 | 0.02 | 4.68E+00 | 57 | 9.11E-06 | **** | [0.05, inf] | 6.10E-01 | 2.12E+03 | 1.00 |
| | pTVA | 0.01 ± 0.11 | 0.01 | 4.90E-01 | 64 | 3.14E-01 | ns | [-0.02, inf] | 6.00E-02 | 3.05E-01 | 0.12 |
| | aTVA | 0.07 ± 0.13 | 0.02 | 3.53E+00 | 41 | 5.14E-04 | *** | [0.04, inf] | 5.50E-01 | 5.87E+01 | 0.97 |
| | TVAs | 0.05 ± 0.12 | 0.01 | 4.87E+00 | 164 | 1.32E-06 | **** | [0.03, inf] | 3.80E-01 | 9.31E+03 | 1.00 |
| all | LA1 | 0.02 ± 0.11 | 0.01 | 2.04E+00 | 98 | 2.19E-02 | * | [0.00, inf] | 2.10E-01 | 1.62E+00 | 0.65 |
| | RA1 | 0.06 ± 0.12 | 0.01 | 4.67E+00 | 98 | 4.87E-06 | **** | [0.04, inf] | 4.70E-01 | 3.24E+03 | 1.00 |
| | LmTVA | 0.12 ± 0.16 | 0.02 | 7.09E+00 | 86 | 1.77E-10 | **** | [0.09, inf] | 7.60E-01 | 5.59E+07 | 1.00 |
| | RmTVA | 0.09 ± 0.09 | 0.01 | 8.47E+00 | 77 | 6.43E-13 | **** | [0.07, inf] | 9.60E-01 | 1.27E+10 | 1.00 |
| | LpTVA | -0.01 ± 0.11 | 0.01 | n/a | n/a | n/a | n/a | n/a | n/a | n/a | n/a |
| | RpTVA | -0.02 ± 0.12 | 0.01 | n/a | n/a | n/a | n/a | n/a | n/a | n/a | n/a |
| | LaTVA | 0.07 ± 0.14 | 0.02 | 4.23E+00 | 83 | 2.96E-05 | **** | [0.04, inf] | 4.60E-01 | 6.36E+02 | 0.99 |
| | RaTVA | 0.05 ± 0.11 | 0.01 | 3.57E+00 | 63 | 3.50E-04 | *** | [0.03, inf] | 4.50E-01 | 7.23E+01 | 0.97 |
| | A1 | 0.04 ± 0.12 | 0.01 | 4.76E+00 | 197 | 1.89E-06 | **** | [0.03, inf] | 3.40E-01 | 6.19E+03 | 1.00 |
| | mTVA | 0.11 ± 0.13 | 0.01 | 1.01E+01 | 164 | 2.66E-19 | **** | [0.09, inf] | 7.90E-01 | 1.88E+16 | 1.00 |
| | pTVA | -0.01 ± 0.12 | 0.01 | n/a | n/a | n/a | n/a | n/a | n/a | n/a | n/a |
| | aTVA | 0.06 ± 0.13 | 0.01 | 5.52E+00 | 147 | 7.61E-08 | **** | [0.04, inf] | 4.50E-01 | 1.46E+05 | 1.00 |
| | TVAs | 0.05 ± 0.14 | 0.01 | 7.88E+00 | 488 | 1.05E-14 | **** | [0.04, inf] | 3.60E-01 | 4.33E+11 | 1.00 |

**Appendix 1—table 4.** Assessing the significance of brain encoding performance with VLS features.
This table reports the significance of the brain encoding performance with VLS features. We compared the distribution of Pearson's correlation coefficients to the chance level of 0.0 by conducting one-sample t-tests. Using a linear model, we calculated the correlation between the voxels in the speaker activity maps and the predicted voxels from the VLS features. s.e.m.=standard error of the mean. all = we combined the scores of all participants before computing the test. Here are reported the results of the statistical tests, t-value, degree of freedom (dof), p-value, degree of significance (unc. sig.), 95% confidence interval (CI95%), effect size (Cohen-d), Bayes Factor (BF10), and statistical power (power) for each participant and ROI.

| Subject | ROI | Correlation VLS | Correlation LIN | s.e.m. VLS | s.e.m. LIN | T VLS vs LIN | dof | p-val | unc. sig. | CI95% | cohen-d | BF10 | power |
|---------|-----|-----------------|-----------------|------------|------------|--------------|-----|-------|-----------|-------|---------|------|-------|
| s1 | LA1 | 0.03 ± 0.11 | 0.13 ± 0.15 | 0.02 | 0.03 | -4.43E+00 | 32 | 1.03E-04 | *** | [-0.14, -0.05] | 7.30E-01 | 2.47E+02 | 0.98 |
| | RA1 | 0.13 ± 0.09 | 0.21 ± 0.14 | 0.02 | 0.03 | -3.75E+00 | 32 | 7.07E-04 | *** | [-0.11, -0.03] | 6.00E-01 | 4.39E+01 | 0.92 |
| | LmTVA | 0.25 ± 0.16 | 0.32 ± 0.13 | 0.03 | 0.02 | -3.90E+00 | 32 | 4.61E-04 | *** | [-0.11, -0.03] | 4.80E-01 | 6.43E+01 | 0.76 |
| | RmTVA | 0.08 ± 0.09 | 0.16 ± 0.07 | 0.02 | 0.01 | -5.48E+00 | 26 | 9.54E-06 | **** | [-0.10, -0.05] | 9.20E-01 | 2.24E+03 | 1.00 |
| | LpTVA | -0.03 ± 0.12 | 0.07 ± 0.13 | 0.02 | 0.02 | -6.49E+00 | 32 | 2.68E-07 | **** | [-0.13, -0.07] | 7.60E-01 | 5.95E+04 | 0.99 |
| | RpTVA | -0.06 ± 0.11 | 0.04 ± 0.08 | 0.02 | 0.02 | -5.09E+00 | 31 | 1.67E-05 | **** | [-0.14, -0.06] | 1.01E+00 | 1.31E+03 | 1.00 |

*Appendix 1—table 4 Continued on next page*

Appendix 1—table 4 Continued

| Subject | ROI | Correlation VLS | Correlation LIN | s.e.m. VLS | s.e.m. LIN | T VLS vs LIN | dof | p-val | unc. sig. | CI95% | cohen-d | BF10 | power |
|---|---|---|---|---|---|---|---|---|---|---|---|---|---|
| | LaTVA | 0.15 ± 0.16 | 0.27 ± 0.15 | 0.03 | 0.03 | -7.34E+00 | 30 | 3.55E-08 | **** | [-0.15, -0.09] | 7.70E-01 | 3.95E+05 | 0.99 |
| | RaTVA | 0.03 ± 0.11 | 0.11 ± 0.10 | 0.02 | 0.02 | -4.24E+00 | 25 | 2.65E-04 | *** | [-0.11, -0.04] | 7.10E-01 | 1.11E+02 | 0.93 |
| | A1 | 0.08 ± 0.11 | 0.17 ± 0.15 | 0.01 | 0.02 | -5.81E+00 | 65 | 2.02E-07 | **** | [-0.12, -0.06] | 6.30E-01 | 6.96E+04 | 1.00 |
| | mTVA | 0.17 ± 0.15 | 0.25 ± 0.14 | 0.02 | 0.02 | -6.24E+00 | 59 | 5.16E-08 | **** | [-0.10, -0.05] | 5.00E-01 | 2.58E+05 | 0.97 |
| | pTVA | -0.04 ± 0.12 | 0.06 ± 0.11 | 0.02 | 0.01 | -8.06E+00 | 64 | 2.58E-11 | **** | [-0.12, -0.08] | 8.60E-01 | 3.62E+08 | 1.00 |
| | aTVA | 0.10 ± 0.15 | 0.20 ± 0.15 | 0.02 | 0.02 | -8.11E+00 | 56 | 5.09E-11 | **** | [-0.13, -0.08] | 6.60E-01 | 1.91E+08 | 1.00 |
| | TVAs | 0.07 ± 0.17 | 0.16 ± 0.16 | 0.01 | 0.01 | -1.29E+01 | 181 | 1.85E-27 | **** | [-0.11, -0.08] | 5.60E-01 | 1.89E+24 | 1.00 |
| s2 | LA1 | 0.04 ± 0.14 | 0.04 ± 0.11 | 0.02 | 0.02 | -2.70E-01 | 32 | 7.93E-01 | ns | [-0.03, 0.02] | 3.00E-02 | 1.92E-01 | 0.05 |
| | RA1 | 0.01 ± 0.12 | -0.01 ± 0.11 | 0.02 | 0.02 | 6.20E-01 | 32 | 5.38E-01 | ns | [-0.03, 0.06] | 1.30E-01 | 2.23E-01 | 0.11 |
| | LmTVA | 0.04 ± 0.07 | -0.02 ± 0.09 | 0.01 | 0.02 | 3.52E+00 | 24 | 1.77E-03 | ** | [0.03, 0.11] | 8.10E-01 | 2.11E+01 | 0.97 |
| | RmTVA | 0.08 ± 0.10 | 0.03 ± 0.11 | 0.02 | 0.02 | 3.74E+00 | 21 | 1.22E-03 | ** | [0.02, 0.09] | 5.20E-01 | 3.01E+01 | 0.65 |
| | LpTVA | -0.01 ± 0.10 | -0.01 ± 0.10 | 0.02 | 0.02 | -4.20E-01 | 28 | 6.78E-01 | ns | [-0.04, 0.02] | 6.00E-02 | 2.14E-01 | 0.06 |
| | RpTVA | 0.02 ± 0.13 | 0.04 ± 0.10 | 0.03 | 0.03 | -4.10E-01 | 16 | 6.88E-01 | ns | [-0.07, 0.05] | 1.00E-01 | 2.68E-01 | 0.07 |
| | LaTVA | -0.01 ± 0.08 | -0.05 ± 0.12 | 0.02 | 0.02 | 2.78E+00 | 28 | 9.51E-03 | ** | [0.01, 0.08] | 4.70E-01 | 4.75E+00 | 0.68 |
| | RaTVA | 0.02 ± 0.08 | 0.03 ± 0.12 | 0.02 | 0.03 | -4.30E-01 | 19 | 6.69E-01 | ns | [-0.07, 0.05] | 1.20E-01 | 2.53E-01 | 0.08 |
| | A1 | 0.02 ± 0.13 | 0.02 ± 0.11 | 0.02 | 0.01 | 4.00E-01 | 65 | 6.87E-01 | ns | [-0.02, 0.03] | 5.00E-02 | 1.46E-01 | 0.07 |
| | mTVA | 0.06 ± 0.09 | 0.00 ± 0.10 | 0.01 | 0.02 | 5.06E+00 | 46 | 7.24E-06 | **** | [0.04, 0.09] | 6.40E-01 | 2.62E+03 | 0.99 |
| | pTVA | -0.00 ± 0.11 | 0.01 ± 0.10 | 0.02 | 0.02 | -5.90E-01 | 45 | 5.57E-01 | ns | [-0.04, 0.02] | 8.00E-02 | 1.89E-01 | 0.08 |
| | aTVA | 0.00 ± 0.08 | -0.02 ± 0.12 | 0.01 | 0.02 | 1.50E+00 | 48 | 1.40E-01 | ns | [-0.01, 0.06] | 2.20E-01 | 4.43E-01 | 0.33 |
| | TVAs | 0.02 ± 0.10 | -0.00 ± 0.11 | 0.01 | 0.01 | 3.06E+00 | 141 | 2.64E-03 | ** | [0.01, 0.04] | 2.40E-01 | 8.00E+00 | 0.83 |
| s3 | LA1 | 0.01 ± 0.09 | 0.04 ± 0.08 | 0.02 | 0.01 | -2.32E+00 | 32 | 2.68E-02 | * | [-0.07, -0.00] | 4.20E-01 | 1.91E+00 | 0.64 |
| | RA1 | 0.03 ± 0.11 | 0.03 ± 0.13 | 0.02 | 0.02 | -1.00E-01 | 32 | 9.17E-01 | ns | [-0.04, 0.03] | 1.00E-02 | 1.87E-01 | 0.05 |
| | LmTVA | 0.05 ± 0.14 | 0.04 ± 0.09 | 0.03 | 0.02 | 7.20E-01 | 28 | 4.79E-01 | ns | [-0.02, 0.04] | 8.00E-02 | 2.50E-01 | 0.07 |
| | RmTVA | 0.09 ± 0.08 | 0.07 ± 0.09 | 0.02 | 0.02 | 9.30E-01 | 28 | 3.59E-01 | ns | [-0.02, 0.05] | 1.80E-01 | 2.94E-01 | 0.16 |
| | LpTVA | 0.00 ± 0.10 | 0.03 ± 0.12 | 0.02 | 0.02 | -1.82E+00 | 28 | 7.91E-02 | ns | [-0.06, 0.00] | 2.50E-01 | 8.47E-01 | 0.26 |
| | RpTVA | 0.01 ± 0.11 | 0.04 ± 0.08 | 0.02 | 0.01 | -2.26E+00 | 35 | 3.03E-02 | * | [-0.06, -0.00] | 3.10E-01 | 1.67E+00 | 0.44 |
| | LaTVA | 0.04 ± 0.12 | 0.09 ± 0.13 | 0.03 | 0.03 | -3.71E+00 | 23 | 1.15E-03 | ** | [-0.07, -0.02] | 3.70E-01 | 3.10E+01 | 0.40 |
| | RaTVA | 0.11 ± 0.12 | 0.07 ± 0.12 | 0.03 | 0.03 | 2.79E+00 | 17 | 1.25E-02 | * | [0.01, 0.07] | 3.00E-01 | 4.41E+00 | 0.23 |
| | A1 | 0.02 ± 0.10 | 0.04 ± 0.11 | 0.01 | 0.01 | -1.60E+00 | 65 | 1.14E-01 | ns | [-0.04, 0.00] | 1.80E-01 | 4.55E-01 | 0.29 |
| | mTVA | 0.07 ± 0.11 | 0.06 ± 0.09 | 0.02 | 0.01 | 1.19E+00 | 57 | 2.40E-01 | ns | [-0.01, 0.03] | 1.20E-01 | 2.79E-01 | 0.15 |
| | pTVA | 0.01 ± 0.11 | 0.04 ± 0.10 | 0.01 | 0.01 | -2.92E+00 | 64 | 4.88E-03 | ** | [-0.05, -0.01] | 2.80E-01 | 6.36E+00 | 0.61 |
| | aTVA | 0.07 ± 0.13 | 0.08 ± 0.13 | 0.02 | 0.02 | -9.50E-01 | 41 | 3.49E-01 | ns | [-0.03, 0.01] | 8.00E-02 | 2.54E-01 | 0.08 |
| | TVAs | 0.05 ± 0.12 | 0.06 ± 0.11 | 0.01 | 0.01 | -1.54E+00 | 164 | 1.25E-01 | ns | [-0.02, 0.00] | 9.00E-02 | 2.77E-01 | 0.20 |
| all | LA1 | 0.02 ± 0.11 | 0.07 ± 0.12 | 0.01 | 0.01 | -4.25E+00 | 98 | 4.92E-05 | **** | [-0.07, -0.02] | 3.80E-01 | 3.57E+02 | 0.97 |
| | RA1 | 0.06 ± 0.12 | 0.08 ± 0.16 | 0.01 | 0.02 | -1.64E+00 | 98 | 1.04E-01 | ns | [-0.04, 0.00] | 1.40E-01 | 4.06E-01 | 0.29 |
| | LmTVA | 0.12 ± 0.16 | 0.13 ± 0.19 | 0.02 | 0.02 | -4.10E-01 | 86 | 6.80E-01 | ns | [-0.03, 0.02] | 3.00E-02 | 1.29E-01 | 0.06 |
| | RmTVA | 0.09 ± 0.09 | 0.09 ± 0.10 | 0.01 | 0.01 | -4.00E-01 | 77 | 6.87E-01 | ns | [-0.03, 0.02] | 4.00E-02 | 1.35E-01 | 0.07 |
| | LpTVA | -0.01 ± 0.11 | 0.03 ± 0.12 | 0.01 | 0.01 | -4.81E+00 | 90 | 5.96E-06 | **** | [-0.07, -0.03] | 4.00E-01 | 2.64E+03 | 0.96 |
| | RpTVA | -0.02 ± 0.12 | 0.04 ± 0.09 | 0.01 | 0.01 | -4.60E+00 | 84 | 1.51E-05 | **** | [-0.08, -0.03] | 5.10E-01 | 1.13E+03 | 1.00 |
| | LaTVA | 0.07 ± 0.14 | 0.11 ± 0.19 | 0.02 | 0.02 | -3.42E+00 | 83 | 9.61E-04 | *** | [-0.07, -0.02] | 2.40E-01 | 2.46E+01 | 0.59 |
| | RaTVA | 0.05 ± 0.11 | 0.07 ± 0.12 | 0.01 | 0.01 | -1.81E+00 | 63 | 7.58E-02 | ns | [-0.05, 0.00] | 2.10E-01 | 6.31E-01 | 0.37 |
| | A1 | 0.04 ± 0.12 | 0.07 ± 0.14 | 0.01 | 0.01 | -4.02E+00 | 197 | 8.19E-05 | **** | [-0.05, -0.02] | 2.50E-01 | 1.70E+02 | 0.94 |

Appendix 1—table 4 Continued on next page

*Appendix 1—table 4 Continued*

| Subject | ROI | Correlation VLS | Correlation LIN | s.e.m. VLS | s.e.m. LIN | T VLS vs LIN | dof | p-val | unc. sig. | CI95% | cohen-d | BF10 | power |
|---|---|---|---|---|---|---|---|---|---|---|---|---|---|
| | mTVA | 0.11 ± 0.13 | 0.11 ± 0.16 | 0.01 | 0.01 | -5.80E-01 | 164 | 5.64E-01 | ns | [-0.02, 0.01] | 3.00E-02 | 1.02E-01 | 0.07 |
| | pTVA | -0.01 ± 0.12 | 0.04 ± 0.11 | 0.01 | 0.01 | -6.65E+00 | 175 | 3.68E-10 | **** | [-0.06, -0.04] | 4.50E-01 | 2.27E+07 | 1.00 |
| | aTVA | 0.06 ± 0.13 | 0.09 ± 0.17 | 0.01 | 0.01 | -3.79E+00 | 147 | 2.23E-04 | *** | [-0.05, -0.02] | 2.30E-01 | 7.55E+01 | 0.78 |
| | TVAs | 0.05 ± 0.14 | 0.08 ± 0.15 | 0.01 | 0.01 | -6.28E+00 | 488 | 7.44E-10 | **** | [-0.04, -0.02] | 2.10E-01 | 7.90E+06 | 1.00 |

**Appendix 1—table 5.** Comparing the performance of brain encoding models.

This table reports the significance of the VLS-LIN difference in the brain encoding performance. We conducted paired t-tests between the brain encoding model's scores trained with the VLS features to predict the speaker activity maps' voxels and those trained with the LIN features. s.e.m.=standard error of the mean. all = we combined the scores of all participants before computing the test. Here are reported the results of the statistical tests, t-value, degree of freedom (dof), p-value, degree of significance (unc. sig.), 95% confidence interval (CI95%), effect size (Cohen-d), Bayes Factor (BF10), and statistical power (power) for each participant and ROI.

| Subject | Model | ROI | Correlation ROI | Correlation A1 | s.e.m. ROI | s.e.m. A1 | T ROI vs A1 | dof | p-val | unc. sig. | CI95% | cohen-d | BF10 | power |
|---|---|---|---|---|---|---|---|---|---|---|---|---|---|---|
| s1 | LIN | mTVA | 0.25±0.14 | 0.17±0.15 | 0.02 | 0.02 | 3.070000 | 124 | 2.62E-03 | ** | [0.03, 0.13] | 5.50E-01 | 1.25E+01 | 0.86 |
| | | pTVA | 0.06±0.11 | 0.17±0.15 | 0.01 | 0.02 | –4.710000 | 129 | 6.34E-06 | **** | [-0.16,–0.06] | 8.20E-01 | 2.60E+03 | 1.00 |
| | | aTVA | 0.20±0.15 | 0.17±0.15 | 0.02 | 0.02 | 1.150000 | 121 | 2.53E-01 | ns | [–0.02, 0.09] | 2.10E-01 | 3.50E-01 | 0.21 |
| | | TVAs | 0.16±0.16 | 0.17±0.15 | 0.01 | 0.02 | –0.130000 | 246 | 8.93E-01 | ns | [–0.05, 0.04] | 2.00E-02 | 1.57E-01 | 0.05 |
| | VLS | mTVA | 0.17±0.15 | 0.08±0.11 | 0.02 | 0.01 | 3.860000 | 124 | 1.81E-04 | *** | [0.05, 0.14] | 6.90E-01 | 1.28E+02 | 0.97 |
| | | pTVA | –0.04±0.12 | 0.08±0.11 | 0.02 | 0.01 | –6.020000 | 129 | 1.68E-08 | **** | [-0.17,–0.08] | 1.05E+00 | 6.23E+05 | 1.00 |
| | | aTVA | 0.10±0.15 | 0.08±0.11 | 0.02 | 0.01 | 0.750000 | 121 | 4.53E-01 | ns | [–0.03, 0.07] | 1.40E-01 | 2.49E-01 | 0.12 |
| | | TVAs | 0.07±0.17 | 0.08±0.11 | 0.01 | 0.01 | –0.350000 | 246 | 7.25E-01 | ns | [–0.05, 0.04] | 5.00E-02 | 1.65E-01 | 0.06 |
| s2 | LIN | mTVA | 0.00±0.10 | 0.02±0.11 | 0.02 | 0.01 | –0.760000 | 111 | 4.48E-01 | ns | [–0.06, 0.03] | 1.50E-01 | 2.62E-01 | 0.12 |
| | | pTVA | 0.01±0.10 | 0.02±0.11 | 0.02 | 0.01 | –0.430000 | 110 | 6.70E-01 | ns | [–0.05, 0.03] | 8.00E-02 | 2.21E-01 | 0.07 |
| | | aTVA | –0.02±0.12 | 0.02±0.11 | 0.02 | 0.01 | –1.580000 | 113 | 1.16E-01 | ns | [–0.08, 0.01] | 3.00E-01 | 6.15E-01 | 0.35 |
| | | TVAs | –0.00±0.11 | 0.02±0.11 | 0.01 | 0.01 | –1.220000 | 206 | 2.22E-01 | ns | [–0.05, 0.01] | 1.80E-01 | 3.24E-01 | 0.23 |
| | VLS | mTVA | 0.06±0.09 | 0.02±0.13 | 0.01 | 0.02 | 1.810000 | 111 | 7.29E-02 | ns | [–0.00, 0.08] | 3.50E-01 | 8.70E-01 | 0.43 |
| | | pTVA | –0.00±0.11 | 0.02±0.13 | 0.02 | 0.02 | –0.960000 | 110 | 3.41E-01 | ns | [–0.07, 0.02] | 1.80E-01 | 3.06E-01 | 0.16 |
| | | aTVA | 0.00±0.08 | 0.02±0.13 | 0.01 | 0.02 | –0.810000 | 113 | 4.20E-01 | ns | [–0.06, 0.03] | 1.50E-01 | 2.69E-01 | 0.13 |
| | | TVAs | 0.02±0.10 | 0.02±0.13 | 0.01 | 0.02 | –0.020000 | 206 | 9.87E-01 | ns | [–0.03, 0.03] | 0.00E+00 | 1.62E-01 | 0.05 |
| s3 | LIN | mTVA | 0.06±0.09 | 0.04±0.11 | 0.01 | 0.01 | 1.170000 | 122 | 2.43E-01 | ns | [–0.01, 0.06] | 2.10E-01 | 3.57E-01 | 0.21 |
| | | pTVA | 0.04±0.10 | 0.04±0.11 | 0.01 | 0.01 | –0.040000 | 129 | 9.71E-01 | ns | [–0.04, 0.03] | 1.00E-02 | 1.87E-01 | 0.05 |
| | | aTVA | 0.08±0.13 | 0.04±0.11 | 0.02 | 0.01 | 1.940000 | 106 | 5.55E-02 | ns | [–0.00, 0.09] | 3.80E-01 | 1.09E+00 | 0.48 |
| | | TVAs | 0.06±0.11 | 0.04±0.11 | 0.01 | 0.01 | 1.190000 | 229 | 2.35E-01 | ns | [–0.01, 0.05] | 1.70E-01 | 3.06E-01 | 0.22 |
| | VLS | mTVA | 0.07±0.11 | 0.02±0.10 | 0.02 | 0.01 | 2.700000 | 122 | 7.97E-03 | ** | [0.01, 0.09] | 4.90E-01 | 4.89E+00 | 0.76 |
| | | pTVA | 0.01±0.11 | 0.02±0.10 | 0.01 | 0.01 | –0.650000 | 129 | 5.16E-01 | ns | [–0.05, 0.02] | 1.10E-01 | 2.27E-01 | 0.10 |
| | | aTVA | 0.07±0.13 | 0.02±0.10 | 0.02 | 0.01 | 2.340000 | 106 | 2.11E-02 | * | [0.01, 0.10] | 4.60E-01 | 2.32E+00 | 0.64 |
| | | TVAs | 0.05±0.12 | 0.02±0.10 | 0.01 | 0.01 | 1.610000 | 229 | 1.08E-01 | ns | [–0.01, 0.06] | 2.30E-01 | 5.30E-01 | 0.36 |
| all | LIN | mTVA | 0.11±0.16 | 0.07±0.14 | 0.01 | 0.01 | 2.360000 | 361 | 1.86E-02 | * | [0.01, 0.07] | 2.50E-01 | 1.69E+00 | 0.65 |
| | | pTVA | 0.04±0.11 | 0.07±0.14 | 0.01 | 0.01 | –2.850000 | 372 | 4.57E-03 | ** | [-0.06,–0.01] | 3.00E-01 | 5.59E+00 | 0.81 |
| | | aTVA | 0.09±0.17 | 0.07±0.14 | 0.01 | 0.01 | 1.200000 | 344 | 2.29E-01 | ns | [–0.01, 0.05] | 1.30E-01 | 2.40E-01 | 0.22 |
| | | TVAs | 0.08±0.15 | 0.07±0.14 | 0.01 | 0.01 | 0.410000 | 685 | 6.79E-01 | ns | [–0.02, 0.03] | 3.00E-02 | 1.02E-01 | 0.07 |
| | VLS | mTVA | 0.11±0.13 | 0.04±0.12 | 0.01 | 0.01 | 4.910000 | 361 | 1.40E-06 | **** | [0.04, 0.09] | 5.20E-01 | 9.29E+03 | 1.00 |
| | | pTVA | –0.01±0.12 | 0.04±0.12 | 0.01 | 0.01 | –4.450000 | 372 | 1.13E-05 | **** | [-0.08,–0.03] | 4.60E-01 | 1.31E+03 | 0.99 |

*Appendix 1—table 5 Continued on next page*

*Appendix 1—table 5 Continued*

| Subject | Model | ROI | Correlation ROI | Correlation A1 | s.e.m. ROI | s.e.m. A1 | T ROI vs A1 | dof | p-val | unc. sig. | CI95% | cohen-d | BF10 | power |
|---------|-------|-----|-----------------|----------------|------------|-----------|-------------|-----|-------|-----------|-------|---------|------|-------|
| | | aTVA | 0.06±0.13 | 0.04±0.12 | 0.01 | 0.01 | 1.410000 | 344 | 1.58E-01 | ns | [−0.01, 0.05] | 1.50E-01 | 3.13E-01 | 0.29 |
| | | TVAs | 0.05±0.14 | 0.04±0.12 | 0.01 | 0.01 | 0.750000 | 685 | 4.56E-01 | ns | [−0.01, 0.03] | 6.00E-02 | 1.23E-01 | 0.12 |

**Appendix 1—table 6.** Comparing the performance of brain encoding ROIs.

This table reports the significance of the A1-TVAs difference in the brain encoding performance. We conducted two-sample t-tests between the brain encoding model's scores trained to predict A1 and those trained to predict temporal voice areas. s.e.m.=standard error of the mean. all = we combined the scores of all participants before computing the test. Here are reported the results of the statistical tests, t-value, degree of freedom (dof), p-value, degree of significance (unc. sig.), 95% confidence interval (CI95%), effect size (Cohen-d), Bayes Factor (BF10), and statistical power (power) for each participant and model.

| Subject | Model | ROI | Correlation | p-unc | p-corr | corr. sig. |
|---------|-------|-----|-------------|-------|--------|------------|
| s1 | LIN | LA1 | 0.07 | 1.39E-02 | 1.79E-01 | ns |
| | | RA1 | 0.08 | 4.20E-03 | 1.04E-01 | ns |
| | | LmTVA | 0.08 | 1.92E-02 | 3.80E-01 | ns |
| | | RmTVA | 0.06 | 7.14E-02 | 5.51E-01 | ns |
| | | LpTVA | 0.04 | 2.05E-01 | 6.53E-01 | ns |
| | | RpTVA | 0.03 | 3.16E-01 | 7.66E-01 | ns |
| | | LaTVA | 0.12 | 1.40E-03 | 5.04E-01 | ns |
| | | RaTVA | 0.07 | 4.26E-02 | 6.61E-01 | ns |
| | VLS | LA1 | 0.09 | 6.52E-02 | 6.53E-02 | ns |
| | | RA1 | 0.08 | 7.47E-02 | 7.49E-02 | ns |
| | | LmTVA | 0.11 | 1.28E-01 | 1.28E-01 | ns |
| | | RmTVA | 0.10 | 1.39E-01 | 1.39E-01 | ns |
| | | LpTVA | 0.09 | 1.94E-01 | 1.94E-01 | ns |
| | | RpTVA | 0.11 | 1.18E-01 | 1.18E-01 | ns |
| | | LaTVA | 0.19 | 4.17E-02 | 4.17E-02 | * |
| | | RaTVA | 0.13 | 1.30E-01 | 1.30E-01 | ns |
| s2 | LIN | LA1 | −0.01 | 5.03E-01 | 6.27E-01 | ns |
| | | RA1 | −0.00 | 1.87E-01 | 4.95E-01 | ns |
| | | LmTVA | 0.01 | 4.72E-01 | 7.19E-01 | ns |
| | | RmTVA | −0.01 | 7.98E-01 | 9.03E-01 | ns |
| | | LpTVA | −0.01 | 7.21E-01 | 8.13E-01 | ns |
| | | RpTVA | 0.00 | 4.07E-01 | 6.00E-01 | ns |
| | | LaTVA | −0.02 | 8.62E-01 | 9.22E-01 | ns |
| | | RaTVA | −0.02 | 7.69E-01 | 7.92E-01 | ns |
| | VLS | LA1 | 0.02 | 2.36E-01 | 2.52E-01 | ns |
| | | RA1 | 0.03 | 1.12E-01 | 1.12E-01 | ns |
| | | LmTVA | 0.06 | 2.29E-01 | 2.29E-01 | ns |
| | | RmTVA | −0.01 | 8.59E-01 | 9.26E-01 | ns |
| | | LpTVA | −0.02 | 8.89E-01 | 9.85E-01 | ns |
| | | RpTVA | 0.01 | 4.22E-01 | 4.54E-01 | ns |

*Appendix 1—table 6 Continued on next page*

*Appendix 1—table 6 Continued*

| Subject | Model | ROI | Correlation | p-unc | p-corr | corr. sig. |
|---|---|---|---|---|---|---|
| | | LaTVA | 0.03 | 3.37E-01 | 3.38E-01 | ns |
| | | RaTVA | 0.00 | 2.76E-01 | 3.23E-01 | ns |
| s3 | LIN | LA1 | –0.00 | 5.71E-01 | 6.66E-01 | ns |
| | | RA1 | 0.05 | 3.00E-04 | 5.00E-02 | * |
| | | LmTVA | 0.05 | 4.10E-03 | 2.04E-01 | ns |
| | | RmTVA | 0.05 | 2.20E-03 | 1.16E-01 | ns |
| | | LpTVA | 0.05 | 5.80E-03 | 1.73E-01 | ns |
| | | RpTVA | 0.04 | 2.66E-02 | 4.60E-01 | ns |
| | | LaTVA | 0.12 | 0.00E+00 | 7.70E-02 | ns |
| | | RaTVA | 0.03 | 3.35E-02 | 3.26E-01 | ns |
| | VLS | LA1 | 0.02 | 1.78E-01 | 2.10E-01 | ns |
| | | RA1 | 0.07 | 1.42E-02 | 1.42E-02 | * |
| | | LmTVA | 0.11 | 7.20E-03 | 7.20E-03 | ** |
| | | RmTVA | 0.05 | 1.23E-01 | 1.23E-01 | ns |
| | | LpTVA | 0.08 | 5.82E-02 | 5.82E-02 | ns |
| | | RpTVA | 0.13 | 1.56E-02 | 1.56E-02 | * |
| | | LaTVA | 0.23 | 1.00E-04 | 1.00E-04 | **** |
| | | RaTVA | 0.04 | 2.34E-01 | 2.34E-01 | ns |

**Appendix 1—table 7.** Assessing the significance of the RSA brain-model correlation.
This table reports the significance of the RSA brain model performance. The brain model correlation coefficients were computed between the ranked representational dissimilarity matrices. The correlation was compared to 0 using a 'maximum statistics' approach in which they are compared to a distribution of correlation coefficients drawn from a large number of random permutations of the model RDMs' rows and columns while controlling for the number of comparisons performed (Methods) (*Maris and Oostenveld, 2007*), for each participant, model, and ROI.

| Subject | ROI | Correlation VLS | Correlation LIN | p-corr | p-unc | corr. sig. |
|---|---|---|---|---|---|---|
| s1 | LA1 | 0.09 | 0.07 | 4.45E-01 | 2.99E-01 | ns |
| | RA1 | 0.08 | 0.08 | 8.30E-01 | 5.05E-01 | ns |
| | LmTVA | 0.11 | 0.08 | 4.63E-01 | 4.51E-01 | ns |
| | RmTVA | 0.10 | 0.06 | 3.98E-01 | 3.97E-01 | ns |
| | LpTVA | 0.09 | 0.04 | 2.86E-01 | 2.84E-01 | ns |
| | RpTVA | 0.11 | 0.03 | 1.11E-01 | 1.11E-01 | ns |
| | LaTVA | 0.19 | 0.12 | 3.94E-01 | 3.94E-01 | ns |
| | RaTVA | 0.13 | 0.07 | 3.48E-01 | 3.48E-01 | ns |
| s2 | LA1 | 0.02 | –0.01 | 3.25E-01 | 1.65E-01 | ns |
| | RA1 | 0.03 | –0.00 | 1.58E-01 | 1.41E-01 | ns |
| | LmTVA | 0.06 | 0.01 | 1.78E-01 | 1.72E-01 | ns |
| | RmTVA | –0.01 | –0.01 | 1.00E+00 | 8.15E-01 | ns |
| | LpTVA | –0.02 | –0.01 | 1.00E+00 | 8.72E-01 | ns |

*Appendix 1—table 7 Continued*

| Subject | ROI | Correlation VLS | Correlation LIN | p-corr | p-unc | corr. sig. |
|---|---|---|---|---|---|---|
| | RpTVA | 0.01 | 0.00 | 7.13E-01 | 4.47E-01 | ns |
| | LaTVA | 0.03 | –0.02 | 1.20E-01 | 1.19E-01 | ns |
| | RaTVA | 0.00 | –0.02 | 3.94E-01 | 1.13E-01 | ns |
| s3 | LA1 | 0.02 | –0.00 | 3.22E-01 | 1.05E-01 | ns |
| | RA1 | 0.07 | 0.05 | 4.83E-01 | 3.22E-01 | ns |
| | LmTVA | 0.11 | 0.05 | 6.61E-02 | 6.25E-02 | ns |
| | RmTVA | 0.05 | 0.05 | 1.00E+00 | 5.38E-01 | ns |
| | LpTVA | 0.08 | 0.05 | 4.30E-01 | 3.08E-01 | ns |
| | RpTVA | 0.13 | 0.04 | 3.66E-02 | 3.66E-02 | * |
| | LaTVA | 0.23 | 0.12 | 1.75E-02 | 1.75E-02 | * |
| | RaTVA | 0.04 | 0.03 | 7.67E-01 | 6.08E-01 | ns |

**Appendix 1—table 8.** Comparing the performance of the RSA models.

This table reports the significance of the RSA brain-model difference. We compared the correlation coefficients between brain RDM and VLS RDM with those from the brain RDM and LIN RDM within participants and hemispheres using one-tailed tests, based on the a priori hypothesis that the VLS models would exhibit greater brain-model correlations than the LIN models (Methods).

| Model | ROI | Accuracy (%) | s.e.m. | T | dof | p-val | unc. sig. | CI95% | cohen-d | BF10 | power |
|---|---|---|---|---|---|---|---|---|---|---|---|
| LIN | LA1 | 43.33±2.22 | 0.51 | n/a | n/a | n/a | n/a | n/a | n/a | n/a | n/a |
| | RA1 | 50.83±1.98 | 0.46 | 1.83E+00 | 19 | 4.14E-02 | * | [50.05, inf] | 4.10E-01 | 1.89E+00 | 0.55 |
| | LmTVA | 38.89±0.00 | 0.00 | n/a | n/a | n/a | n/a | n/a | n/a | n/a | n/a |
| | RmTVA | 61.39±1.21 | 0.28 | 4.10E+01 | 19 | 2.61E-20 | **** | [60.91, inf] | 9.17E+00 | 1.04E+17 | 1.00 |
| | LpTVA | 66.67±0.02 | 0.00 | 4.59E+03 | 19 | 3.32E-59 | **** | [66.66, inf] | 1.03E+03 | 6.88E+33 | 1.00 |
| | RpTVA | 77.50±1.21 | 0.28 | 9.90E+01 | 19 | 1.51E-27 | **** | [77.02, inf] | 2.21E+01 | 7.38E+23 | 1.00 |
| | LaTVA | 44.44±0.00 | 0.00 | n/a | n/a | n/a | n/a | n/a | n/a | n/a | n/a |
| | RaTVA | 44.44±0.00 | 0.00 | n/a | n/a | n/a | n/a | n/a | n/a | n/a | n/a |
| | A1 | 47.08±4.30 | 0.69 | n/a | n/a | n/a | n/a | n/a | n/a | n/a | n/a |
| | mTVA | 50.14±11.28 | 1.81 | 8.00E-02 | 39 | 4.70E-01 | ns | [47.09, inf] | 1.00E-02 | 3.42E-01 | 0.06 |
| | pTVA | 72.08±5.48 | 0.88 | 2.51E+01 | 39 | 5.18E-26 | **** | [70.60, inf] | 3.98E+00 | 5.89E+22 | 1.00 |
| | aTVA | 44.44±0.00 | 0.00 | n/a | n/a | n/a | n/a | n/a | n/a | n/a | n/a |
| | TVAs | 55.56±13.94 | 1.28 | 4.35E+00 | 119 | 1.47E-05 | **** | [53.44, inf] | 4.00E-01 | 1.08E+03 | 1.00 |
| VLS | LA1 | 61.94±1.98 | 0.46 | 2.62E+01 | 19 | 1.08E-16 | **** | [61.16, inf] | 5.87E+00 | 3.93E+13 | 1.00 |
| | RA1 | 60.28±1.98 | 0.46 | 2.26E+01 | 19 | 1.73E-15 | **** | [59.49, inf] | 5.05E+00 | 2.88E+12 | 1.00 |
| | LmTVA | 55.56±0.02 | 0.00 | 1.53E+03 | 19 | 3.86E-50 | **** | [55.55, inf] | 3.42E+02 | 4.95E+33 | 1.00 |
| | RmTVA | 44.44±0.00 | 0.00 | n/a | n/a | n/a | n/a | n/a | n/a | n/a | n/a |
| | LpTVA | 66.67±0.02 | 0.00 | 4.59E+03 | 19 | 3.32E-59 | **** | [66.66, inf] | 1.03E+03 | 6.88E+33 | 1.00 |
| | RpTVA | 61.11±0.02 | 0.00 | 3.06E+03 | 19 | 7.36E-56 | **** | [61.10, inf] | 6.85E+02 | 6.53E+33 | 1.00 |
| | LaTVA | 50.83±1.98 | 0.46 | 1.83E+00 | 19 | 4.14E-02 | * | [50.05, inf] | 4.10E-01 | 1.89E+00 | 0.55 |
| | RaTVA | 44.17±1.21 | 0.28 | n/a | n/a | n/a | n/a | n/a | n/a | n/a | n/a |
| | A1 | 61.11±2.15 | 0.34 | 3.22E+01 | 39 | 4.92E-30 | **** | [60.53, inf] | 5.10E+00 | 4.78E+26 | 1.00 |
| | mTVA | 50.00±5.56 | 0.89 | n/a | n/a | n/a | n/a | n/a | n/a | n/a | n/a |

*Appendix 1—table 8 Continued on next page*

*Appendix 1—table 8 Continued*

| Model | ROI | Accuracy (%) | s.e.m. | T | dof | p-val | unc. sig. | CI95% | cohen-d | BF10 | power |
|-------|-----|--------------|--------|---|-----|-------|-----------|-------|---------|------|-------|
| | pTVA | 63.89±2.78 | 0.44 | 3.12E+01 | 39 | 1.65E-29 | **** | [63.14, inf] | 4.94E+00 | 1.47E+26 | 1.00 |
| | aTVA | 47.50±3.72 | 0.60 | n/a | n/a | n/a | n/a | n/a | n/a | n/a | n/a |
| | TVAs | 53.80±8.33 | 0.76 | 4.97E+00 | 119 | 1.14E-06 | **** | [52.53, inf] | 4.50E-01 | 1.20E+04 | 1.00 |

**Appendix 1—table 9.** Assessing the significance of speaker gender decoding performance using VLS and LIN models based on voxel activity.

This table reports the significance of the speaker's gender decoding performance. Linear classifiers were pre-trained to detect speaker gender (2 classes) from either the VLS or the LIN models. The speaker gender of the 18 Test Stimuli (3 participants x 6 stimuli per participant) was classified using either the VLS coordinates or the LIN features with these classifiers. We used one-sample t-tests to compare the mean of the accuracy distribution across 20 random classifier initializations (20 classifiers trained with a different initialization seed) with a chance level of 50%. s.e.m.=standard error of the mean. Here are reported the results of the statistical tests, t-value, degree of freedom (dof), p-value, degree of significance (unc. sig.), 95% confidence interval (CI95%), effect size (Cohen-d), Bayes Factor (BF10), and statistical power (power) for each model and ROI.

| Model | ROI | Accuracy (%) | s.e.m. | T | dof | p-val | unc. sig. | CI95% | cohen-d | BF10 | power |
|-------|-----|--------------|--------|---|-----|-------|-----------|-------|---------|------|-------|
| LIN | LA1 | 50.42±4.15 | 0.95 | 4.40E-01 | 19 | 3.33E-01 | ns | [48.77, inf] | 1.00E-01 | 5.07E-01 | 0.11 |
| | RA1 | 10.83±3.82 | 0.88 | n/a | n/a | n/a | n/a | n/a | n/a | n/a | n/a |
| | LmTVA | 44.17±3.82 | 0.88 | n/a | n/a | n/a | n/a | n/a | n/a | n/a | n/a |
| | RmTVA | 50.42±6.71 | 1.54 | 2.70E-01 | 19 | 3.95E-01 | ns | [47.76, inf] | 6.00E-02 | 4.80E-01 | 0.08 |
| | LpTVA | 52.50±3.82 | 0.88 | 2.85E+00 | 19 | 5.08E-03 | ** | [50.99, inf] | 6.40E-01 | 1.01E+01 | 0.87 |
| | RpTVA | 56.67±3.33 | 0.76 | 8.72E+00 | 19 | 2.29E-08 | **** | [55.34, inf] | 1.95E+00 | 6.13E+05 | 1.00 |
| | LaTVA | 52.50±3.82 | 0.88 | 2.85E+00 | 19 | 5.08E-03 | ** | [50.99, inf] | 6.40E-01 | 1.01E+01 | 0.87 |
| | RaTVA | 75.42±6.17 | 1.41 | 1.80E+01 | 19 | 1.11E-13 | **** | [72.97, inf] | 4.02E+00 | 5.71E+10 | 1.00 |
| | A1 | 30.62±20.19 | 3.23 | n/a | n/a | n/a | n/a | n/a | n/a | n/a | n/a |
| | mTVA | 47.29±6.29 | 1.01 | n/a | n/a | n/a | n/a | n/a | n/a | n/a | n/a |
| | pTVA | 54.58±4.15 | 0.66 | 6.90E+00 | 39 | 1.45E-08 | **** | [53.46, inf] | 1.09E+00 | 9.41E+05 | 1.00 |
| | aTVA | 63.96±12.55 | 2.01 | 6.94E+00 | 39 | 1.28E-08 | **** | [60.57, inf] | 1.10E+00 | 1.06E+06 | 1.00 |
| | TVAs | 55.28±10.86 | 1.00 | 5.30E+00 | 119 | 2.69E-07 | **** | [53.63, inf] | 4.80E-01 | 4.69E+04 | 1.00 |
| VLS | LA1 | 66.67±0.02 | 0.00 | 4.59E+03 | 19 | 3.32E-59 | **** | [66.66, inf] | 1.03E+03 | 6.88E+33 | 1.00 |
| | RA1 | 8.33±0.00 | 0.00 | n/a | n/a | n/a | n/a | n/a | n/a | n/a | n/a |
| | LmTVA | 49.17±12.61 | 2.89 | n/a | n/a | n/a | n/a | n/a | n/a | n/a | n/a |
| | RmTVA | 41.67±0.00 | 0.00 | n/a | n/a | n/a | n/a | n/a | n/a | n/a | n/a |
| | LpTVA | 58.33±0.02 | 0.00 | 2.30E+03 | 19 | 1.74E-53 | **** | [58.33, inf] | 5.14E+02 | 6.07E+33 | 1.00 |
| | RpTVA | 71.67±4.08 | 0.94 | 2.31E+01 | 19 | 1.11E-15 | **** | [70.05, inf] | 5.17E+00 | 4.36E+12 | 1.00 |
| | LaTVA | 56.67±3.33 | 0.76 | 8.72E+00 | 19 | 2.29E-08 | **** | [55.34, inf] | 1.95E+00 | 6.13E+05 | 1.00 |
| | RaTVA | 64.17±3.82 | 0.88 | 1.62E+01 | 19 | 7.29E-13 | **** | [62.65, inf] | 3.62E+00 | 9.69E+09 | 1.00 |
| | A1 | 37.50±29.17 | 4.67 | n/a | n/a | n/a | n/a | n/a | n/a | n/a | n/a |
| | mTVA | 45.42±9.67 | 1.55 | n/a | n/a | n/a | n/a | n/a | n/a | n/a | n/a |
| | pTVA | 65.00±7.26 | 1.16 | 1.29E+01 | 39 | 6.05E-16 | **** | [63.04, inf] | 2.04E+00 | 1.05E+13 | 1.00 |
| | aTVA | 60.42±5.19 | 0.83 | 1.25E+01 | 39 | 1.46E-15 | **** | [59.02, inf] | 1.98E+00 | 4.51E+12 | 1.00 |
| | TVAs | 56.94±11.30 | 1.04 | 6.70E+00 | 119 | 3.59E-10 | **** | [55.23, inf] | 6.10E-01 | 2.64E+07 | 1.00 |

**Appendix 1—table 10.** Assessing the significance of speaker age decoding performance using VLS and LIN models based on voxel activity.

This table reports the significance of the speaker age decoding performance. Linear classifiers were pre-trained to detect speaker age (2 classes) from either the VLS or the LIN models. The speaker age of the 18 Test Stimuli (3 participants x 6 stimuli per participant) was classified using either the VLS or LIN coordinates with these classifiers. We used one-sample t-tests to compare the mean of the accuracy distribution across 20 random classifier initializations (20 classifiers trained with a different initialization seed) with the chance level of 50%. s.e.m.=standard error of the mean. Here are reported the results of the statistical tests, t-value, degree of freedom (dof), p-value, and degree of significance (unc. sig.), 95% confidence interval (CI95%), effect size (Cohen-d), Bayes Factor (BF10), and statistical power (power) for each model and ROI.

| Model | ROI | Accuracy (%) | s.e.m. | T | dof | p-val | unc. sig. | CI95% | cohen-d | BF10 | power |
|---|---|---|---|---|---|---|---|---|---|---|---|
| LIN | LA1 | 0.29±1.28 | 0.29 | n/a | n/a | n/a | n/a | n/a | n/a | n/a | n/a |
| | RA1 | 18.09±3.26 | 0.75 | 1.63E+01 | 19 | 6.14E-13 | **** | [16.80, inf] | 3.65E+00 | 1.14E+10 | 1.00 |
| | LmTVA | 11.18±4.01 | 0.92 | 5.75E+00 | 19 | 7.61E-06 | **** | [9.59, inf] | 1.29E+00 | 3.01E+03 | 1.00 |
| | RmTVA | 2.35±3.03 | 0.69 | n/a | n/a | n/a | n/a | n/a | n/a | n/a | n/a |
| | LpTVA | 12.21±3.39 | 0.78 | 8.13E+00 | 19 | 6.54E-08 | **** | [10.86, inf] | 1.82E+00 | 2.32E+05 | 1.00 |
| | RpTVA | 6.76±4.66 | 1.07 | 8.30E-01 | 19 | 2.10E-01 | ns | [4.92, inf] | 1.80E-01 | 6.29E-01 | 0.20 |
| | LaTVA | 11.47±1.28 | 0.29 | 1.90E+01 | 19 | 4.04E-14 | **** | [10.96, inf] | 4.25E+00 | 1.48E+11 | 1.00 |
| | RaTVA | 7.35±8.29 | 1.90 | 7.70E-01 | 19 | 2.25E-01 | ns | [4.06, inf] | 1.70E-01 | 6.07E-01 | 0.18 |
| | A1 | 9.19±9.24 | 1.48 | 2.24E+00 | 39 | 1.55E-02 | * | [6.70, inf] | 3.50E-01 | 3.15E+00 | 0.71 |
| | mTVA | 6.76±5.67 | 0.91 | 9.70E-01 | 39 | 1.68E-01 | ns | [5.24, inf] | 1.50E-01 | 5.30E-01 | 0.25 |
| | pTVA | 9.49±4.90 | 0.78 | 4.59E+00 | 39 | 2.24E-05 | **** | [8.16, inf] | 7.30E-01 | 1.01E+03 | 1.00 |
| | aTVA | 9.41±6.28 | 1.01 | 3.51E+00 | 39 | 5.75E-04 | *** | [7.72, inf] | 5.50E-01 | 5.39E+01 | 0.96 |
| | TVAs | 8.55±5.78 | 0.53 | 5.04E+00 | 119 | 8.44E-07 | **** | [7.68, inf] | 4.60E-01 | 1.59E+04 | 1.00 |
| VLS | LA1 | 0.15±0.64 | 0.15 | n/a | n/a | n/a | n/a | n/a | n/a | n/a | n/a |
| | RA1 | 11.47±5.09 | 1.17 | 4.79E+00 | 19 | 6.37E-05 | **** | [9.45, inf] | 1.07E+00 | 4.49E+02 | 1.00 |
| | LmTVA | 11.47±4.73 | 1.09 | 5.15E+00 | 19 | 2.87E-05 | **** | [9.59, inf] | 1.15E+00 | 9.13E+02 | 1.00 |
| | RmTVA | 0.59±1.50 | 0.34 | n/a | n/a | n/a | n/a | n/a | n/a | n/a | n/a |
| | LpTVA | 9.71±3.37 | 0.77 | 4.95E+00 | 19 | 4.44E-05 | **** | [8.37, inf] | 1.11E+00 | 6.19E+02 | 1.00 |
| | RpTVA | 22.65±2.10 | 0.48 | 3.48E+01 | 19 | 5.69E-19 | **** | [21.81, inf] | 7.78E+00 | 5.62E+15 | 1.00 |
| | LaTVA | 10.29±4.51 | 1.03 | 4.27E+00 | 19 | 2.09E-04 | *** | [8.51, inf] | 9.50E-01 | 1.57E+02 | 0.99 |
| | RaTVA | 6.18±3.94 | 0.90 | 3.30E-01 | 19 | 3.74E-01 | ns | [4.62, inf] | 7.00E-02 | 4.88E-01 | 0.09 |
| | A1 | 5.81±6.72 | 1.08 | n/a | n/a | n/a | n/a | n/a | n/a | n/a | n/a |
| | mTVA | 6.03±6.48 | 1.04 | 1.40E-01 | 39 | 4.44E-01 | ns | [4.28, inf] | 2.00E-02 | 3.44E-01 | 0.07 |
| | pTVA | 16.18±7.05 | 1.13 | 9.12E+00 | 39 | 1.65E-11 | **** | [14.27, inf] | 1.44E+00 | 5.85E+08 | 1.00 |
| | aTVA | 8.24±4.71 | 0.75 | 3.12E+00 | 39 | 1.69E-03 | ** | [6.97, inf] | 4.90E-01 | 2.09E+01 | 0.92 |
| | TVAs | 10.15±7.55 | 0.69 | 6.17E+00 | 119 | 4.96E-09 | **** | [9.00, inf] | 5.60E-01 | 2.12E+06 | 1.00 |

**Appendix 1—table 11.** Assessing the significance of speaker identity decoding performance using VLS and LIN models based on voxel activity.

This table reports the significance of the speaker decoding performance. Linear classifiers were pre-trained to detect speaker identity (17 classes) from either the VLS or the LIN models. The speaker identity of the 18 Test Stimuli (3 participants x 6 stimuli per participant) was classified using either the VLS or LIN coordinates with these classifiers. We used one-sample t-tests to compare the mean of the accuracy distribution across 20 random classifier initializations (20 classifiers trained with a different initialization seed) with the chance level of 5.88%. s.e.m.=standard error of the mean. Here are reported the results of the statistical tests, t-value, degree of freedom (dof), p-value, and degree of significance (unc. sig.), 95% confidence interval (CI95%), effect size (Cohen-d), Bayes Factor (BF10), and statistical power (power) for each model and ROI.

| Category | ROI | Accuracy VLS (%) | Accuracy LIN (%) | s.e.m. VLS | s.e.m. LIN | T VLS vs LIN | dof | p-val | unc. sig. | CI95% | cohen-d | BF10 | power |
|---|---|---|---|---|---|---|---|---|---|---|---|---|---|
| Gender | LA1 | 61.94±1.98 | 43.33±2.22 | 0.46 | 0.51 | 3.06E+01 | 19 | 1.24E-17 | **** | [17.34, 19.88] | 8.610000 | 2.94E+14 | 1.00 |
| | RA1 | 60.28±1.98 | 50.83±1.98 | 0.46 | 0.46 | 1.62E+01 | 19 | 1.46E-12 | **** | [8.22, 10.67] | 4.640000 | 4.85E+09 | 1.00 |
| | LmTVA | 55.56±0.02 | 38.89±0.02 | 0.00 | 0.00 | INF | 19 | 0.00E+00 | **** | [nan, nan] | 1027.400000 | nan | 1.00 |
| | RmTVA | 44.44±0.02 | 61.39±1.21 | 0.00 | 0.28 | –6.10E+01 | 19 | 2.91E-23 | **** | [–17.53,–16.36] | 19.290000 | 6.23E+19 | 1.00 |
| | LpTVA | 66.67±0.02 | 66.67±0.02 | 0.00 | 0.00 | nan | 19 | nan | ns | [nan, nan] | 0.000000 | nan | 0.05 |
| | RpTVA | 61.11±0.02 | 77.50±1.21 | 0.00 | 0.28 | –5.90E+01 | 19 | 5.47E-23 | **** | [–16.97,–15.81] | 18.660000 | 3.43E+19 | 1.00 |
| | LaTVA | 50.83±1.98 | 44.44±0.02 | 0.46 | 0.00 | 1.41E+01 | 19 | 1.65E-11 | **** | [5.44, 7.34] | 4.440000 | 4.96E+08 | 1.00 |
| | RaTVA | 44.17±1.21 | 44.44±0.02 | 0.28 | 0.00 | –1.00E+00 | 19 | 3.30E-01 | ns | [–0.86, 0.30] | 0.320000 | 3.61E-01 | 0.27 |
| | A1 | 61.11±2.15 | 47.08±4.30 | 0.34 | 0.69 | 1.66E+01 | 39 | 2.67E-19 | **** | [12.32, 15.73] | 4.070000 | 1.78E+16 | 1.00 |
| | mTVA | 50.00±5.56 | 50.14±11.28 | 0.89 | 1.81 | –5.00E-02 | 39 | 9.59E-01 | ns | [–5.59, 5.31] | 0.020000 | 1.71E-01 | 0.05 |
| | pTVA | 63.89±2.78 | 72.08±5.48 | 0.44 | 0.88 | –6.21E+00 | 39 | 2.64E-07 | **** | [–10.86,–5.53] | 1.860000 | 5.92E+04 | 1.00 |
| | aTVA | 47.50±3.72 | 44.44±0.01 | 0.60 | 0.00 | 5.14E+00 | 39 | 8.05E-06 | **** | [1.85, 4.26] | 1.150000 | 2.45E+03 | 1.00 |
| | TVAs | 53.80±8.33 | 55.56±13.94 | 0.76 | 1.28 | –1.60E+00 | 119 | 1.12E-01 | ns | [–3.94, 0.42] | 0.150000 | 3.49E-01 | 0.38 |
| Age | LA1 | 66.67±0.02 | 50.42±4.15 | 0.00 | 0.95 | 1.71E+01 | 19 | 5.58E-13 | **** | [14.26, 18.24] | 5.400000 | 1.20E+10 | 1.00 |
| | RA1 | 8.33±0.02 | 10.83±3.82 | 0.00 | 0.88 | –2.85E+00 | 19 | 1.03E-02 | * | [–4.34,–0.66] | 0.900000 | 5.02E+00 | 0.97 |
| | LmTVA | 49.17±12.61 | 44.17±3.82 | 2.89 | 0.88 | 1.71E+00 | 19 | 1.04E-01 | ns | [–1.12, 11.12] | 0.520000 | 7.97E-01 | 0.60 |
| | RmTVA | 41.67±0.02 | 50.42±6.71 | 0.00 | 1.54 | –5.69E+00 | 19 | 1.76E-05 | **** | [–11.97,–5.53] | 1.800000 | 1.32E+03 | 1.00 |
| | LpTVA | 58.33±0.02 | 52.50±3.82 | 0.00 | 0.88 | 6.67E+00 | 19 | 2.24E-06 | **** | [4.00, 7.66] | 2.110000 | 8.58E+03 | 1.00 |
| | RpTVA | 71.67±4.08 | 56.67±3.33 | 0.94 | 0.76 | 1.16E+01 | 19 | 4.80E-10 | **** | [12.29, 17.71] | 3.920000 | 2.12E+07 | 1.00 |
| | LaTVA | 56.67±3.33 | 52.50±3.82 | 0.76 | 0.88 | 3.68E+00 | 19 | 1.58E-03 | ** | [1.80, 6.53] | 1.130000 | 2.46E+01 | 1.00 |
| | RaTVA | 64.17±3.82 | 75.42±6.17 | 0.88 | 1.41 | –6.90E+00 | 19 | 1.40E-06 | **** | [–14.66,–7.84] | 2.140000 | 1.31E+04 | 1.00 |
| | A1 | 37.50±29.17 | 30.62±20.19 | 4.67 | 3.23 | 4.21E+00 | 39 | 1.43E-04 | *** | [3.58, 10.17] | 0.270000 | 1.75E+02 | 0.39 |
| | mTVA | 45.42±9.67 | 47.29±6.29 | 1.55 | 1.01 | –9.50E-01 | 39 | 3.47E-01 | ns | [–5.86, 2.11] | 0.230000 | 2.60E-01 | 0.29 |
| | pTVA | 65.00±7.26 | 54.58±4.15 | 1.16 | 0.66 | 9.78E+00 | 39 | 4.83E-12 | **** | [8.26, 12.57] | 1.740000 | 1.83E+09 | 1.00 |
| | aTVA | 60.42±5.19 | 63.96±12.55 | 0.83 | 2.01 | –2.25E+00 | 39 | 3.03E-02 | * | [–6.73,–0.35] | 0.360000 | 1.61E+00 | 0.61 |
| | TVAs | 56.94±11.30 | 55.28±10.86 | 1.04 | 1.00 | 1.56E+00 | 119 | 1.22E-01 | ns | [–0.45, 3.78] | 0.150000 | 3.28E-01 | 0.37 |
| Identity | LA1 | 0.15±0.64 | 0.29±1.28 | 0.15 | 0.29 | –4.40E-01 | 19 | 6.66E-01 | ns | [–0.85, 0.56] | 0.140000 | 2.53E-01 | 0.09 |
| | RA1 | 11.47±5.09 | 18.09±3.26 | 1.17 | 0.75 | –4.58E+00 | 19 | 2.05E-04 | *** | [–9.64,–3.59] | 1.510000 | 1.47E+02 | 1.00 |
| | LmTVA | 11.47±4.73 | 11.18±4.01 | 1.09 | 0.92 | 2.10E-01 | 19 | 8.39E-01 | ns | [–2.70, 3.29] | 0.070000 | 2.37E-01 | 0.06 |
| | RmTVA | 0.59±1.50 | 2.35±3.03 | 0.34 | 0.69 | –2.11E+00 | 19 | 4.86E-02 | * | [–3.52,–0.01] | 0.720000 | 1.42E+00 | 0.86 |
| | LpTVA | 9.71±3.37 | 12.21±3.39 | 0.77 | 0.78 | –2.43E+00 | 19 | 2.53E-02 | * | [–4.65,–0.35] | 0.720000 | 2.39E+00 | 0.86 |
| | RpTVA | 22.65±2.10 | 6.76±4.66 | 0.48 | 1.07 | 1.31E+01 | 19 | 5.99E-11 | **** | [13.34, 18.42] | 4.280000 | 1.48E+08 | 1.00 |
| | LaTVA | 10.29±4.51 | 11.47±1.28 | 1.03 | 0.29 | –1.00E+00 | 19 | 3.30E-01 | ns | [–3.64, 1.29] | 0.350000 | 3.61E-01 | 0.31 |
| | RaTVA | 6.18±3.94 | 7.35±8.29 | 0.90 | 1.90 | –7.50E-01 | 19 | 4.64E-01 | ns | [–4.47, 2.12] | 0.180000 | 2.98E-01 | 0.12 |

*Appendix 1—table 11 Continued on next page*

*Appendix 1—table11 Continued*

| Category | ROI | Accuracy VLS (%) | Accuracy LIN (%) | s.e.m. VLS | s.e.m. LIN | T VLS vs LIN | dof | p-val | unc. sig. | CI95% | cohen-d | BF10 | power |
|---|---|---|---|---|---|---|---|---|---|---|---|---|---|
| | A1 | 5.81±6.72 | 9.19±9.24 | 1.08 | 1.48 | –3.77E+00 | 39 | 5.40E-04 | *** | [-5.20,–1.57] | 0.410000 | 5.30E+01 | 0.72 |
| | mTVA | 6.03±6.48 | 6.76±5.67 | 1.04 | 0.91 | –8.80E-01 | 39 | 3.83E-01 | ns | [–2.42, 0.95] | 0.120000 | 2.45E-01 | 0.11 |
| | pTVA | 16.18±7.05 | 9.49±4.90 | 1.13 | 0.78 | 4.01E+00 | 39 | 2.65E-04 | *** | [3.32, 10.07] | 1.090000 | 1.00E+02 | 1.00 |
| | aTVA | 8.24±4.71 | 9.41±6.28 | 0.75 | 1.01 | –1.21E+00 | 39 | 2.32E-01 | ns | [–3.14, 0.79] | 0.210000 | 3.37E-01 | 0.25 |
| | TVAs | 10.15±7.55 | 8.55±5.78 | 0.69 | 0.53 | 2.07E+00 | 119 | 4.06E-02 | * | [0.07, 3.12] | 0.240000 | 7.94E-01 | 0.73 |

**Appendix 1—table 12.** Comparing the performance of the models decoding speaker identity-related information. This table reports the significance of the speaker identity decoding VLS-LIN difference. Paired t-tests were conducted between the mean scores of linear classifiers pre-trained to detect gender (2 classes), age (2 classes), and identity (17 classes) from the VLS features, and those trained with the LIN features. These scores were obtained after classifying the VLS or LIN coordinates of the 18 Test Stimuli (3 participants x 6 stimuli per participant). s.e.m.=standard error of the mean. Here are reported the results of the statistical tests, t-value, degree of freedom (dof), p-value, and degree of significance (unc. sig.), 95% confidence interval (CI95%), effect size (Cohen-d), Bayes Factor (BF10), and statistical power (power) for each speaker information and ROI.

| Category | Model | ROI | Accuracy ROI (%) | Accuracy A1 (%) | s.e.m. ROI | s.e.m. A1 | T ROI vs A1 | dof | p-val | unc. sig. | CI95% | cohen-d | BF10 | power |
|---|---|---|---|---|---|---|---|---|---|---|---|---|---|---|
| Gender | LIN | mTVA | 50.14±11.28 | 47.08±4.30 | 1.81 | 0.69 | 1.58E+00 | 78 | 1.20E-01 | ns | [–0.79, 6.90] | 3.50E-01 | 6.83E-01 | 0.35 |
| | | pTVA | 72.08±5.48 | 47.08±4.30 | 0.88 | 0.69 | 2.24E+01 | 78 | 0.00E+00 | **** | [22.78, 27.22] | 5.01E+00 | 1.30E+32 | 1.00 |
| | | aTVA | 44.44±0.00 | 47.08±4.30 | 0.00 | 0.69 | –3.83E+00 | 78 | 0.00E+00 | *** | [-4.01,–1.27] | 8.60E-01 | 9.78E+01 | 0.97 |
| | | TVAs | 55.56±13.94 | 47.08±4.30 | 1.28 | 0.69 | 3.76E+00 | 158 | 0.00E+00 | *** | [4.02, 12.92] | 6.90E-01 | 9.90E+01 | 0.96 |
| | VLS | mTVA | 50.00±5.56 | 61.11±2.15 | 0.89 | 0.34 | –1.17E+01 | 78 | 0.00E+00 | **** | [-13.01,–9.21] | 2.60E+00 | 3.45E+15 | 1.00 |
| | | pTVA | 63.89±2.78 | 61.11±2.15 | 0.44 | 0.34 | 4.94E+00 | 78 | 0.00E+00 | **** | [1.66, 3.90] | 1.10E+00 | 3.59E+03 | 1.00 |
| | | aTVA | 47.50±3.72 | 61.11±2.15 | 0.60 | 0.34 | –1.98E+01 | 78 | 0.00E+00 | **** | [-14.98,–12.24] | 4.43E+00 | 4.06E+28 | 1.00 |
| | | TVAs | 53.80±8.33 | 61.11±2.15 | 0.76 | 0.34 | –5.46E+00 | 158 | 0.00E+00 | **** | [-9.96,–4.67] | 1.00E+00 | 6.55E+04 | 1.00 |
| Age | LIN | mTVA | 47.29±6.29 | 30.62±20.19 | 1.01 | 3.23 | 4.92E+00 | 78 | 0.00E+00 | **** | [9.93, 23.41] | 1.10E+00 | 3.41E+03 | 1.00 |
| | | pTVA | 54.58±4.15 | 30.62±20.19 | 0.66 | 3.23 | 7.26E+00 | 78 | 0.00E+00 | **** | [17.39, 30.53] | 1.62E+00 | 3.02E+07 | 1.00 |
| | | aTVA | 63.96±12.55 | 30.62±20.19 | 2.01 | 3.23 | 8.76E+00 | 78 | 0.00E+00 | **** | [25.75, 40.91] | 1.96E+00 | 1.68E+10 | 1.00 |
| | | TVAs | 55.28±10.86 | 30.62±20.19 | 1.00 | 3.23 | 9.72E+00 | 158 | 0.00E+00 | **** | [19.65, 29.66] | 1.78E+00 | 4.55E+14 | 1.00 |
| | VLS | mTVA | 45.42±9.67 | 37.50±29.17 | 1.55 | 4.67 | 1.61E+00 | 78 | 1.10E-01 | ns | [–1.88, 17.71] | 3.60E-01 | 7.10E-01 | 0.36 |
| | | pTVA | 65.00±7.26 | 37.50±29.17 | 1.16 | 4.67 | 5.71E+00 | 78 | 0.00E+00 | **** | [17.92, 37.08] | 1.28E+00 | 6.21E+04 | 1.00 |
| | | aTVA | 60.42±5.19 | 37.50±29.17 | 0.83 | 4.67 | 4.83E+00 | 78 | 0.00E+00 | **** | [13.47, 32.36] | 1.08E+00 | 2.48E+03 | 1.00 |
| | | TVAs | 56.94±11.30 | 37.50±29.17 | 1.04 | 4.67 | 6.03E+00 | 158 | 0.00E+00 | **** | [13.07, 25.82] | 1.10E+00 | 8.68E+05 | 1.00 |
| Identity | LIN | mTVA | 6.76±5.67 | 9.19±9.24 | 0.91 | 1.48 | –1.40E+00 | 78 | 1.70E-01 | ns | [–5.88, 1.03] | 3.10E-01 | 5.42E-01 | 0.28 |
| | | pTVA | 9.49±4.90 | 9.19±9.24 | 0.78 | 1.48 | 1.80E-01 | 78 | 8.60E-01 | ns | [–3.04, 3.63] | 4.00E-02 | 2.36E-01 | 0.05 |
| | | aTVA | 9.41±6.28 | 9.19±9.24 | 1.01 | 1.48 | 1.20E-01 | 78 | 9.00E-01 | ns | [–3.34, 3.78] | 3.00E-02 | 2.34E-01 | 0.05 |
| | | TVAs | 8.55±5.78 | 9.19±9.24 | 0.53 | 1.48 | –5.10E-01 | 158 | 6.10E-01 | ns | [–3.11, 1.83] | 9.00E-02 | 2.19E-01 | 0.08 |
| | VLS | mTVA | 6.03±6.48 | 5.81±6.72 | 1.04 | 1.08 | 1.50E-01 | 78 | 8.80E-01 | ns | [–2.76, 3.20] | 3.00E-02 | 2.35E-01 | 0.05 |
| | | pTVA | 16.18±7.05 | 5.81±6.72 | 1.13 | 1.08 | 6.65E+00 | 78 | 0.00E+00 | **** | [7.26, 13.47] | 1.49E+00 | 2.43E+06 | 1.00 |
| | | aTVA | 8.24±4.71 | 5.81±6.72 | 0.75 | 1.08 | 1.85E+00 | 78 | 7.00E-02 | ns | [–0.19, 5.04] | 4.10E-01 | 1.01E+00 | 0.45 |
| | | TVAs | 10.15±7.55 | 5.81±6.72 | 0.69 | 1.08 | 3.21E+00 | 158 | 0.00E+00 | ** | [1.67, 7.00] | 5.90E-01 | 1.91E+01 | 0.89 |

**Appendix 1—table 13.** Comparing the performance of the models decoding speaker identity-related information by ROI. This table reports the significance of the speaker identity decoding A1-TVAs difference. Two-sample t-tests were conducted for each model to determine if there was an A1-TVAs difference between the mean scores of linear classifiers pre-trained to detect gender (2 classes), age (2 classes), and identity (17 classes). These scores were obtained by classifying the VLS coordinates or LIN features, reconstructed by different ROIs, for the 18 Test Stimuli (3 participants x 6 stimuli per participant). s.e.m.=standard error of the mean. Here are reported the results of the statistical tests, t-value, degree of freedom (dof), p-value, degree of significance (unc. sig.), 95% confidence interval (CI95%), effect size (Cohen-d), Bayes Factor (BF10), and statistical power (power) for each speaker information and model.

| Model | ROI | Accuracy (%) | s.e.m. | T | dof | p-val | unc. sig. | CI95% | cohen-d | BF10 | power |
|---|---|---|---|---|---|---|---|---|---|---|---|
| LIN | LA1 | 44.87±7.99 | 2.31 | n/a | n/a | n/a | n/a | n/a | n/a | n/a | n/a |
| | RA1 | 51.28±10.02 | 2.89 | 4.40E-01 | 12 | 3.33E-01 | ns | [46.13, inf] | 1.20E-01 | 6.06E-01 | 0.11 |
| | LmTVA | 51.71±10.76 | 3.11 | 5.50E-01 | 12 | 2.96E-01 | ns | [46.17, inf] | 1.50E-01 | 6.35E-01 | 0.13 |
| | RmTVA | 43.59±8.40 | 2.42 | n/a | n/a | n/a | n/a | n/a | n/a | n/a | n/a |
| | LpTVA | 50.00±8.72 | 2.52 | n/a | n/a | n/a | n/a | n/a | n/a | n/a | n/a |
| | RpTVA | 52.99±10.36 | 2.99 | 1.00E+00 | 12 | 1.69E-01 | ns | [47.66, inf] | 2.80E-01 | 8.49E-01 | 0.24 |
| | LaTVA | 51.28±8.48 | 2.45 | 5.20E-01 | 12 | 3.05E-01 | ns | [46.92, inf] | 1.50E-01 | 6.27E-01 | 0.13 |
| | RaTVA | 45.30±9.71 | 2.80 | n/a | n/a | n/a | n/a | n/a | n/a | n/a | n/a |
| | A1 | 48.08±9.62 | 1.92 | n/a | n/a | n/a | n/a | n/a | n/a | n/a | n/a |
| | mTVA | 47.65±10.47 | 2.09 | n/a | n/a | n/a | n/a | n/a | n/a | n/a | n/a |
| | pTVA | 51.50±9.69 | 1.94 | 7.70E-01 | 25 | 2.24E-01 | ns | [48.18, inf] | 1.50E-01 | 5.43E-01 | 0.19 |
| | aTVA | 48.29±9.59 | 1.92 | n/a | n/a | n/a | n/a | n/a | n/a | n/a | n/a |
| | TVAs | 49.15±10.07 | 1.15 | n/a | n/a | n/a | n/a | n/a | n/a | n/a | n/a |
| VLS | LA1 | 50.00±11.32 | 3.27 | n/a | n/a | n/a | n/a | n/a | n/a | n/a | n/a |
| | RA1 | 61.54±6.34 | 1.83 | 6.31E+00 | 12 | 1.96E-05 | **** | [58.28, inf] | 1.75E+00 | 1.31E+03 | 1.00 |
| | LmTVA | 51.71±5.06 | 1.46 | 1.17E+00 | 12 | 1.32E-01 | ns | [49.11, inf] | 3.20E-01 | 9.85E-01 | 0.29 |
| | RmTVA | 45.73±8.76 | 2.53 | n/a | n/a | n/a | n/a | n/a | n/a | n/a | n/a |
| | LpTVA | 63.25±7.40 | 2.14 | 6.20E+00 | 12 | 2.29E-05 | **** | [59.44, inf] | 1.72E+00 | 1.14E+03 | 1.00 |
| | RpTVA | 60.26±6.48 | 1.87 | 5.48E+00 | 12 | 7.01E-05 | **** | [56.92, inf] | 1.52E+00 | 4.30E+02 | 1.00 |
| | LaTVA | 60.26±6.10 | 1.76 | 5.82E+00 | 12 | 4.10E-05 | **** | [57.12, inf] | 1.61E+00 | 6.86E+02 | 1.00 |
| | RaTVA | 50.00±8.98 | 2.59 | n/a | n/a | n/a | n/a | n/a | n/a | n/a | n/a |
| | A1 | 55.77±10.84 | 2.17 | 2.66E+00 | 25 | 6.70E-03 | ** | [52.07, inf] | 5.20E-01 | 7.39E+00 | 0.83 |
| | mTVA | 48.72±7.75 | 1.55 | n/a | n/a | n/a | n/a | n/a | n/a | n/a | n/a |
| | pTVA | 61.75±7.12 | 1.42 | 8.26E+00 | 25 | 6.56E-09 | **** | [59.32, inf] | 1.62E+00 | 2.00E+06 | 1.00 |
| | aTVA | 55.13±9.24 | 1.85 | 2.78E+00 | 25 | 5.13E-03 | ** | [51.97, inf] | 5.40E-01 | 9.24E+00 | 0.85 |
| | TVAs | 55.20±9.68 | 1.10 | 4.71E+00 | 77 | 5.29E-06 | **** | [53.36, inf] | 5.30E-01 | 3.23E+03 | 1.00 |

**Appendix 1—table 14.** Assessing the significance of the speaker gender categorization task.

This table reports the significance of the speaker's gender categorization performance. 342 voice stimuli were used in the experiments: the original stimuli (N=18), directly reconstructed stimuli using the LIN and the VLS models (N=36), and brain-reconstructed stimuli (18 stimuli x 2 models x 4 regions of interest x 2 hemispheres, N=288). The participants were tasked with identifying the gender of the presented voice in each trial by clicking either the 'Female' or 'Male' button. To evaluate the accuracy of the binary responses, we computed the classification accuracy for each participant and region of interest (ROI). We then utilized one-sample t-tests to compare the mean accuracy distribution across all participants to the chance level of 50%. s.e.m.=standard error of the mean. Here are reported the results of the statistical tests, t-value, degree of freedom (dof), p-value, degree of significance (unc. sig.), 95% confidence interval (CI95%), effect size (Cohen-d), Bayes Factor (BF10), and statistical power (power) for each model and ROI.

| Model | ROI | Accuracy (%) | s.e.m. | T | dof | p-val | unc. sig. | CI95% | cohen-d | BF10 | power |
|-------|-----|--------------|--------|---|-----|-------|-----------|-------|---------|------|-------|
| LIN | LA1 | 46.15±14.48 | 4.18 | n/a | n/a | n/a | n/a | n/a | n/a | n/a | n/a |
| | RA1 | 44.23±11.50 | 3.32 | n/a | n/a | n/a | n/a | n/a | n/a | n/a | n/a |
| | LmTVA | 50.00±9.81 | 2.83 | n/a | n/a | n/a | n/a | n/a | n/a | n/a | n/a |
| | RmTVA | 57.69±12.85 | 3.71 | 2.07E+00 | 12 | 3.02E-02 | * | [51.08, inf] | 5.80E-01 | 2.80E+00 | 0.62 |
| | LpTVA | 50.00±10.34 | 2.98 | 0.00E+00 | 12 | 5.00E-01 | ns | [44.68, inf] | 0.00E+00 | 5.56E-01 | 0.05 |
| | RpTVA | 50.64±10.57 | 3.05 | 2.10E-01 | 12 | 4.19E-01 | ns | [45.20, inf] | 6.00E-02 | 5.67E-01 | 0.07 |
| | LaTVA | 48.72±13.01 | 3.76 | n/a | n/a | n/a | n/a | n/a | n/a | n/a | n/a |
| | RaTVA | 62.82±13.32 | 3.85 | 3.33E+00 | 12 | 2.98E-03 | ** | [55.97, inf] | 9.20E-01 | 1.77E+01 | 0.93 |
| | A1 | 45.19±13.11 | 2.62 | n/a | n/a | n/a | n/a | n/a | n/a | n/a | n/a |
| | mTVA | 53.85±12.06 | 2.41 | 1.59E+00 | 25 | 6.17E-02 | ns | [49.73, inf] | 3.10E-01 | 1.26E+00 | 0.46 |
| | pTVA | 50.32±10.46 | 2.09 | 1.50E-01 | 25 | 4.40E-01 | ns | [46.75, inf] | 3.00E-02 | 4.19E-01 | 0.07 |
| | aTVA | 55.77±14.94 | 2.99 | 1.93E+00 | 25 | 3.24E-02 | * | [50.67, inf] | 3.80E-01 | 2.06E+00 | 0.59 |
| | TVAs | 53.31±12.82 | 1.46 | 2.27E+00 | 77 | 1.31E-02 | * | [50.88, inf] | 2.60E-01 | 2.77E+00 | 0.73 |
| VLS | LA1 | 54.49±10.65 | 3.07 | 1.46E+00 | 12 | 8.50E-02 | ns | [49.01, inf] | 4.00E-01 | 1.32E+00 | 0.39 |
| | RA1 | 50.00±8.01 | 2.31 | 0.00E+00 | 12 | 5.00E-01 | ns | [45.88, inf] | 0.00E+00 | 5.56E-01 | 0.05 |
| | LmTVA | 51.28±10.26 | 2.96 | 4.30E-01 | 12 | 3.36E-01 | ns | [46.01, inf] | 1.20E-01 | 6.04E-01 | 0.11 |
| | RmTVA | 54.49±10.65 | 3.07 | 1.46E+00 | 12 | 8.50E-02 | ns | [49.01, inf] | 4.00E-01 | 1.32E+00 | 0.39 |
| | LpTVA | 45.51±11.14 | 3.22 | n/a | n/a | n/a | n/a | n/a | n/a | n/a | n/a |
| | RpTVA | 56.41±8.74 | 2.52 | 2.54E+00 | 12 | 1.30E-02 | * | [51.91, inf] | 7.00E-01 | 5.38E+00 | 0.77 |
| | LaTVA | 64.74±7.42 | 2.14 | 6.88E+00 | 12 | 8.46E-06 | **** | [60.93, inf] | 1.91E+00 | 2.74E+03 | 1.00 |
| | RaTVA | 61.54±14.81 | 4.28 | 2.70E+00 | 12 | 9.68E-03 | ** | [53.92, inf] | 7.50E-01 | 6.79E+00 | 0.82 |
| | A1 | 52.24±9.68 | 1.94 | 1.16E+00 | 25 | 1.29E-01 | ns | [48.94, inf] | 2.30E-01 | 7.57E-01 | 0.30 |
| | mTVA | 52.88±10.58 | 2.12 | 1.36E+00 | 25 | 9.24E-02 | ns | [49.27, inf] | 2.70E-01 | 9.47E-01 | 0.38 |
| | pTVA | 50.96±11.40 | 2.28 | 4.20E-01 | 25 | 3.38E-01 | ns | [47.07, inf] | 8.00E-02 | 4.50E-01 | 0.11 |
| | aTVA | 63.14±11.82 | 2.36 | 5.56E+00 | 25 | 4.45E-06 | **** | [59.10, inf] | 1.09E+00 | 4.79E+03 | 1.00 |
| | TVAs | 55.66±12.48 | 1.42 | 3.98E+00 | 77 | 7.72E-05 | **** | [53.29, inf] | 4.50E-01 | 2.69E+02 | 0.99 |

**Appendix 1—table 15.** Assessing the significance of the speaker age categorization task.

This table reports the significance of the speaker age categorization performance. 342 voice stimuli were used in the experiments: the original stimuli (N=18), directly reconstructed stimuli using the LIN and the VLS models (N=36), and brain-reconstructed stimuli (18 stimuli x 2 models x 4 regions of interest x 2 hemispheres, N=288). The participants were tasked with identifying the approximate age of the presented voice in each trial by clicking either the 'Younger' or 'Older' button. To evaluate the accuracy of the binary responses, we computed the classification accuracy for each participant and region of interest (ROI). We then utilized one-sample t-tests to compare the mean accuracy distribution across all participants to the chance level of 50%. s.e.m.=standard error of the mean. Here are reported the results of the statistical tests, t-value, degree of freedom (dof), p-value, degree of significance (unc. sig.), 95% confidence interval (CI95%), effect size (Cohen-d), Bayes Factor (BF10), and statistical power (power) for each model and ROI.

| Model | ROI | Accuracy (%) | s.e.m. | T | dof | p-val | unc. sig. | CI95% | cohen-d | BF10 | power |
|---|---|---|---|---|---|---|---|---|---|---|---|
| LIN | LA1 | 54.70 ± 9.89 | 3.50 | 1.34E+00 | 8 | 1.08E-01 | ns | [48.20, inf] | 4.50E-01 | 1.29E+00 | 0.34 |
| | RA1 | 57.41 ± 8.69 | 3.07 | 2.41E+00 | 8 | 2.12E-02 | * | [51.70, inf] | 8.00E-01 | 4.14E+00 | 0.71 |
| | LmTVA | 57.04 ± 7.77 | 2.75 | 2.56E+00 | 8 | 1.68E-02 | * | [51.93, inf] | 8.50E-01 | 4.94E+00 | 0.76 |
| | RmTVA | 42.36 ± 8.22 | 2.91 | n/a | n/a | n/a | n/a | n/a | n/a | n/a | n/a |
| | LpTVA | 57.78 ± 7.70 | 2.72 | 2.86E+00 | 8 | 1.06E-02 | * | [52.72, inf] | 9.50E-01 | 7.04E+00 | 0.83 |
| | RpTVA | 50.98 ± 9.61 | 3.40 | 2.90E-01 | 8 | 3.90E-01 | ns | [44.67, inf] | 1.00E-01 | 6.67E-01 | 0.08 |
| | LaTVA | 40.48 ± 9.52 | 3.37 | n/a | n/a | n/a | n/a | n/a | n/a | n/a | n/a |
| | RaTVA | 40.52 ± 7.04 | 2.49 | n/a | n/a | n/a | n/a | n/a | n/a | n/a | n/a |
| | A1 | 56.05 ± 9.41 | 2.28 | 2.65E+00 | 17 | 8.36E-03 | ** | [52.09, inf] | 6.30E-01 | 6.92E+00 | 0.82 |
| | mTVA | 49.70 ± 10.85 | 2.63 | n/a | n/a | n/a | n/a | n/a | n/a | n/a | n/a |
| | pTVA | 54.38 ± 9.34 | 2.27 | 1.93E+00 | 17 | 3.51E-02 | * | [50.44, inf] | 4.60E-01 | 2.22E+00 | 0.58 |
| | aTVA | 40.50 ± 8.37 | 2.03 | n/a | n/a | n/a | n/a | n/a | n/a | n/a | n/a |
| | TVAs | 48.19 ± 11.18 | 1.54 | n/a | n/a | n/a | n/a | n/a | n/a | n/a | n/a |
| VLS | LA1 | 72.22 ± 9.16 | 3.24 | 6.86E+00 | 8 | 6.48E-05 | **** | [66.20, inf] | 2.29E+00 | 4.52E+02 | 1.00 |
| | RA1 | 48.33 ± 5.77 | 2.04 | n/a | n/a | n/a | n/a | n/a | n/a | n/a | n/a |
| | LmTVA | 51.46 ± 7.76 | 2.74 | 5.30E-01 | 8 | 3.04E-01 | ns | [46.36, inf] | 1.80E-01 | 7.25E-01 | 0.12 |
| | RmTVA | 41.11 ± 6.57 | 2.32 | n/a | n/a | n/a | n/a | n/a | n/a | n/a | n/a |
| | LpTVA | 60.61 ± 5.67 | 2.00 | 5.29E+00 | 8 | 3.68E-04 | *** | [56.88, inf] | 1.76E+00 | 1.06E+02 | 1.00 |
| | RpTVA | 66.05 ± 6.65 | 2.35 | 6.83E+00 | 8 | 6.70E-05 | **** | [61.68, inf] | 2.28E+00 | 4.40E+02 | 1.00 |
| | LaTVA | 52.02 ± 8.33 | 2.94 | 6.90E-01 | 8 | 2.56E-01 | ns | [46.54, inf] | 2.30E-01 | 7.83E-01 | 0.15 |
| | RaTVA | 50.00 ± 7.53 | 2.66 | n/a | n/a | n/a | n/a | n/a | n/a | n/a | n/a |
| | A1 | 60.28 ± 14.19 | 3.44 | 2.99E+00 | 17 | 4.14E-03 | ** | [54.29, inf] | 7.00E-01 | 1.24E+01 | 0.89 |
| | mTVA | 46.29 ± 8.86 | 2.15 | n/a | n/a | n/a | n/a | n/a | n/a | n/a | n/a |
| | pTVA | 63.33 ± 6.75 | 1.64 | 8.14E+00 | 17 | 1.44E-07 | **** | [60.48, inf] | 1.92E+00 | 1.11E+05 | 1.00 |
| | aTVA | 51.01 ± 8.00 | 1.94 | 5.20E-01 | 17 | 3.05E-01 | ns | [47.63, inf] | 1.20E-01 | 5.49E-01 | 0.13 |
| | TVAs | 53.54 ± 10.69 | 1.47 | 2.41E+00 | 53 | 9.69E-03 | ** | [51.08, inf] | 3.30E-01 | 4.15E+00 | 0.77 |

**Appendix 1—table 16.** Assessing the significance of the speaker identity discrimination task.

This table reports the significance of the speaker identity discrimination performance. The participants listened to 684 voice stimuli with short breaks in between. Each trial contained 2 short sound samples, and the participants had to indicate whether the samples were from the same speaker or different speakers. We then utilized one-sample t-tests to compare the mean accuracy distribution across all participants to the chance level of 50%. s.e.m.=standard error of the mean. Here are reported the results of the statistical tests, t-value, degree of freedom (dof), p-value, degree of significance (unc. sig.), 95% confidence interval (CI95%), effect size (Cohen-d), Bayes Factor (BF10), and statistical power (power) for each model and ROI.

| Category | ROI | Accuracy VLS (%) | Accuracy LIN (%) | s.e.m. VLS | s.e.m. LIN | T VLS vs LIN | dof | p-val | unc. sig. | CI95% | cohen-d | BF10 | power |
|---|---|---|---|---|---|---|---|---|---|---|---|---|---|
| Gender | LA1 | 48.33±10.56 | 46.11±6.60 | 3.52 | 2.20 | 4.70E-01 | 9 | 6.48E-01 | ns | [–8.41, 12.85] | 0.240000 | 3.40E-01 | 0.10 |
| | RA1 | 60.00±5.98 | 53.33±8.31 | 1.99 | 2.77 | 1.86E+00 | 9 | 9.63E-02 | ns | [–1.46, 14.79] | 0.870000 | 1.08E+00 | 0.69 |
| | LmTVA | 51.67±5.58 | 50.56±10.08 | 1.86 | 3.36 | 4.10E-01 | 9 | 6.93E-01 | ns | [–5.05, 7.27] | 0.130000 | 3.32E-01 | 0.07 |
| | RmTVA | 46.67±9.03 | 43.33±8.53 | 3.01 | 2.84 | 1.20E+00 | 9 | 2.60E-01 | ns | [–2.94, 9.60] | 0.360000 | 5.51E-01 | 0.18 |
| | LpTVA | 63.89±7.95 | 49.44±9.44 | 2.65 | 3.15 | 3.03E+00 | 9 | 1.43E-02 | * | [3.65, 25.24] | 1.570000 | 4.66E+00 | 0.99 |
| | RpTVA | 62.22±4.84 | 53.33±10.60 | 1.61 | 3.53 | 1.95E+00 | 9 | 8.26E-02 | ns | [–1.41, 19.18] | 1.020000 | 1.21E+00 | 0.82 |
| | LaTVA | 60.00±5.44 | 50.56±8.77 | 1.81 | 2.92 | 2.68E+00 | 9 | 2.50E-02 | * | [1.49, 17.40] | 1.230000 | 3.00E+00 | 0.93 |
| | RaTVA | 50.00±9.94 | 45.56±9.88 | 3.31 | 3.29 | 1.15E+00 | 9 | 2.80E-01 | ns | [–4.30, 13.19] | 0.430000 | 5.26E-01 | 0.23 |
| | A1 | 54.17±10.37 | 49.72±8.33 | 2.38 | 1.91 | 1.52E+00 | 19 | 1.45E-01 | ns | [–1.67, 10.56] | 0.460000 | 6.25E-01 | 0.50 |
| | mTVA | 49.17±7.91 | 46.94±10.01 | 1.81 | 2.30 | 1.16E+00 | 19 | 2.58E-01 | ns | [–1.77, 6.21] | 0.240000 | 4.21E-01 | 0.18 |
| | pTVA | 63.06±6.64 | 51.39±10.23 | 1.52 | 2.35 | 3.57E+00 | 19 | 2.06E-03 | ** | [4.82, 18.51] | 1.320000 | 1.95E+01 | 1.00 |
| | aTVA | 55.00±9.44 | 48.06±9.67 | 2.17 | 2.22 | 2.66E+00 | 19 | 1.54E-02 | * | [1.49, 12.40] | 0.710000 | 3.59E+00 | 0.85 |
| | TVAs | 55.74±9.88 | 48.80±10.15 | 1.29 | 1.32 | 4.37E+00 | 59 | 5.05E-05 | **** | [3.77, 10.12] | 0.690000 | 4.08E+02 | 1.00 |
| Age | LA1 | 54.49±10.65 | 46.15±14.48 | 3.07 | 4.18 | 1.54E+00 | 12 | 1.50E-01 | ns | [–3.48, 20.14] | 0.630000 | 7.18E-01 | 0.55 |
| | RA1 | 50.00±8.01 | 44.23±11.50 | 2.31 | 3.32 | 1.24E+00 | 12 | 2.39E-01 | ns | [–4.38, 15.92] | 0.560000 | 5.25E-01 | 0.46 |
| | LmTVA | 51.28±10.26 | 50.00±9.81 | 2.96 | 2.83 | 2.70E-01 | 12 | 7.94E-01 | ns | [–9.17, 11.73] | 0.120000 | 2.87E-01 | 0.07 |
| | RmTVA | 54.49±10.65 | 57.69±12.85 | 3.07 | 3.71 | –7.20E-01 | 12 | 4.88E-01 | ns | [–12.97, 6.55] | 0.260000 | 3.47E-01 | 0.14 |
| | LpTVA | 45.51±11.14 | 50.00±10.34 | 3.22 | 2.98 | –1.17E+00 | 12 | 2.66E-01 | ns | [–12.87, 3.89] | 0.400000 | 4.90E-01 | 0.27 |
| | RpTVA | 56.41±8.74 | 50.64±10.57 | 2.52 | 3.05 | 1.74E+00 | 12 | 1.08E-01 | ns | [–1.47, 13.00] | 0.570000 | 9.09E-01 | 0.47 |
| | LaTVA | 64.74±7.42 | 48.72±13.01 | 2.14 | 3.76 | 5.25E+00 | 12 | 2.04E-04 | *** | [9.38, 22.68] | 1.450000 | 1.55E+02 | 1.00 |
| | RaTVA | 61.54±14.81 | 62.82±13.32 | 4.28 | 3.85 | –2.50E-01 | 12 | 8.08E-01 | ns | [–12.51, 9.95] | 0.090000 | 2.86E-01 | 0.06 |
| | A1 | 52.24±9.68 | 45.19±13.11 | 1.94 | 2.62 | 2.01E+00 | 25 | 5.55E-02 | ns | [–0.18, 14.28] | 0.600000 | 1.16E+00 | 0.84 |
| | mTVA | 52.88±10.58 | 53.85±12.06 | 2.12 | 2.41 | –3.00E-01 | 25 | 7.70E-01 | ns | [–7.65, 5.72] | 0.080000 | 2.16E-01 | 0.07 |
| | pTVA | 50.96±11.40 | 50.32±10.46 | 2.28 | 2.09 | 2.40E-01 | 25 | 8.14E-01 | ns | [–4.90, 6.19] | 0.060000 | 2.13E-01 | 0.06 |
| | aTVA | 63.14±11.82 | 55.77±14.94 | 2.36 | 2.99 | 2.16E+00 | 25 | 4.02E-02 | * | [0.35, 14.39] | 0.540000 | 1.50E+00 | 0.75 |
| | TVAs | 55.66±12.48 | 53.31±12.82 | 1.42 | 1.46 | 1.28E+00 | 77 | 2.03E-01 | ns | [–1.29, 6.00] | 0.180000 | 2.74E-01 | 0.36 |
| Identity | LA1 | 72.22±9.16 | 54.70±9.89 | 3.24 | 3.50 | 3.64E+00 | 8 | 6.61E-03 | ** | [6.41, 28.63] | 1.730000 | 8.84E+00 | 0.99 |
| | RA1 | 48.33±5.77 | 57.41±8.69 | 2.04 | 3.07 | –1.97E+00 | 8 | 8.49E-02 | ns | [–19.72, 1.57] | 1.160000 | 1.23E+00 | 0.86 |
| | LmTVA | 51.46±7.76 | 57.04±7.77 | 2.74 | 2.75 | –1.44E+00 | 8 | 1.87E-01 | ns | [–14.49, 3.34] | 0.680000 | 7.08E-01 | 0.43 |
| | RmTVA | 41.11±6.57 | 42.36±8.22 | 2.32 | 2.91 | –2.50E-01 | 8 | 8.08E-01 | ns | [–12.72, 10.22] | 0.160000 | 3.30E-01 | 0.07 |
| | LpTVA | 60.61±5.67 | 57.78±7.70 | 2.00 | 2.72 | 1.05E+00 | 8 | 3.23E-01 | ns | [–3.37, 9.02] | 0.390000 | 5.02E-01 | 0.18 |
| | RpTVA | 66.05±6.65 | 50.98±9.61 | 2.35 | 3.40 | 4.37E+00 | 8 | 2.39E-03 | ** | [7.11, 23.03] | 1.720000 | 2.01E+01 | 0.99 |
| | LaTVA | 52.02±8.33 | 40.48±9.52 | 2.94 | 3.37 | 2.00E+00 | 8 | 8.02E-02 | ns | [–1.75, 24.84] | 1.220000 | 1.29E+00 | 0.89 |
| | RaTVA | 50.00±7.53 | 40.52±7.04 | 2.66 | 2.49 | 2.60E+00 | 8 | 3.16E-02 | * | [1.07, 17.88] | 1.230000 | 2.59E+00 | 0.89 |
| | A1 | 60.28±14.19 | 56.05±9.41 | 3.44 | 2.28 | 9.20E-01 | 17 | 3.68E-01 | ns | [–5.42, 13.86] | 0.340000 | 3.54E-01 | 0.28 |
| | mTVA | 46.29±8.86 | 49.70±10.85 | 2.15 | 2.63 | –1.10E+00 | 17 | 2.86E-01 | ns | [–9.96, 3.13] | 0.330000 | 4.11E-01 | 0.27 |
| | pTVA | 63.33±6.75 | 54.38±9.34 | 1.64 | 2.27 | 3.46E+00 | 17 | 3.02E-03 | ** | [3.49, 14.41] | 1.070000 | 1.45E+01 | 0.99 |
| | aTVA | 51.01±8.00 | 40.50±8.37 | 1.94 | 2.03 | 3.17E+00 | 17 | 5.62E-03 | ** | [3.51, 17.51] | 1.250000 | 8.55E+00 | 1.00 |
| | TVAs | 53.54±10.69 | 48.19±11.18 | 1.47 | 1.54 | 2.80E+00 | 53 | 7.15E-03 | ** | [1.51, 9.18] | 0.480000 | 4.88E+00 | 0.94 |

**Appendix 1—table 17.** Comparing human listeners' performance in discriminating speaker identity-related information decoded with VLS versus LIN.

This table reports the significance of the VLS-LIN difference in the speaker identity categorization and discrimination performance. Paired t-tests were conducted between the scores of human listeners at discriminating the speaker gender (2 classes), age (2 classes), and identity (17 classes) of the 18 Test Stimuli reconstructed from the VLS features with those from LIN features. s.e.m.=standard error of the mean. Here are reported the results of the statistical tests, t-value, degree of freedom (dof), p-value, degree of significance (unc. sig.), 95% confidence interval (CI95%), effect size (Cohen-d), Bayes Factor (BF10), and statistical power (power) for each speaker identity information and ROI.

| Category | Model | ROI | Accuracy ROI (%) | Accuracy A1 (%) | s.e.m. ROI | s.e.m. A1 | T ROI vs A1 | dof | p-val | unc. sig. | CI95% | cohen-d | BF10 | power |
|---|---|---|---|---|---|---|---|---|---|---|---|---|---|---|
| Gender | LIN | mTVA | 46.94±10.01 | 49.72±8.33 | 2.30 | 1.91 | −9.30E-01 | 38 | 3.60E-01 | ns | [−8.83, 3.27] | 2.90E-01 | 4.35E-01 | 0.15 |
| | | pTVA | 51.39±10.23 | 49.72±8.33 | 2.35 | 1.91 | 5.50E-01 | 38 | 5.80E-01 | ns | [−4.46, 7.79] | 1.70E-01 | 3.48E-01 | 0.08 |
| | | aTVA | 48.06±9.67 | 49.72±8.33 | 2.22 | 1.91 | −5.70E-01 | 38 | 5.70E-01 | ns | [−7.59, 4.26] | 1.80E-01 | 3.51E-01 | 0.09 |
| | | TVAs | 48.80±10.15 | 49.72±8.33 | 1.32 | 1.91 | −3.60E-01 | 78 | 7.20E-01 | ns | [−5.99, 4.14] | 9.00E-02 | 2.77E-01 | 0.06 |
| | VLS | mTVA | 49.17±7.91 | 54.17±10.37 | 1.81 | 2.38 | −1.67E+00 | 38 | 1.00E-01 | ns | [−11.06, 1.06] | 5.30E-01 | 9.19E-01 | 0.37 |
| | | pTVA | 63.06±6.64 | 54.17±10.37 | 1.52 | 2.38 | 3.15E+00 | 38 | 0.00E+00 | ** | [3.17, 14.61] | 9.90E-01 | 1.21E+01 | 0.87 |
| | | aTVA | 55.00±9.44 | 54.17±10.37 | 2.17 | 2.38 | 2.60E-01 | 38 | 8.00E-01 | ns | [−5.68, 7.35] | 8.00E-02 | 3.17E-01 | 0.06 |
| | | TVAs | 55.74±9.88 | 54.17±10.37 | 1.29 | 2.38 | 6.00E-01 | 78 | 5.50E-01 | ns | [−3.64, 6.78] | 1.60E-01 | 3.05E-01 | 0.09 |
| Age | LIN | mTVA | 53.85±12.06 | 45.19±13.11 | 2.41 | 2.62 | 2.43E+00 | 50 | 2.00E-02 | * | [1.50, 15.81] | 6.70E-01 | 2.95E+00 | 0.66 |
| | | pTVA | 50.32±10.46 | 45.19±13.11 | 2.09 | 2.62 | 1.53E+00 | 50 | 1.30E-01 | ns | [−1.61, 11.87] | 4.20E-01 | 7.23E-01 | 0.32 |
| | | aTVA | 55.77±14.94 | 45.19±13.11 | 2.99 | 2.62 | 2.66E+00 | 50 | 1.00E-02 | * | [2.59, 18.56] | 7.40E-01 | 4.65E+00 | 0.74 |
| | | TVAs | 53.31±12.82 | 45.19±13.11 | 1.46 | 2.62 | 2.75E+00 | 102 | 1.00E-02 | ** | [2.27, 13.97] | 6.20E-01 | 5.95E+00 | 0.78 |
| | VLS | mTVA | 52.88±10.58 | 52.24±9.68 | 2.12 | 1.94 | 2.20E-01 | 50 | 8.20E-01 | ns | [−5.12, 6.40] | 6.00E-02 | 2.84E-01 | 0.06 |
| | | pTVA | 50.96±11.40 | 52.24±9.68 | 2.28 | 1.94 | −4.30E-01 | 50 | 6.70E-01 | ns | [−7.29, 4.73] | 1.20E-01 | 3.00E-01 | 0.07 |
| | | aTVA | 63.14±11.82 | 52.24±9.68 | 2.36 | 1.94 | 3.56E+00 | 50 | 0.00E+00 | *** | [4.76, 17.04] | 9.90E-01 | 3.70E+01 | 0.94 |
| | | TVAs | 55.66±12.48 | 52.24±9.68 | 1.42 | 1.94 | 1.26E+00 | 102 | 2.10E-01 | ns | [−1.95, 8.79] | 2.90E-01 | 4.67E-01 | 0.24 |
| Identity | LIN | mTVA | 49.70±10.85 | 56.05±9.41 | 2.63 | 2.28 | −1.82E+00 | 34 | 8.00E-02 | ns | [−13.43, 0.72] | 6.10E-01 | 1.15E+00 | 0.43 |
| | | pTVA | 54.38±9.34 | 56.05±9.41 | 2.27 | 2.28 | −5.20E-01 | 34 | 6.10E-01 | ns | [−8.21, 4.86] | 1.70E-01 | 3.58E-01 | 0.08 |
| | | aTVA | 40.50±8.37 | 56.05±9.41 | 2.03 | 2.28 | −5.09E+00 | 34 | 0.00E+00 | **** | [−21.76,−9.35] | 1.70E+00 | 1.25E+03 | 1.00 |
| | | TVAs | 48.19±11.18 | 56.05±9.41 | 1.54 | 2.28 | −2.65E+00 | 70 | 1.00E-02 | * | [−13.79,−1.94] | 7.20E-01 | 4.69E+00 | 0.74 |
| | VLS | mTVA | 46.29±8.86 | 60.28±14.19 | 2.15 | 3.44 | −3.45E+00 | 34 | 0.00E+00 | ** | [−22.24,−5.75] | 1.15E+00 | 2.21E+01 | 0.92 |
| | | pTVA | 63.33±6.75 | 60.28±14.19 | 1.64 | 3.44 | 8.00E-01 | 34 | 4.30E-01 | ns | [−4.69, 10.79] | 2.70E-01 | 4.13E-01 | 0.12 |
| | | aTVA | 51.01±8.00 | 60.28±14.19 | 1.94 | 3.44 | −2.35E+00 | 34 | 2.00E-02 | * | [−17.30,−1.24] | 7.80E-01 | 2.53E+00 | 0.63 |
| | | TVAs | 53.54±10.69 | 60.28±14.19 | 1.47 | 3.44 | −2.09E+00 | 70 | 4.00E-02 | * | [−13.16,−0.31] | 5.70E-01 | 1.64E+00 | 0.54 |

