## [Editor Report · eLife Assessment]

This study used deep neural networks (DNN) to reconstruct voice information (viz., speaker identity), from fMRI responses in the auditory cortex and temporal voice areas, and assessed the representational content in these areas with decoding. A DNN-derived feature space approximated the neural representation of speaker identity-related information. The findings are **valuable** and the approach **solid**, yielding insight into how a specific model architecture can be used to relate the latent spaces of neural data and auditory stimuli to each other.

---

## [Referee Report · Reviewer #1 (Public review)]

Summary:

In this study, the authors trained a variational autoencoder (VAE) to create a high-dimensional "voice latent space" (VLS) using extensive voice samples, and analyzed how this space corresponds to brain activity through fMRI studies focusing on the temporal voice areas (TVAs). Their analyses included encoding and decoding techniques, as well as representational similarity analysis (RSA), which showed that the VLS could effectively map onto and predict brain activity patterns, allowing for the reconstruction of voice stimuli that preserve key aspects of speaker identity.

Strengths:

This paper is well-written and easy to follow. Most of the methods and results were clearly described. The authors combined a variety of analytical methods in neuroimaging studies, including encoding, decoding, and RSA. In addition to commonly used DNN encoding analysis, the authors performed DNN decoding and resynthesized the stimuli using VAE decoders. Furthermore, in addition to machine learning classifiers, the authors also included human behavioral tests to evaluate the reconstruction performance.

Weaknesses:

This manuscript presents a variational autoencoder (VAE) model to study voice identity representations from brain activity. While the model's ability to preserve speaker identity is expected due to its reconstruction objective, its broader utility remains unclear. Specifically, the VAE is not benchmarked against state-of-the-art speech models such as Wav2Vec2, HuBERT, or Whisper, which have demonstrated strong performance on standard speech tasks and alignment with cortical responses. Without comparisons on downstream tasks like automatic speech recognition (ASR) or phoneme classification, it is difficult to assess the relevance or advantages of the VLS representation.

Furthermore, the neural basis of the observed correlations between VLS and brain activity is not well characterized. It remains unclear whether the VLS aligns with high-level abstract identity representations or lower-level acoustic features like pitch. Prior studies (e.g., Tang et al., Science 2017; Feng et al., NeuroImage 2021) have shown both types of coding in STG. The experimental design also does not clarify whether speech content was controlled across speakers, raising concerns about confounding acoustic-phonetic features. For example, PC2 in Figure 1b appears to reflect absolute pitch height, suggesting that identity decoding may partly rely on simpler acoustic cues. A more detailed analysis of the representational content of VLS would strengthen the conclusions.

---

## [Referee Report · Reviewer #2 (Public review)]

Summary:

Lamothe et al. collected fMRI responses to many voice stimuli in 3 subjects. The authors trained two different autoencoders on voice audio samples and predicted latent space embeddings from the fMRI responses, allowing the voice spectrograms to be reconstructed. The degree to which reconstructions from different auditory ROIs correctly represented speaker identity, gender or age was assessed by machine classification and human listener evaluations. Complementing this, the representational content was also assessed using representational similarity analysis. The results broadly concur with the notion that temporal voice areas are sensitive to different types of categorical voice information.

Strengths:

The single-subject approach that allow thousands of responses to unique stimuli to be recorded and analyzed is powerful. The idea of using this approach to probe cortical voice representations is strong and the experiment is technically solid.

---

## [Referee Report · Reviewer #3 (Public review)]

Summary:

In this manuscript, Lamothe et al. sought to identify the neural substrates of voice identity in the human brain by correlating fMRI recordings with the latent space of a variational autoencoder (VAE) trained on voice spectrograms. They used encoding and decoding models, and showed that the "voice" latent space (VLS) of the VAE performs, in general, (slightly) better than a linear autoencoder's latent space. Additionally, they showed dissociations in the encoding of voice identity across the temporal voice areas.

Strengths:

The geometry of the neural representations of voice identity has not been studied so far. Previous studies on the content of speech and faces in vision suggest that such geometry could exist. This study demonstrates this point systematically, leveraging a specifically trained variational autoencoder.

The size of the voice dataset and the length of the fMRI recordings ensure that the findings are robust.

Comments on revisions:

The authors addressed my previous recommendations.

---

## [Author Response]

The following is the authors’ response to the original reviews.

**Reviewer #1 (Public Review):**
Summary:In this study, the authors trained a variational autoencoder (VAE) to create a high-dimensional "voice latent space" (VLS) using extensive voice samples, and analyzed how this space corresponds to brain activity through fMRI studies focusing on the temporal voice areas (TVAs). Their analyses included encoding and decoding techniques, as well as representational similarity analysis (RSA), which showed that the VLS could effectively map onto and predict brain activity patterns, allowing for the reconstruction of voice stimuli that preserve key aspects of speaker identity.Strengths:This paper is well-written and easy to follow. Most of the methods and results were clearly described. The authors combined a variety of analytical methods in neuroimaging studies, including encoding, decoding, and RSA. In addition to commonly used DNN encoding analysis, the authors performed DNN decoding and resynthesized the stimuli using VAE decoders. Furthermore, in addition to machine learning classifiers, the authors also included human behavioral tests to evaluate the reconstruction performance.Weaknesses:This manuscript presents a variational autoencoder (VAE) to evaluate voice identity representations from brain recordings. However, the study's scope is limited by testing only one model, leaving unclear how generalizable or impactful the findings are. The preservation of identity-related information in the voice latent space (VLS) is expected, given the VAE model's design to reconstruct original vocal stimuli. Nonetheless, the study lacks a deeper investigation into what specific aspects of auditory coding these latent dimensions represent. The results in Figure 1c-e merely tested a very limited set of speech features. Moreover, there is no analysis of how these features and the whole VAE model perform in standard speech tasks like speech recognition or phoneme recognition. It is not clear what kind of computations the VAE model presented in this work is capable of. Inclusion of comparisons with state-of-the-art unsupervised or self-supervised speech models known for their alignment with auditory cortical responses, such as Wav2Vec2, HuBERT, and Whisper, would strengthen the validation of the VAE model and provide insights into its relative capabilities and limitations.The claim that the VLS outperforms a linear model (LIN) in decoding tasks does not significantly advance our understanding of the underlying brain representations. Given the complexity of auditory processing, it is unsurprising that a nonlinear model would outperform a simpler linear counterpart. The study could be improved by incorporating a comparative analysis with alternative models that differ in architecture, computational strategies, or training methods. Such comparisons could elucidate specific features or capabilities of the VLS, offering a more nuanced understanding of its effectiveness and the computational principles it embodies. This approach would allow the authors to test specific hypotheses about how different aspects of the model contribute to its performance, providing a clearer picture of the shared coding in VLS and the brain.The manuscript overlooks some crucial alternative explanations for the discriminant representation of vocal identity. For instance, the discriminant representation of vocal identity can be either a higher-level abstract representation or a lower-level coding of pitch height. Prior studies using fMRI and ECoG have identified both types of representation within the superior temporal gyrus (STG) (e.g., Tang et al., Science 2017; Feng et al., NeuroImage 2021). Additionally, the methodology does not clarify whether the stimuli from different speakers contained identical speech content. If the speech content varied across speakers, the approach of averaging trials to obtain a mean vector for each speaker-the "identity-based analysis"-may not adequately control for confounding acoustic-phonetic features. Notably, the principal component 2 (PC2) in Figure 1b appears to correlate with absolute pitch height, suggesting that some aspects of the model's effectiveness might be attributed to simpler acoustic properties rather than complex identity-specific information.Methodologically, there are issues that warrant attention. In characterizing the autoencoder latent space, the authors initialized logistic regression classifiers 100 times and calculated the tstatistics using degrees of freedom (df) of 99. Given that logistic regression is a convex optimization problem typically converging to a global optimum, these multiple initializations of the classifier were likely not entirely independent. Consequently, the reported degrees of freedom and the effect size estimates might not accurately reflect the true variability and independence of the classifier outcomes. A more careful evaluation of these aspects is necessary to ensure the statistical robustness of the results.

We thank Reviewer #1 for their thoughtful and constructive comments. Below, we address the key points raised:

New comparitive models. We agree there are still many open questions on the structure of the VLS and the specific aspects of auditory coding that its latent dimensions represent. The features tested in Figure 1c-e are not speech features, but aspects related to speaker identity: age, gender and unique identity. Nevertheless we agree the VLS could be compared to recent speech models (not available when we started this project): we have now included comparisons with Wav2Vec and HuBERT in the encoding section (new Figure 2-S3). The comparison of encoding results based on LIN, the VLS, Wav2Vec and HuBERT (new Fig2S3) indicates no clear superiority of one model over the others; rather, different sets of voxels are better explained by the different models. Interestingly all four models yielded best encoding results for the m and a TVA, indicating some consistency across models.

On decoding directly from spectrograms. We have now added decoding results obtained directly from spectrograms, as requested in the private review. These are presented in the revised Figure 4, and allow for comparison with the LIN- and VLS-based reconstructions. As noted, spectrogram-based reconstructions sounded less vocal-like and faithful to the original, confirming that the latent spaces capture more abstract and cerebral-like voice representations.

On the number and length of stimuli. The rationale for using a large number of brief, randomly spliced speech excerpts from different languages was to extract identity features independent of specific linguistic cues. Indeed, the PC2 could very well correlate with pitch; we were not able to extract reliable f0 information from the thousands of brief stimuli, many of which are largely inharmonic (e.g., fricatives), such that this assumption could not be tested empirically. But it would be relevant that the weight of PC2 correlates with pitch: although the average fundamental frequency of phonation is not a linguistic cue, it is a major acoustical feature differentiating speaker identities.

Statistics correction. To address the issue of potential dependence between multiple runs of logistic regression, we replaced our previous analysis with a Wilcoxon signedrank test comparing decoding accuracies to chance. The results remain significant across classifications, and the revised figure and text reflect this change.

**Reviewer #2 (Public Review):**
Summary:Lamothe et al. collected fMRI responses to many voice stimuli in 3 subjects. The authors trained two different autoencoders on voice audio samples and predicted latent space embeddings from the fMRI responses, allowing the voice spectrograms to be reconstructed. The degree to which reconstructions from different auditory ROIs correctly represented speaker identity, gender, or age was assessed by machine classification and human listener evaluations. Complementing this, the representational content was also assessed using representational similarity analysis. The results broadly concur with the notion that temporal voice areas are sensitive to different types of categorical voice information.Strengths:The single-subject approach that allows thousands of responses to unique stimuli to be recorded and analyzed is powerful. The idea of using this approach to probe cortical voice representations is strong and the experiment is technically solid.Weaknesses:The paper could benefit from more discussion of the assumptions behind the reconstruction analyses and the conclusions it allows. The authors write that reconstruction of a stimulus from brain responses represents 'a robust test of the adequacy of models of brain activity' (L138). I concur that stimulus reconstruction is useful for evaluating the nature of representations, but the notion that they can test the adequacy of the specific autoencoder presented here as a model of brain activity should be discussed at more length. Natural sounds are correlated in many feature dimensions and can therefore be summarized in several ways, and similar information can be read out from different model representations. Models trained to reconstruct natural stimuli can exploit many correlated features and it is quite possible that very different models based on different features can be used for similar reconstructions. Reconstructability does not by itself imply that the model is an accurate brain model. Non-linear networks trained on natural stimuli are arguably not tested in the same rigorous manner as models built to explicitly account for computations (they can generate predictions and experiments can be designed to test those predictions). While it is true that there is increasing evidence that neural network embeddings can predict brain data well, it is still a matter of debate whether good predictability by itself qualifies DNNs as 'plausible computational models for investigating brain processes' (L72). This concern is amplified in the context of decoding and naturalistic stimuli where many correlated features can be represented in many ways. It is unclear how much the results hinge on the specificities of the specific autoencoder architectures used. For instance, it would be useful to know the motivations for why the specific VAE used here should constitute a good model for probing neural voice representations.Relatedly, it is not clear how VAEs as generative models are motivated as computational models of voice representations in the brain. The task of voice areas in the brain is not to generate voice stimuli but to discriminate and extract information. The task of reconstructing an input spectrogram is perhaps useful for probing information content, but discriminative models, e.g., trained on the task of discriminating voices, would seem more obvious candidates. Why not include discriminatively trained models for comparison?The autoencoder learns a mapping from latent space to well-formed voice spectrograms. Regularized regression then learns a mapping between this latent space and activity space. All reconstructions might sound 'natural', which simply means that the autoencoder works. It would be good to have a stronger test of how close the reconstructions are to the original stimulus. For instance, is the reconstruction the closest stimulus to the original in latent space coordinates out of using the experimental stimuli, or where does it rank? How do small changes in beta amplitudes impact the reconstruction? The effective dimensionality of the activity space could be estimated, e.g. by PCA of the voice samples' contrast maps, and it could then be estimated how the main directions in the activity space map to differences in latent space. It would be good to get a better grasp of the granularity of information that can be decoded/ reconstructed.What can we make of the apparent trend that LIN is higher than VLS for identity classification (at least VLS does not outperform LIN)? A general argument of the paper seems to be that VLS is a better model of voice representations compared to LIN as a 'control' model. Then we would expect VLS to perform better on identity classification. The age and gender of a voice can likely be classified from many acoustic features that may not require dedicated voice processing.The RDM results reported are significant only for some subjects and in some ROIs. This presumably means that results are not significant in the other subjects. Yet, the authors assert general conclusions (e.g. the VLS better explains RDM in TVA than LIN). An assumption typically made in single-subject studies (with large amounts of data in individual subjects) is that the effects observed and reported in papers are robust in individual subjects. More than one subject is usually included to hint that this is the case. This is an intriguing approach. However, reports of effects that are statistically significant in some subjects and some ROIs are difficult to interpret. This, in my view, runs contrary to the logic and leverage of the single-subject approach. Reporting results that are only significant in 1 out of 3 subjects and inferring general conclusions from this seems less convincing.The first main finding is stated as being that '128 dimensions are sufficient to explain a sizeable portion of the brain activity' (L379). What qualifies this? From my understanding, only models of that dimensionality were tested. They explain a sizeable portion of brain activity, but it is difficult to follow what 'sizable' is without baseline models that estimate a prediction floor and ceiling. For instance, would autoencoders that reconstruct any spectrogram (not just voice) also predict a sizable portion of the measured activity? What happens to reconstruction results as the dimensionality is varied?A second main finding is stated as being that the 'VLS outperforms the LIN space' (L381). It seems correct that the VAE yields more natural-sounding reconstructions, but this is a technical feature of the chosen autoencoding approach. That the VLS yields a 'more brain-like representational space' I assume refers to the RDM results where the RDM correlations were mainly significant in one subject. For classification, the performance of features from the reconstructions (age/ gender/ identity) gives results that seem more mixed, and it seems difficult to draw a general conclusion about the VLS being better. It is not clear that this general claim is well supported.It is not clear why the RDM was not formed based on the 'stimulus GLM' betas. The 'identity GLM' is already biased towards identity and it would be stronger to show associations at the stimulus level.Multiple comparisons were performed across ROIs, models, subjects, and features in the classification analyses, but it is not clear how correction for these multiple comparisons was implemented in the statistical tests on classification accuracies.Risks of overfitting and bias are a recurrent challenge in stimulus reconstruction with fMRI. It would be good with more control analyses to ensure that this was not the case. For instance, how were the repeated test stimuli presented? Were they intermingled with the other stimuli used for training or presented in separate runs? If intermingled, then the training and test data would have been preprocessed together, which could compromise the test set. The reconstructions could be performed on responses from independent runs, preprocessed separately, as a control. This should include all preprocessing, for instance, estimating stimulus/identity GLMs on separately processed run pairs rather than across all runs. Also, it would be good to avoid detrending before GLM denoising (or at least testing its effects) as these can interact.

We appreciate Reviewer #2’s careful reading and numerous suggestions for improving clarity and presentation. We have implemented the suggested text edits, corrected ambiguities, and clarified methodological details throughout the manuscript. In particular, we have toned down several sentences that we agree were making strong claims (L72, L118, L378, L380-381).

Clarifications, corrections and additional information:

We streamlined the introduction by reducing overly specific details and better framing the VLS concept before presenting specifics.

Clarified the motivation for the age classification split and corrected several inaccuracies and ambiguities in the methods, including the hearing thresholds, balancing of category levels, and stimulus energy selection procedure.

Provided additional information on the temporal structure of runs and experimental stimuli selection.

Corrected the description of technical issues affecting one participant and ensured all acronyms are properly defined in the text and figure legends.

Confirmed that audiograms were performed repeatedly to monitor hearing thresholds and clarified our use of robust scaling and normalization procedures.

Regarding the test of RDM correlations, we clarified in the text that multiple comparisons were corrected using a permutation-based framework.

**Reviewer #3 (Public Review):**
Summary:In this manuscript, Lamothe et al. sought to identify the neural substrates of voice identity in the human brain by correlating fMRI recordings with the latent space of a variational autoencoder (VAE) trained on voice spectrograms. They used encoding and decoding models, and showed that the "voice" latent space (VLS) of the VAE performs, in general, (slightly) better than a linear autoencoder's latent space. Additionally, they showed dissociations in the encoding of voice identity across the temporal voice areas.Strengths:The geometry of the neural representations of voice identity has not been studied so far. Previous studies on the content of speech and faces in vision suggest that such geometry could exist. This study demonstrates this point systematically, leveraging a specifically trained variational autoencoder.The size of the voice dataset and the length of the fMRI recordings ensure that the findings are robust.Weaknesses:Overall, the VLS is often only marginally better than the linear model across analysis, raising the question of whether the observed performance improvements are due to the higher number of parameters trained in the VAE, rather than the non-linearity itself. A fair comparison would necessitate that the number of parameters be maintained consistently across both models, at least as an additional verification step.The encoding and RSM results are quite different. This is unexpected, as similar embedding geometries between the VLS and the brain activations should be reflected by higher correlation values of the encoding model.The consistency across participants is not particularly high, for instance, S1 seemed to have demonstrated excellent performances, while S2 showed poor performance.An important control analysis would be to compare the decoding results with those obtained by a decoder operating directly on the latent spaces, in order to further highlight the interest of the non-linear transformations of the decoder model. Currently, it is unclear whether the non-linearity of the decoder improves the decoding performance, considering the poor resemblance between the VLS and brain-reconstructed spectrograms.

We thank Reviewer #3 for their comments. In response:

Code and preprocessed data are now available as indicated in the revised manuscript.

While we appreciate the suggestion to display supplementary analyses as boxplots split by hemisphere, we opted to retain the current format as we do not have hypotheses regarding hemispheric lateralization, and the small sample size per hemisphere would preclude robust conclusions.

Confirmed that the identities in Figure 3a are indeed ordered by age and have clarified this in the legend.

The higher variance observed in correlations for the aTVA in Figure 3b reflects the small number of data points (3 participants × 2 hemispheres), and this is now explained.

Regarding the cerebral encoding of gender and age, we acknowledge this interesting pattern. Prior work (e.g., Charest et al., 2013) found overlapping processing regions for voice gender without clear subregional differences in the TVAs. Evidence on voice age encoding remains sparse, and we highlight this novel finding in our discussion.

We again thank the reviewers for their insightful comments, which have greatly improved the quality and clarity of our work.

**Reviewer #1 (Recommendations For The Authors):**
(1) A set of recent advances have shown that embeddings of unsupervised/self-supervised speech models aligned to auditory responses to speech in the temporal cortex (e.g. Wav2Vec2: Millet et al NeurIPS 2022; HuBERT: Li et al. Nat Neurosci 2023; Whisper: Goldstein et al.bioRxiv 2023). These models are known to preserve a variety of speech information (phonetics, linguistic information, emotions, speaker identity, etc) and perform well in a variety of downstream tasks. These other models should be evaluated or at least discussed in the study.

We fully agree - the pace of progress in this area of voice technology has been incredible. Many of these models were not yet available at the time this work started so we could not use them in our comparison with cerebral representations.

We have now implemented Reviewer #1’s suggestion and evaluated Wav2Vec and HuBERT. The results are presented in supplementary Figure 2-S3. Correlations between activity predicted by the model and the real activity were globally comparable with those obtained with the LIN and VLS models. Interestingly both HuBERT and Wav2Vec yielded highest correlations in the mTVA, and to a lesser extent, the aTVA, as the LIN and VLS models.

(2) The test statistics of the results in Fig 1c-e need to be revised. Given that logistic regression is a convex optimization problem typically converging to a global optimum, these multiple initializations of the classifier were likely not entirely independent. Consequently, the reported degrees of freedom and the effect size estimates might not accurately reflect the true variability and independence of the classifier outcomes. A more careful evaluation of these aspects is necessary to ensure the statistical robustness of the results.

We thank Reviewer #1 for pointing out this important issue regarding the potential dependence between multiple runs of the logistic regression model. To address this concern, we have revised our analyses and used a Wilcoxon signed-rank test to compare the decoding accuracy to chance level. The results showed that the accuracy was significantly above chance for all classifications (Wilcoxon signed-rank test, all W=15, p=0.03125). We updated Figure 1c-e and the corresponding text (L154-L155) to reflect the revised analysis. Because the focus of this section is to probe the informational content of the autoencoder’s latent spaces, and since there are only 5 decoding accuracy values per model, we dropped the inter-model statistical test.

(3) In Line 198, the authors discuss the number of dimensions used in their models. To provide a comprehensive comparison, it would be informative to include direct decoding results from the original spectrograms alongside those from the VLS and LIN models. Given the vast diversity in vocal speech characteristics, it is plausible that the speaker identities might correlate with specific speech-related features also represented in both the auditory cortex and the VLS. Therefore, a clearer understanding of the original distribution of voice identities in the untransformed auditory space would be beneficial. This addition would help ascertain the extent to which transformations applied by the VLS or LIN models might be capturing or obscuring relevant auditory information.

We have now implemented Reviewer #1’s suggestion. The graphs on the right panel b of revised Figure 4 now show decoding results obtained from the regression performed directly on the spectrograms, rather than on representations of them, for our two example test stimuli. They can be listened to and compared to the LIN- and VLS-based reconstructions in Supplementary Audio 2. Compared to the LIN and VLS, the SPEC-based reconstructions sounded much less vocal or similar to the original, indicating that the latent spaces indeed capture more abstract voice representations, more similar to cerebral ones.

**Reviewer #2 (Recommendations For The Authors):**
L31: 'in voice' > consider rewording (from a voice?).L33: consider splitting sentence (after interactions).L39: 'brain' after parentheses.L45-: certainly DNNs 'as a powerful tool' extend to audio (not just image and video) beyond their use in brain models.L52: listened to / heard.L63: use second/s consistently.L64: the reference to Figure 5D is maybe a bit confusing here in the introduction.

We thank Reviewer #2 for these recommendations, which we have implemented.

L79-88: this section is formulated in a way that is too detailed for the introduction text (confusing to read). Consider a more general introduction to the VLS concept here and the details of this study later.L99-: again, I think the experimental details are best saved for later. It's good to provide a feel for the analysis pipeline here, but some of the details provided (number of averages, denoising, preprocessing), are anyway too unspecific to allow the reader to fully follow the analysis.

Again, thank you for these suggestions for improving readability: we have modified the text accordingly.

L159: what was the motivation for classifying age as a 2-class classification problem? Rather than more classes or continuous prediction? How did you choose the age split?

The motivation for the 2 age classes was to align on the gender classification task for better comparison. The cutoff (30 years) was not driven by any scientific consideration, but by practical ones, based on the median age in our stimulus set. This is now clarified in the manuscript (L149).

L263: Is the test of RDM correlation>0 corrected for multiple comparisons across ROIs, subjects, and models?

The test of RDM correlation>0 was indeed corrected for multiple comparisons for models using the permutation-based ‘maximum statistics’ framework for multiple comparison correction (described in Giordano et al., 2023 and Maris & Oostenveld, 2007). This framework was applied for each ROI and subject. It was described in the Methods (L745) but not clearly enough in the text—we thank Reviewer #2 and clarified it in the text (L246, L260-L261).

L379: 'these stimuli' - weren't the experimental stimuli different from those used to train the V/AE?

We thank Reviewer #2 for spotting this issue. Indeed, the experimental stimuli are different from those used to train the models. We corrected the text to reflect this distinction (L84-L85).

L443: what are 'technical issues' that prevented subject 3 from participating in 48 runs??

We thank Reviewer #2 for pointing out the ambiguity in our previous statement. Participant 3 actually experienced personal health concerns that prevented them from completing the whole number of runs. We corrected this to provide a more accurate description (L442-L443).

L444: participants were instructed to 'stay in the scanner'!? Do you mean 'stay still', or something?

We thank the Reviewer for spotting this forgotten word. We have corrected the passage (L444).

L463: Hearing thresholds of 15 dB: do you mean that all had thresholds lower than 15 dB at all frequencies and at all repeated audiogram measurements?

We thank Reviewer #2 for spotting this error: we meant thresholds below 15dB HL. This has been corrected (L463). Indeed participants were submitted to several audiograms between fMRI sessions, to ensure no hearing loss could be caused by the scanner noise in these repeated sessions.

L472: were the 4 category levels balanced across the dataset (in number of occurrences of each category combination)?

The dataset was fully balanced, with an equal number of samples for each combination of language, gender, age, and identity. Furthermore, to minimize potential adaptation effects, the stimuli were also balanced within each run according to these categories, and identity was balanced across sessions. We made this clearer in Main voice stimuli (L492-L496).

L482: the test stimuli were selected as having high energy by the amplitude envelope. It is unclear what this means (how is the envelope extracted, what feature of it is used to measure 'high energy'?)

The selection of sounds with high energy was based on analyzing the amplitude envelope of each signal, which was extracted using the Hilbert transform and then filtered to refine the envelope. This envelope, which represents the signal's intensity over time, was used to measure the energy of each stimulus, and those that exceeded an arbitrary threshold were selected. From this pool of high-energy stimuli, likely including vowels, we selected six stimuli to be repeated during the scanning session, then reconstructed via decoding. This has been clarified in the text (L483-L484).

L500 was the audio filtered to account for the transfer function of the Sensimetrics headphones?

We did not perform any filtering, as the transfer function of the Sensimetrics is already very satisfactory as is. This has been clarified in the text (L503).

L500: what does 'comfortable level' correspond to and was it set per session (i.e. did it vary across sessions)?

By comfortable we mean around 85 dB SPL. The audio settings were kept similar across sessions. This has been added to the text (L504).

L526- does the normalization imply that the reconstructed spectrograms are normalized? Were the reconstructions then scaled to undo the normalization before inversion?

The paragraph on spectrogram standardization was not well placed inducing confusion. We have placed this paragraph in its more suitable location, in the Deep learning section (L545L550)

L606: does the identity GLM model the denoised betas from the first GLM or simply the BOLD data? The text indicates the latter, but I suspect the former.

Indeed: this has been clarified (L601-L602).

L704: could you unpack this a bit more? It is not easy to see why you specify the summing in the objective. Shouldn't this just be the ridge objective for a given voxel/ROI? Then you could just state it in matrix notation.

Thanks for pointing this out: we kept the formula unchanged but clarified the text, in particular specified that the voxel id is the ith index (L695).

L716: you used robust scaling for the classifications in latent space but haven't mentioned scaling here. Are we to assume that the same applies?

Indeed we also used robust scaling here, this is now made clear (L710-L711).

L720: Pearson correlation as a performance metric and its variance will depend on the choice of test/train split sizes. Can you show that the results generalize beyond your specific choices? Maybe the report explained variance as well to get a better idea of performance.

We used a standard 80/20 split. We think it is beyond the scope of this study to examine the different possible choices of splits, and prefer not to spend additional time on this point which we think is relatively minor.

Could you specify (somewhere) the stimulus timing in a run? ISI and stimulus duration are mentioned in different places, but it would be nice to have a summary of the temporal structure of runs.

This is now clarified at the beginning of the Methods section (L437-441)

**Reviewer #3 (Recommendations For The Authors):**
Code and data are not currently available.

Code and preprocessed data are now available (L826-827).

In the supplementary material, it would be beneficial to present the different analyses as boxplots, as in the main text, but with the ROIs in the left and right hemispheres separated, to better show potential hemispheric effect. Although this information is available in the Supplementary Tables, it is currently quite tedious to access it.

Although we provide the complete data split by hemisphere in the Tables, we do not believe it is relevant to illustrate left/right differences, as we do not have any hypotheses regarding hemispheric lateralization–and we would be underpowered in any case to test them with only three points by hemisphere.

In Figure 3a, it might be beneficial to order the identities by age for each gender in order to more clearly illustrate the structure of the RDMs,

The identities are indeed already ordered by increasing age: we now make this clear.

In Figure 3b, the variance for the correlations for the aTVA is higher than in other regions, why?

Please note that the error bar indicates variance across only 6 data points (3 subjects x 2 hemispheres) such that some fluctuations are to be expected.

Please make sure that all acronyms are defined, and that they are redefined in the figure legends.

This has been done.

Gender and age are primarily encoded by different brain regions (Figure 5, pTVA vs aTVA). How does this finding compare with existing literature?

This interesting finding was not expected. The cerebral processing of voice gender has been investigated by several groups including ours (Charest et al., 2013, Cerebral Cortex). Using an fMRI-adaptation design optimized using a continuous carry-over protocol and voice gender continua generated by morphing, we found that regions dealing with acoustical differences between voices of varying gender largely overlapped with the TVAs, without clear differentiation between the different subparts. Evidence for the role of the different TVAs in voice age processing remains scarce.